# Transfer Learning Under High-Dimensional Graph Convolutional Regression Model for Node Classification

## Abstract

Node classification is a fundamental task, but obtaining node classification labels can be challenging and expensive in many real-world scenarios. Transfer learning has emerged as a promising solution to address this challenge by leveraging knowledge from source domains to enhance learning in a target domain. Existing transfer learning methods for node classification primarily focus on integrating Graph Convolutional Networks (GCNs) with various transfer learning techniques. While these approaches have shown promising results, they often suffer from a lack of theoretical guarantees, restrictive conditions, and high sensitivity to hyperparameter choices. To overcome these limitations, we employ a Graph Convolutional Multinomial Logistic Lasso Regression (GCR) model which simplifies GCN, and develop a transfer learning method called Trans-GCR based on the GCR model. We provide theoretical guarantees of the estimate obtained under the GCR model in high-dimensional settings. Moreover, Trans-GCR demonstrates superior empirical performance, has a low computational cost, and requires fewer hyperparameters than existing methods.

## 1 Introduction

Network (a.k.a graph) data is ubiquitous in various domains, including social networks (Barabási, 2013), citation networks (Ji et al., 2022), and biological networks (Zitnik et al., 2018; Han et al., 2019). A fundamental task in network analysis is node classification (Kipf & Welling, 2016), which aims to predict the class labels of a node based on its own features and its neighboring nodes' features. Usually, the node features are high-dimensional (Hamilton et al., 2017). Thus, we focus on high-dimensional settings in this paper. Nevertheless, obtaining node classification labels can be challenging and expensive in many real-world scenarios (Dai et al., 2022). For example, classifying genes into disease categories using a gene-gene interaction network faces a scarcity of disease labels, as experimentally annotating genes is expensive (Guney et al., 2016).

To address the challenge of limited labeled data, transfer learning, which uses knowledge from source domains to enhance learning in a target domain, has emerged as a promising solution (Dai et al., 2022). Continuing the aforementioned example, despite the scarcity of disease labels, abundant functional annotations of genes exist in curated databases like KEGG pathways (Kanehisa et al., 2012), offering ample source data. In the existing literature, various transfer learning methods based on Graph Convolutional Networks (GCNs) (Sperduti & Starita, 1997; Bruna et al., 2014; Defferrard et al., 2016; Kipf & Welling, 2017) have been proposed to enhance node classification accuracy. These GCN-based transfer learning methods can be broadly summarized into three main areas.

First, pre-training and fine-tuning approaches (Hu et al., 2019; 2020; Lu et al., 2021; Yang et al., 2022; Kooverjee et al., 2022; Xu et al., 2023), which usually pre-train a GCN on a large-scale dataset to learn transferable representations and then fine-tune the pre-trained model on a target task. While effective, these methods usually lack theoretical guarantees. Second, theoretical transferability analysis. Researchers have investigated the theoretical transferability properties of GCNs for graphs sampled from the same underlying space or graphon model (Nilsson & Bresson, 2020; Ruiz et al., 2020; Levie et al., 2021; Ruiz et al., 2023). Despite valuable theoretical insights, they often assume that the source and target domain are drawn from exactly the same underlying model, which usually

does not hold in practical scenarios where domain shifts occur. Third, to address the domain shifts challenge, various domain adaptation techniques have been proposed, such as unsupervised adaptation (Wu et al., 2020), local structure transfer (Zhu et al., 2021), adversarial domain alignment (Dai et al., 2022), and noise-resistant transfer (Yuan et al., 2023). Despite promising results, they often lack theoretical guarantees or can be sensitive to hyperparameter choices. In summary, existing methods suffer from a lack of theoretical guarantees, restrictive conditions, and high sensitivity to hyperparameters.

To address these limitations, we propose a novel statistical transfer learning framework based on a Graph Convolutional Multinomial Logistic Lasso Regression (GCR) model. The GCR model assumes that the classification label depends on the graph-aggregated node features (obtained through multiple graph aggregation layers), followed by a multinomial logistic lasso regression model which assumes a linear relationship between aggregated features and labels. This assumption is inspired by empirical observations suggesting that removing nonlinear activation functions (e.g., ReLU) in GCN's hidden layers achieves comparable performance to the original GCN architecture (Wu et al., 2019). Our GCR model extends beyond the work of (Wu et al., 2019) by introducing a rigorous statistical formulation with the ability to handle high-dimensional features. Building upon this GCR model, the main contribution of this paper is to develop a two-step transfer learning method Trans-GCR. Specifically, we let $\boldsymbol{\beta}^s$ and $\boldsymbol{\beta}^t = \boldsymbol{\beta}^s + \delta$ denote the GCR's high-dimensional sparse model parameters in source and target data, respectively, where $\delta$ measures the domain shift. In the first step, we obtain the estimate of source domain parameters, denoted as $\hat{\boldsymbol{\beta}}^s$, by minimizing the $l_1$-regularized negative likelihood function of the GCR model using source data. In the second step, we estimate the shift term $\delta$ by substituting $\boldsymbol{\beta}^t$ with $\hat{\boldsymbol{\beta}}^s + \delta$ in the GCR negative likelihood function and minimizing it using the target data. This step leverages the knowledge learned from $\hat{\boldsymbol{\beta}}^s$. Finally, our estimate of the target domain parameters is given by $\hat{\boldsymbol{\beta}}^t = \hat{\boldsymbol{\beta}}^s + \hat{\delta}$.

Our method enjoys the following advantages: (1) We demonstrate through extensive empirical studies that our proposed method achieves superior or comparable performance compared with existing complicated GCN-based transfer learning approaches for node classification. (2) We provide theoretical guarantees of the estimate obtained under GCR model in high-dimensional settings under mild conditions. (3) By leveraging the simplified model GCR, our method involves fewer parameters to be trained than more complex GCN-based models, resulting in reduced computational cost. (4) Our framework has only two hyperparameters, i.e., the number of graph aggregation layers and the $l_1-$norm penalty strength.

## 2 PRELIMINARIES

A graph with $n$ nodes is represented by an adjacency matrix $\mathbf{A} = (\mathbf{A}_{ij}) \in \{0, 1\}^{n \times n}, i, j = 1, \dots, n$, where $\mathbf{A}_{ij} = 1$ if there is an edge between nodes $i$ and $j$, and $\mathbf{A}_{ij} = 0$ otherwise. We only consider a graph with no self-loops, so all diagonal entries of $\mathbf{A}$ are 0. In addition, each node is associated with a $d$-dimensional covariates and a $C$-dimensional one-hot class label. The entire covariate matrix is $\mathbf{X} \in \mathbb{R}^{n \times d}$, and the entire classification label matrix is $\mathbf{Y} \in \{0, 1\}^{n \times C}$. Node classification aims to predict $\mathbf{Y}$ based on $\mathbf{X}$ and $\mathbf{A}$. The normalized adjacency matrix is

$$\mathbf{S} = \tilde{\mathbf{D}}^{-\frac{1}{2}} \tilde{\mathbf{A}} \tilde{\mathbf{D}}^{-\frac{1}{2}}, \text{ where } \tilde{\mathbf{A}} = \mathbf{A} + \mathbf{I}_n, \tag{2.1}$$

where $\mathbf{I}_n$ is an $n \times n$ identity matrix, $\tilde{\mathbf{A}}$ denote the adjacency matrix with added self-connections. Here, $\tilde{\mathbf{D}}$ is the degree matrix of $\tilde{\mathbf{A}}$, with diagonal entry $\tilde{\mathbf{D}}_{ii}$ representing the degree of node $i$, and all off-diagonal elements being zero.

### 2.1 RELATED WORK: GRAPH CONVOLUTIONAL NETWORKS

GCNs and their variants have gained increasing popularity for node classification tasks (Hamilton et al., 2017; Kipf & Welling, 2017; Wang, 2019; Chien et al., 2020). The core strength of GCNs lies in their ability to leverage the underlying graph structure to propagate and aggregate information from neighboring nodes. A standard two-layer GCN works by $\mathbf{H}^{(1)} = \text{Relu}(\mathbf{A}\mathbf{X}\mathbf{W}^{(1)}); \mathbf{H}^{(2)} = \text{Softmax}(\mathbf{A}\mathbf{H}^{(1)}\mathbf{W}^{(2)})$, where $\mathbf{W}^{(1)}$ and $\mathbf{W}^{(2)}$ are parameters to be learned. As pointed out by Wu et al. (2019), the key to GCN's success lies in its ability to perform graph convolutions, and not necessarily in its use of nonlinearity through activation functions such as ReLU. In fact, empirical

observations in Wu et al. (2019) suggested that removing the nonlinear activation function from the GCN's hidden layers does not substantially impact the model's performance. From a practical perspective, the removal of nonlinearities results in a simpler model, which offers several advantages. First, it reduces the computational complexity of the model, making training and inference faster and more efficient. This is particularly important in large-scale graph datasets, where computational resources may be limited. Second, the simplified architecture requires fewer hyperparameters, reducing tuning complexity and preventing overfitting.

## 2.2 HIGH-DIMENSIONAL GRAPH CONVOLUTIONAL MULTINOMIAL LOGISTIC LASSO REGRESSION MODEL

Motivated by the aforementioned observations, we propose the GCR model to model the relationship between $\mathbf{Y}$, $\mathbf{X}$, and $\mathbf{A}$. GCR assumes that the classification labels are not only affected by their own, but also their neighbors' features based on the graph structure captured by the adjacency matrix. Usually, a normalized adjacency matrix is used, and a common choice is $\mathbf{S}$ in Eq. 2.1. In what follows, we present the GCR model using $\mathbf{S}$. In GCR, there is a vector of coefficients $\boldsymbol{\beta}_c = (\boldsymbol{\beta}_{1c}, \ldots, \boldsymbol{\beta}_{dc})^T$ for each category $c = 1, \ldots, C$. To ensure the identifiability of $\boldsymbol{\beta}_c$, we take the last category as the reference category, i.e., $\boldsymbol{\beta}_C = \mathbf{0}_d$. In GCR model, given predictors $\mathbf{X}$ and $\mathbf{A}$, for each node $i = 1 \ldots, n$, the probability of node $i$'s classification label $\mathbf{Y}_i$ belonging to category $c$ is

$$\mathbf{P}_{ic} = \mathbb{P}(\mathbf{Y}_i = c | \mathbf{X}, \mathbf{A}) = \frac{\exp^{\sum_{j=1}^n \mathbf{S}_{ij}^M \mathbf{x}_j \boldsymbol{\beta}_c}}{1 + \sum_{l=1}^{C-1} \exp^{\sum_{j=1}^n \mathbf{S}_{ij}^M \mathbf{x}_j \boldsymbol{\beta}_l}}, c = 1, \ldots, C, \qquad (2.2)$$

where $\mathbf{X}_j = (\mathbf{X}_{j1}, \ldots, \mathbf{X}_{jd})$ is the $j$th row of $\mathbf{X}$, $\mathbf{S}_{ij}^M$ is the $(i, j)$th element in $\mathbf{S}^M$, $\mathbf{S}$ is defined in Eq. 2.1, and $M$ is the number of convolution layer. The sum $\sum_{j=1}^n \mathbf{S}_{ij}^M \mathbf{X}_j$ aggregates neighboring nodal features, encouraging neighboring nodes to have similar aggregated features, thus having similar classification labels. A larger value of $M$ enables the model to capture higher-order neighborhood dependencies. In the high-dimensional setting, where $d$ can be larger than $n$, we assume $\boldsymbol{\beta}_c$ to be sparse such that $s << d$, where $s$ is the number of nonzero elements of $\boldsymbol{\beta}_c$.

**Parameter estimation.** Given observed data $(\mathbf{A}, \mathbf{X}, \mathbf{Y})$, the parameter estimate $\hat{\boldsymbol{\beta}} = (\hat{\boldsymbol{\beta}}_1, \ldots, \hat{\boldsymbol{\beta}}_{C-1})$ is obtained by minimizing the following negative log-likelihood with $l_1$ regularization, $l(\boldsymbol{\beta}; \mathbf{Y}, \mathbf{X}, \mathbf{A}) = -\sum_{i=1}^n \sum_{c=1}^C (\mathbf{Y}_{ic} \log \mathbf{P}_{ic} + (1 - \mathbf{Y}_{ic}) \log(1 - \mathbf{P}_{ic})) + \lambda \sum_{c=1}^{C-1} ||\boldsymbol{\beta}_c||_1$, where $\mathbf{P}_{ic}$ is defined in Eq. 2.2, $|| \cdot ||_1$ is the $l_1-$norm, $\lambda \geq 0$ is a regularization hyperparameter controlling the trade-off between the log-likelihood and the penalty. Such regularization is commonly used in other high-dimensional penalized regression/classification problem (e.g., Ridge, Elastic Net etc.). $l(\boldsymbol{\beta}; \mathbf{Y}, \mathbf{X}, \mathbf{A})$ can be further simplified as

$$l(\boldsymbol{\beta}; \mathbf{Y}, \mathbf{X}, \mathbf{A}) = -\sum_{i=1}^n \sum_{c=1}^C (\mathbf{Y}_{ic} \sum_{j=1}^n \mathbf{S}_{ij}^M \mathbf{X}_j \boldsymbol{\beta}_c - \log(1 + e^{\sum_{j=1}^n \mathbf{S}_{ij}^M \mathbf{X}_j \boldsymbol{\beta}_c})) + \lambda \sum_{c=1}^{C-1} ||\boldsymbol{\beta}_c||_1$$
$$(2.3)$$

To minimize Eq. 2.3, we use the standard coordinate descent algorithm (Wright, 2015), which iteratively minimizes the objective function with respect to each coordinate of $\boldsymbol{\beta}$, while keeping the others fixed.

## 3 TRANSFER LEARNING

In this paper, we consider the following transfer learning problem. Let $(\mathbf{A}^{(0)}, \mathbf{X}^{(0)}, \mathbf{Y}^{(0)})$ denote the target data, where $\mathbf{A}^{(0)} \in \{0, 1\}^{n_0 \times n_0}$, $\mathbf{X}^{(0)} \in \mathbb{R}^{n_0 \times d}$, $\mathbf{Y}^{(0)} \in \{0, 1\}^{n_0 \times C}$. Let $\boldsymbol{\beta}^{(0)} = (\boldsymbol{\beta}_1^{(0)}, \ldots, \boldsymbol{\beta}_{C-1}^{(0)}) \in \mathbb{R}^{d \times (C-1)}$ denote the true coefficient matrix associated with target data under GCR model. Let $(\mathbf{A}^{(k)}, \mathbf{X}^{(k)}, \mathbf{Y}^{(k)})$ denote the $k$th source data, $k = 1, \ldots, K$, where $\mathbf{A}^{(k)} \in \{0, 1\}^{n_k \times n_k}$, $\mathbf{X}^{(k)} \in \mathbb{R}^{n_k \times d}$, $\mathbf{Y}^{(k)} \in \{0, 1\}^{n_k \times C}$. Let $\boldsymbol{\beta}^{(k)} = (\boldsymbol{\beta}_1^{(k)}, \ldots, \boldsymbol{\beta}_{C-1}^{(k)}) \in \mathbb{R}^{d \times (C-1)}$ denote the true coefficient matrix associated with the $k$th source data, for the $C - 1$ classes. Note that the last class's coefficient $\boldsymbol{\beta}_C^{(k)} = \mathbf{0}_d$ is fixed to avoid identifiability issues.

The difference between the $k$th source domain's coefficient and the target domain's coefficient is $\delta^{(k)} = \boldsymbol{\beta}^{(0)} - \boldsymbol{\beta}^{(k)}$, where $\delta^{(k)} \in \mathbb{R}^{d \times (C-1)}$. The overall domain shift for the $k$-th source domain is $(C - 1)^{-1} \sum_{c=1}^{C-1} ||\delta_c^{(k)}||_1$, where $\delta_c^{(k)}$ is the $c$th column of $\delta^{(k)}$. A source sample is defined

as $h-$level transferable if its domain shift level is lower than a threshold $h$. The set of $h-$level transferable source data is $\mathcal{A}_h = \{k : (C-1)^{-1}\sum_{c=1}^{C-1}||\delta_c^{(k)}||_1 \leq h\}$. To ensure that transferring sources within the set $\mathcal{A}_h$ is beneficial, $h$ should be reasonably small. In the subsequent sections, we will abbreviate the notation $\mathcal{A}_h$ as $\mathcal{A}$ for brevity without special emphasis.

### 3.1 Transfer Learning When the Transferable Source Set is Known

In this subsection, we present a transfer learning method under GCR (abbreviated as Trans-GCR) for the node classification task when the transferable source set $\mathcal{A}$ is known, i.e., we have prior knowledge of which source data to utilize. Our method Trans-GCR is motivated by the transfer learning literature in the conventional regression model (without considering graph structure) (Bastani, 2018; Li et al., 2022; Tian & Feng, 2023). Figure 1 shows the workflow of Trans-GCR, which works in the following steps. We first preprocess $\mathbf{A}^{(k)}$ to obtain the normalized adjacency matrices $\mathbf{S}^{(k)}$, $k \in \{0, \mathcal{A}\}$.

**Source sample pooling.** We then pooled all source domain data in $\mathcal{A}$ into a single source sample, which includes a normalized adjacency matrix $\mathbf{S}^{\mathcal{A}} \in \mathbb{R}^{n_{\mathcal{A}} \times n_{\mathcal{A}}}$, node features $\mathbf{X}^{\mathcal{A}} \in \mathbb{R}^{n_{\mathcal{A}} \times d}$, and labels $\mathbf{Y}^{\mathcal{A}} \in \{0,1\}^{n_{\mathcal{A}} \times C}$, where the total number of nodes is $n_{\mathcal{A}} = \sum_{k \in \mathcal{A}} n_k$. $\mathbf{S}^{\mathcal{A}}$ has a block structure with diagonal blocks corresponding to the individual $\mathbf{S}^{(k)}$ matrices, $k \in \mathcal{A}$. The pooled node features $\mathbf{X}^{\mathcal{A}}$ and labels $\mathbf{Y}^{\mathcal{A}}$ are obtained by concatenating the respective $\mathbf{X}^{(k)}$ and $\mathbf{Y}^{(k)}$ matrices row-wise, $k \in \mathcal{A}$. The rationale behind pooling the source samples in $\mathcal{A}$ together is due to the assumption that the source domains in $\mathcal{A}$ are similar to the target domain, with only small domain shifts, implying that they share similar underlying model parameters. By pooling the source samples (Wang, 2019), the algorithm effectively increases the sample size, which leads to a more accurate and stable estimate.

**Source domain parameter estimation.** Let $\boldsymbol{\beta}^{\mathcal{A}}$ denote the latent parameter in GCR that are used to generate $\mathbf{Y}^{\mathcal{A}}$. To obtain $\hat{\boldsymbol{\beta}}^{\mathcal{A}}$, we minimize the aforementioned negative log-likelihood with $l_1-$norm penalty in Eq. 2.3 using the pooled source samples. **Domain shift estimation.** Let $\delta^{\mathcal{A}}$ denote the difference between the pooled source parameters $\boldsymbol{\beta}^{\mathcal{A}}$ and the target domain parameters $\boldsymbol{\beta}^{(0)}$, i.e., $\boldsymbol{\beta}^{(0)} = \boldsymbol{\beta}^{\mathcal{A}} + \delta^{\mathcal{A}}$. We then estimate $\delta^{\mathcal{A}}$ by minimizing the following loss function, which essentially replaces $\boldsymbol{\beta}^{(0)}$ with $\hat{\boldsymbol{\beta}}^{\mathcal{A}} + \delta^{\mathcal{A}}$ in the likelihood function of GCR model for the target data. Specifically, we solve the following optimization problem to estimate $\delta^{\mathcal{A}}$:

$$-\sum_{i=1}^{n_0}\sum_{c=1}^{C}[\mathbf{Y}_{ic}^{(0)}(\sum_{j=1}^{n_0}s_{ij}\mathbf{X}_j^{(0)})(\hat{\boldsymbol{\beta}}_c^{\mathcal{A}}+\delta_c^{\mathcal{A}})-\psi(\sum_{j=1}^{n_0}s_{ij}\mathbf{X}_j^{(0)}(\hat{\boldsymbol{\beta}}_c^{\mathcal{A}}+\delta_c^{\mathcal{A}}))]+\lambda\sum_{c=1}^{C-1}||\hat{\boldsymbol{\beta}}_c^{\mathcal{A}}+\delta_c^{\mathcal{A}}||_1, \quad (3.1)$$

where $\psi(x) = \log(1+e^x)$, $s_{ij}$ is the $(i,j)$th element of $(\mathbf{S}^{(0)})^M$ for the target domain, $\hat{\boldsymbol{\beta}}_c^{\mathcal{A}}$ and $\delta_c^{\mathcal{A}}$ are the $c$th column of $\hat{\boldsymbol{\beta}}^{\mathcal{A}}$ and $\delta^{\mathcal{A}}$, respectively. This reformulation transforms the unknown parameter from $\boldsymbol{\beta}^{(0)}$ to $\delta^{\mathcal{A}}$. Finally, once the domain shift $\delta^{\mathcal{A}}$ is calculated, we compute the final target domain parameter estimate as $\hat{\boldsymbol{\beta}}^{(0)} = \hat{\boldsymbol{\beta}}^{\mathcal{A}} + \hat{\delta}^{\mathcal{A}}$.

We summarize our procedure in Algorithm 1 in Appendix G, which involves two hyperparameters: the graph convolution layers $M$ and the $l_1-$norm penalty strength $\lambda$. Note that while Trans-GCR requires two hyperparameters: $\lambda$ and $M$, it avoids additional hyperparameters required by GCN, such as dropout rates, learning rates, and hidden sizes, which require extensive tuning. When implementing Trans-GCR, we apply cross-validation procedures to select these hyperparameters.

### 3.2 Transferable Source Detection

In section 3.1, we presented a method when the transferable set $\mathcal{A}$ is known. Nevertheless, in real applications, such prior knowledge might be unavailable. When the source domain differs significantly from the target domain, negative transfer can occur, leading to decreased performance on the target task (Li et al., 2022; Tian & Feng, 2023). To address this issue, we propose a data-driven cross-validation approach to select the transferable set $\mathcal{A}$ automatically.

Our method begins by partitioning nodes in the target data into $V$ folds, i.e., $s_1, \ldots, s_V$. For each fold $v$, the labels of nodes in $s_v$ in the target data (denoted as $\mathbf{Y}_{s_v}^{(0)}$) are held out as the testing target data, while the labels in the remaining folds $\mathbf{Y}_{-s_v}^{(0)}$ serve as the training target data. We then apply the transfer learning Algorithm 1 using the $k$th source data $\{\mathbf{A}^{(k)}, \mathbf{X}^{(k)}, \mathbf{Y}^{(k)}\}$ and training target

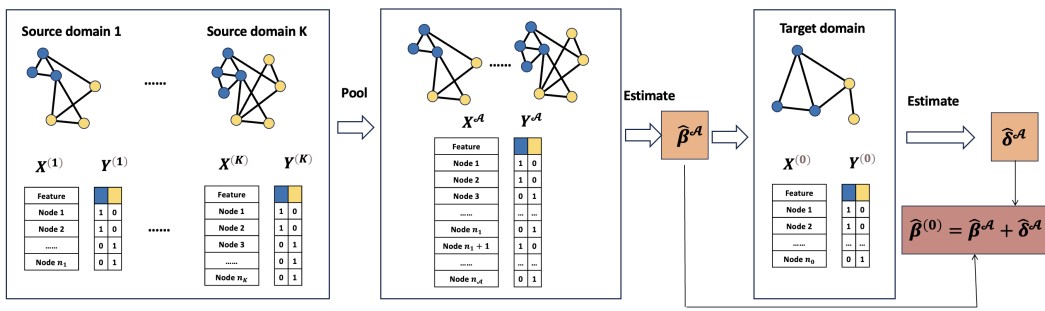

Figure 1: Workflow of Trans-GCR. We first pool all source domains to get an estimate $\hat{\boldsymbol{\beta}}^{\mathcal{A}}$. We then use target data and the knowledge from $\hat{\boldsymbol{\beta}}^{\mathcal{A}}$ to estimate domain shift $\hat{\delta}^{\mathcal{A}}$. The final estimate for the target data is $\hat{\boldsymbol{\beta}}^{(0)} = \hat{\boldsymbol{\beta}}^{\mathcal{A}} + \hat{\delta}^{\mathcal{A}}$.

data $\{\mathbf{A}^{(0)}, \mathbf{X}^{(0)}, \mathbf{Y}^{(0)}_{-s_v}\}$ to obtain the estimate $\hat{\boldsymbol{\beta}}^{(0)}_{vk}$ for the target data. Note that we use the entire network structure of the target data $\mathbf{A}^{(0)}$ and all node covariates $\mathbf{X}^{(0)}$ to obtain aggregated features. However, we only use the classification labels of the training nodes $\mathbf{Y}^{(0)}_{-s_v}$ when minimizing the negative log-likelihood in Eq. 2.3 during the estimation process.

Using the model estimate $\hat{\boldsymbol{\beta}}^{(0)}_{vk}$, we predict labels for the testing nodes in the target data, calculating $\hat{\mathbb{P}}(\mathbf{Y}^{(0)}_i = c)$, $i \in s_v$. The model's performance is assessed through the negative log-likelihood, $\mathrm{NL}^{(k)}_v = -\sum_{i \in s_v} \sum_{c=1}^C (\mathbf{Y}^{(0)}_{ic} \log \hat{\mathbb{P}}(\mathbf{Y}^{(0)}_i = c) + (1 - \mathbf{Y}^{(0)}_{ic}) \log(1 - \hat{\mathbb{P}}(\mathbf{Y}^{(0)}_i = c))$. We then average $\mathrm{NL}^{(k)}_v$ across all $V$ folds, yielding $\mathrm{NL}^{(k)} = V^{-1} \sum_{v=1}^V \mathrm{NL}^{(k)}_v$. A lower score of $\mathrm{NL}^{(k)}$ indicates higher transferability of the $k$th source data. We rank the $K$ sources by their corresponding $\mathrm{NL}^{(k)}$ values and select the top $L$ sources with the lowest scores as the estimated transferable set $\hat{\mathcal{A}}$, where $L$ is a user-defined hyperparameter specifying the number of source data to include. We summarize our procedure in Algorithm 2 in the Appendix G. Note that in practice, our cross-validation procedure can be efficiently implemented using parallel computing techniques.

# 4 THEORETICAL PROPERTIES

In this section, we present our main theoretical results. Establishing theoretical guarantees for the GCR model under high-dimensional network data is extremely challenging as: (1) The network dependency prevents the application of standard concentration results, valid only for i.i.d. data. For instance, standard M-estimation theory requires independence among observations to derive consistency and the rate of convergence. (2) The ultra-high dimensionality, where the dimension of $X$ can be much larger than $n$, further complicates the theoretical analysis. Analyzing any high-dimensional model is difficult because it requires a thorough understanding of high-dimensional geometry. Consequently, we need new tools to address these challenges, which is a key part of our theoretical contribution.

For simplicity and ease of presentation, this section focuses on results for the case where $C = 2$ corresponds to a two-class classification problem and $M = 1$. Nevertheless, our theoretical results can be easily generalized to the multiclass cases. In what follows, we build theoretical guarantees under the normalized adjacency matrix $\mathbf{A}\mathbf{X}/\sqrt{np}$, where $p$ is the network connectivity probability, and $np$ is the expected degree. To ease presentations, we let $\mathbf{Z} = (\mathbf{A}\mathbf{X})/\sqrt{np}$ (replace $p$ by $\hat{p}$ when uknown). We would like to highlight that rows of $\mathbf{Z}$ are *not* independent. Let $\|\boldsymbol{\beta}^{(0)}\|_0 = s$ for some $s \ll d$. The estimate $\hat{\boldsymbol{\beta}}$ is obtained by minimizing the loss function in Eq. 2.2, which can further rewritten using $\mathbf{Z}$, as shown below,

$$\hat{\boldsymbol{\beta}} = \arg\min_{\boldsymbol{\beta}} \left\{ -\sum_{i=1}^n \left[ \mathbf{Y}_i \mathbf{Z}_i^\top \boldsymbol{\beta} - \log\left(1 + \exp(\mathbf{Z}_i^\top \boldsymbol{\beta})\right) \right] + \lambda \|\boldsymbol{\beta}\|_1 \right\}, \tag{4.1}$$

where $\mathbf{Z}_i$ is the $i$th row of $\mathbf{Z}$. The key difference between our analysis of $\hat{\boldsymbol{\beta}}$ and that of a standard high dimensional generalized linear model (e.g., Van de Geer et al. (2014)) is precisely the dependence among observations. Below, we present the assumptions required to establish the theoretical guarantees of our penalized estimator $\hat{\boldsymbol{\beta}}$:

**Assumption 4.1.** *We assume $\mathbf{X}_i$'s are generated independently from a sub-gaussian distribution with parameter $\sigma_{\mathbf{X}}$, i.e., $\mathbb{E}[\exp(\lambda a^\top \mathbf{X}_i)] \leq \exp(\lambda^2 \sigma_{\mathbf{X}}^2/2)$ for all $\lambda \in \mathbb{R}$ and for all $a \in S^{p-1}$. Let $\Sigma_{\mathbf{X}}$ be the covariance matrix of $\mathbf{X}_i$. Assume that $\lambda_{\min}(\Sigma_{\mathbf{X}}) \geq \kappa_l > 0$ and $\max_j \Sigma_{\mathbf{X},jj} \leq \sigma_+$ for some constant $(\kappa_l, \sigma_+)$.*

**Assumption 4.2** (Network connectivity). *The network connectivity parameter $p$ satisfies $(np)/\log n \to \infty$, and $p \log d \to 0$ as $n \to \infty$.*

**Assumption 4.3** (Sparsity). *The sparsity parameter $s$ of $\boldsymbol{\beta}^{(0)}$ satisfies: $s \frac{\log^2 d}{n} \frac{(1-2p)}{4p \log((1-p)/p)} = o(1)$.*

All three assumptions above are quite standard in the literature on the analysis of high-dimensional linear and generalized linear models. Assumption 4.1 requires the covariates to be sub-gaussian, a ubiquitous assumption, which is often needed to deal with ultra-dimensional setup (i.e., $d$ can be as large as $e^{n^\gamma}$ for some $\gamma < 1$) (see Chapter 6 of Bühlmann & Van De Geer (2011) for details). One may relax this assumption as the cost of the trade-off between $d$ and $n$; the thicker the tail of $\mathbf{X}$, the more stringent condition is needed on $d$ for estimation, or one may use robust loss function (Goldsmith (2015)). Assumption 4.2 is also a standard assumption in the literature of network estimation, which merely assumes $p$ to be (slightly) larger than $n^{-1}$ in order (Lei & Rinaldo, 2015). As for Assumption 4.3, in standard i.i.d. setup we require $s \log d/n \to 0$. However, here, our condition is slightly different; we have an additional factor involving a factor of $p$. Whether this dependence is optimal is out of the scope of this paper. Given these assumptions, we are now ready to state our main theorem:

**Theorem 4.4.** *Under Assumption 4.1-4.3, the $\ell_1$-penalized estimator $\hat{\boldsymbol{\beta}}$ obtained in equation 4.1 satisfies;*

$$\|\hat{\boldsymbol{\beta}} - \boldsymbol{\beta}^{(0)}\|_2^2 \leq c\frac{s \log d}{n}$$

*for some constant $c > 0$ with probability $1 - g(n, p, d)$, where $g(n, p, d)$ goes to 0 as $n \to \infty$, mentioned explicitly in the proof.*

The proof of the theorem is deferred to the Appendix. One of the main technical challenges lies in establishing a condition equivalent to the standard restricted strong convexity (RSC) or restricted eigenvalue (RE) condition in the presence of network dependency. It requires that the sample covariance matrix has a minimum eigenvalue bounded away from zero in certain directions (the global minimum eigenvalue is always zero since the sample covariance matrix is low-rank due to the dimensionality exceeding the sample size). This task is difficult even in an i.i.d. setup, as demonstrated by the collaborative efforts of mathematicians and statisticians over the past decade Raskutti et al. (2010); Zhou (2009); Rudelson & Zhou (2012); Negahban et al. (2009) for related references. The modified RSC condition under network dependency is presented in Lemma C.1, which is of independent interest.

**Remark 4.5.** *It may seem surprising that the convergence rate of $\hat{\boldsymbol{\beta}}$, as established in Theorem 4.4 does not depend on $p$. This is precisely because we have appropriately scaled $\mathbf{Z} = (\mathbf{AX})/\sqrt{np}$ by $p$. If we change the scale, i.e., say we use symmetric normalized Laplacian $\mathbf{D}^{-1/2}\mathbf{AD}^{-1/2}$, then the effective calling would be $\mathbf{Z} = (\mathbf{AX})/np$. In that case, the estimation rate of $\hat{\boldsymbol{\beta}}$ would depend on $p$, and this dependence can be easily quantified by carefully tracking the steps of our proof. However, as this involves technical algebra without adding further insight, we refrain from pursuing it here.*

**Remark 4.6.** *Our theorem precisely quantifies the estimation rate of $\boldsymbol{\beta}^{(0)}$ in a single domain, which can be viewed as the estimation rate using only data from the target domain. Recently, estimating $\boldsymbol{\beta}^{(0)}$ using related source samples under the high-dimensional generalized linear model setup has been explored by Tian & Feng (2023); Li et al. (2023). As previously mentioned, extending these ideas to incorporate network dependency is quite challenging. However, we have conjectured an estimation rate for $\boldsymbol{\beta}^{(0)}$ obtained using Trans-GCR. See Appendix E for details.*

**Remark 4.7.** *Although in our theory we have assumed all the entries of $\mathbf{A}$ are generated independently from $\mathsf{Ber}(p)$, Theorem 4.4 continues to hold under the self-loop (resp. no self-loop), i.e. if we set $\mathbf{A}_{ii} = 1$ (resp. $\mathbf{A}_{ii} = 0$) and generate off-diagonal elements from $\mathsf{Ber}(p)$, with different constant $c'$ instead of $c$. In the proof, we have pointed out the required modifications precisely to adapt proof when i) we have self-loop, and ii) $p$ is unknown and estimated from the data.*

# 5 SIMULATION STUDIES

**Simulation setup.** For the $k$th dataset, we generate simulation data in the following steps. We first generate node features $\mathbf{X}^{(k)} \in \mathbb{R}^{n_k \times d}$ from i.i.d. Gaussian with mean zero and identity covariance matrix. We then generate adjacency matrix $\mathbf{A}^{(k)} \in \{0,1\}^{n_k \times n_k}$ considering three different models: ER random graph model (Erdős & Rényi, 1959), stochastic block model (Holland et al., 1983), and graphon model (Lovász & Szegedy, 2006). We then generate a $C$-classes response $\mathbf{Y}^{(k)} \in \{0,1\}^{n_k \times C}$ using Eq. 2.2 with $M = 1$.

In this simulation, we consider a three-classes outcome, i.e., $C = 3$. For the target data, we set the target sample size $n_0 = 200$. We consider high-dimensional sparse node covariates, where the number of covariates is $d = 500$, and the number of non-zero covariates in $\boldsymbol{\beta}_c^{(0)}$ is $s = 50$. We set non-zero model coefficients for the first $s$ covariates, i.e., $\boldsymbol{\beta}_1^{(0)} = (0.4 \cdot \mathbf{1}_s, \mathbf{0}_{d-s})$, $\boldsymbol{\beta}_2^{(0)} = (0.5 \cdot \mathbf{1}_s, \mathbf{0}_{d-s})$, where $\mathbf{1}_s$ has all $s$ elements 1 and $\mathbf{0}_{d-s}$ has all $d - s$ elements 0. For the $k$th source data, we set $\boldsymbol{\beta}_1^{(k)} = \boldsymbol{\beta}_1^{(0)} - (h_k \cdot \mathbf{1}_s, \mathbf{0}_{d-s})$, $\boldsymbol{\beta}_2^{(k)} = \boldsymbol{\beta}_2^{(0)} + (h_k \cdot \mathbf{1}_s, \mathbf{0}_{d-s})$, $k = 1, \ldots, K$. For the first five sources, we set $h_k = h$ where $h$ is a small value and consider varying $h$ to study its effect. For the subsequent sources, $h_k$ is set to be a large value, i.e., 10. We regard the first five sources as transferable datasets, that is, $\mathcal{A} = \{1, \ldots, 5\}$, and view the remaining sources as non-transferable datasets. We set the source sample sizes as the same value, i.e., $n_1 = \ldots = n_K = n$.

**Evaluation metric and baselines.** We evaluate the performance by calculating the mean squared estimation errors (MSE) between the estimated coefficients and the true target coefficient, i.e., $\text{MSE} = \frac{1}{2d}(||\hat{\boldsymbol{\beta}}_1^{(0)} - \boldsymbol{\beta}_1^{(0)}||_F^2 + ||\hat{\boldsymbol{\beta}}_2^{(0)} - \boldsymbol{\beta}_2^{(0)}||_F^2)$. All experiments are replicated 100 times to calculate the averaged MSE. We compare our method Trans-GCR with two baselines: (1) GCR which uses target data only, and (2) Naive transfer learning (Naive TL), which pools the source and target data together in a brute-force way and trains a single model on the combined dataset to obtain an estimation. Specifically, the adjacency matrices are merged into a block-diagonal matrix, while the node features and outcome vectors are concatenated row-wise.

## 5.1 SIMULATION RESULTS WHEN THE TRANSFERABLE SOURCE SET IS KNOWN

We first show results when $\mathcal{A} = \{1, \ldots, 5\}$ is known, i.e., we will only use the first five source data to perform transfer learning. Due to page limit, we only show the results when network is generated under ER model in the main text, but additional simulation results under SBM and graphon model can be found in Figures S1 and S2 in the Appendix. In addition, we also show our method's superior performance with two convolution layers in Figure S3.

**Asymptotic performance.** To investigate the asymptotic performance of the Trans-GCR method, we vary the sample size of source data $n \in \{100, \ldots, 1000\}$ while fixing source-target domain shift $h = 1$, fixing the ER edge probability in the target data and source data $p_0 = \ldots = p_5 = 0.05$. Figure 2(a) shows that our method Trans-GCR demonstrates a marked decrease in MSE as the source data sample size increases. Naive TL also shows a decreasing MSE trend but at a significantly higher error rate compared with Trans-GCR. In contrast, the GCR method has the highest MSE across all source sample sizes and remains unchanged since it does not utilize source data information.

**Effect of source-target domain shift.** To investigate the impact of $h$, we vary $h \in \{1, \ldots, 10\}$, while fixing source sample size $n = 600$, and fixing the ER edge probability $p_0 = \ldots = p_5 = 0.05$. Figure 2(b) reveals that the MSE of Trans-GCR and Naive TL increases gradually as the source-target domain shift grows, which is expected since a larger shift implies reduced transferability between source and target domains. When $h$ increases to 10, the advantage of transfer learning disappears.

**Effect of source-target network density discrepancy.** To investigate the impact of the network density difference between the source and target networks, we vary the ER edge probability in the source data $p_1 = \ldots = p_5 \in \{0.01, \ldots, 0.1\}$ while fixing the ER edge probability in the target data as 0.05, and fixing $n = 600$, $h = 1$. Figure 2(c) reveals that both Trans-GCR and Naive TL show a slight U-shape trend, with the best results achieved when the source and target network densities are similar. As the density difference increases in either direction, the performance degrades. While network density discrepancies can impact the performance of transfer learning approaches, Trans-GCR demonstrates strong robustness in handling these differences, consistently outperforming GCR and Naive TL across the range of density variations tested.

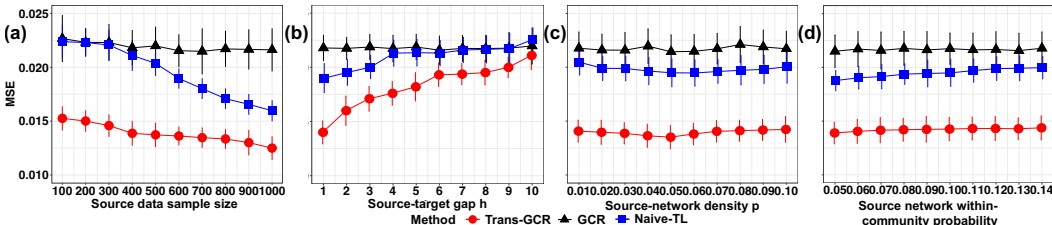

Figure 2: Performance comparison (MSE) of Trans-GCR (red), GCR (black), Naive TL (blue) across varying (a) Source sample size, (b) Source-target gap $h$, (c) Source network density (0.05 means identical densities) (d) Source network within-community probability (higher value means more discrepancy).

**Effect of source-target network distribution discrepancy.** To investigate the impact of distribution discrepancy between source and target networks, we consider a scenario where the target network is generated from the ER model with parameter 0.05 while the source networks are generated from a balanced two-block SBM. In the SBM, we fix the between-community connection probability as 0.05 and the vary within-community probability from 0.05 to 0.14. When the within-community probability is 0.05, the SBM becomes identical to the ER(0.05) model used for the target network. We fix the source sample size $n = 600$, and source-target domain shift $h = 1$. Figure 2(d) shows that the MSE of the Trans-GCR method slightly increases as the distribution discrepancy increases. In contrast, the MSE of Naive TL shows a clear upward trend.

## 5.2 TRANSFERABLE SOURCE DETECTION RESULTS

Given $K$ source domains, we apply Algorithm 2 to obtain the transferability score for each source domain. Here, we use a three-fold cross-validation. To assess the effectiveness of these scores in identifying transferable sources, we treat the task as a binary classification problem and compute the AUC (area under the ROC curve). A higher AUC indicates better performance in distinguishing transferable and non-transferable sources based on the transferability scores. Following the aforementioned setting, the first five source domains are defined as transferable. We vary the number of total candidate source domains $K$ from 6 to 10, while fixing ER edge probability $p_0 = \ldots = p_K = 0.05$, and source data sample size $n = 600$. Figure 3 shows that as $K$ increases, indicating a more challenging detection task, the AUC slightly decreases. Despite this, the AUC consistently

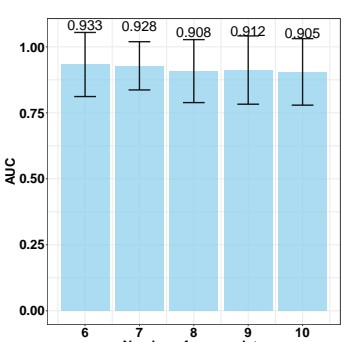

Figure 3: AUC for detecting transferable sources.

remains above 0.9, showcasing the method's robustness and accuracy in identifying transferable source domains.

## 6 REAL DATA EXPERIMENTS

**Data description.** We conduct experiments on three widely-used real-world citation networks (Tang et al., 2008; Shen et al., 2020): (1) DBLPv7 (abbreviated as D), containing 5484 nodes and 8130 edges, extracted from the DBLP database; (2) Citationv1 (abbreviated as C), containing 8935 nodes and 15113 edges, obtained from the Microsoft Academic Graph database; (3) ACMv9 (abbreviated as A), containing 9360 nodes and 15602 edges, derived from the ACM digital library. In these networks, each node represents a paper, and the adjacency matrix $\mathbf{A}$ encodes the citation relationships between papers. The bag-of-words attribute vectors $\mathbf{X}$ are derived from keywords extracted from paper titles, with a combined vocabulary of 6775 unique attributes across all networks. Thus, the feature dimension of $\mathbf{X}$ is 6775. Each paper is associated with a label belonging to one of these five classes: Databases, Artificial Intelligence, Computer Vision, Information Security, and Networking.

**Transfer learning tasks.** To evaluate the effectiveness of our method Trans-GCR, we conduct nine transfer learning tasks between different source and target: (1) C → D, (2) A → D, (3) C & A → D, (4) D → C, (5) A → C, (6) D & A → C, (7) D → A, (8) C → A, and (9) D & C → A.

**Baselines.** We compare our proposed method with several baselines: GCN-based domain adaption methods, including (1) AdaGCN (Dai et al., 2022), (2) UDAGCN (Wu et al., 2020), and (3) pre-training GNNs (Hu et al., 2020); Naive transfer learning methods, as detailed in Section 5. Briefly, this involves pooling the source and target data into a single dataset and training a single model on the combined data. We apply this naive transfer learning strategy across different models, including: (3) GCR model, (4) GPR-GNN (Chien et al., 2020), (5) GRAND (Feng et al., 2020) (6) Node2vec (Grover & Leskovec, 2016), (7) GraphSAGE (Hamilton et al., 2017), (8) GCN (Kipf & Welling, 2017), (9) APPNP (Gasteiger et al., 2018), (10) attri2vec (Zhang et al., 2019), (11) SGC (Wu et al., 2019), and (12) GAT (Velickovic et al., 2017). In addition to the naive transfer learning results of methods in (3)-(12), we also report the performance of selected methods trained solely on the target domain data for comparison. We take a cross-validated grid-search and present in Table 1 the performance of the baselines in the best configuration of hyperparameters.

**Evaluation.** We evaluate the performance of our Trans-GCR method using a standard metric, i.e., micro-F1 (Dai et al., 2022). These metrics assess the model's predictions on the testing subset of the target data. All experiments are replicated 10 times. Our evaluation aims to answer three key questions: (1) How does the performance of our Trans-GCR method compare with other baseline methods, given the fixed training rate? (2) How does the computational time of our Trans-GCR method compare with other baseline methods? (3) How does the training rate of the source and target networks affect the performance of Trans-GCR? Here, the training rate of a network refers to the proportion of nodes whose labels were utilized for training the model.

**Results.** (1) **Performance comparison with fixed training rate.** Table 1 shows the averaged micro-F1 when a 75% source training and a 3% target training rate. Due to space constraints, we only present the results of seven naive transfer learning methods here, with additional results available in Table S3. The performance of these methods when trained solely on target data is provided in Table S4. The results demonstrate that Trans-GCR consistently outperforms the baseline methods across all tasks. AdaGCN generally performs better than naive TL methods, indicating the advantages of domain adaptation. However, despite AdaGCN's good performance, it suffers from high computational costs, which we will discuss later. Notably, Trans-GCR performs better with two source domains (e.g., C & A → D) than with one (e.g., C → D). This suggests that Trans-GCR effectively leverages complementary information from multiple source domains, enhancing transfer learning performance on the target task. (2) **Computational time.** Figure 4 shows that Trans-GCR has significantly lower computational time than AdaGCN, demonstrating its efficiency.

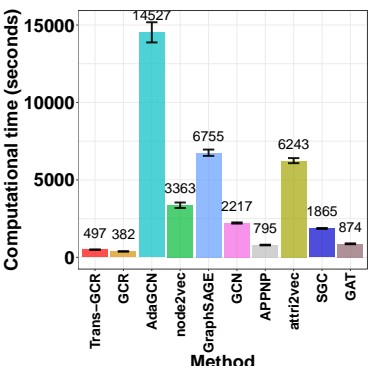

Figure 4: Averaged computational time (seconds) of different methods.

(3) **Effect of training rate.** Figure 5(a) shows the micro-F1 for the transfer learning tasks D → C, as the source training rate increases from 1% to 90%, while the target rate is fixed at 3%. Figure 5(b) depicts the micro-F1 results as the target training rate increases from 1% to 10%, while fixing the source training rate at 75%. As we can see, our proposed method Trans-GCR consistently outperforms AdaGCN across all different source training rates and target training rates, while increasing rates leads to improved performance of our method. Results on other transfer learning tasks, leading to similar conclusions, are shown in Appendix F.2.

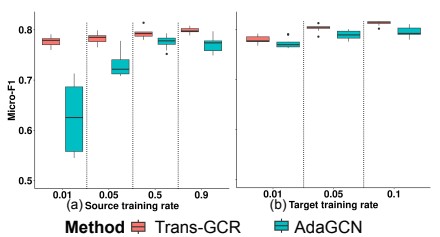

Figure 5: (a) Varying source training rates (b) Varying target training rates.

**(4) Sensitivity analysis to hyperparameters.** When implementing Trans-GCR, we employed a cross-validation procedure to select the optimal $M$ and $\lambda$. We systematically varied $M$ from 2 to 7 and $\lambda$ from 0.005 to 0.0015. Table S5 and S6 in the Appendix shows the results. From Table S5, we have the following observations: (1) For each target-source pair, the performance generally improves with increasing $M$. The performance is optimal when $M = 5$, after which it may stabilize or slightly decrease. (2) Even though there is some variability in the micro-F1 scores for different values of $M$, it is not overly sensitive. For example, the micro-F1 score for the target domain D with source

domain C slightly fluctuates between 75.21% and 76.53%. From Table S6, we observe that: For each target-source pair, the performance varies with different $\lambda$ values. The performance is generally higher at the smallest $\lambda$ value (0.0005). Nevertheless, the changes in micro-F1 scores are relatively modest, indicating that our method is robust to variations in $\lambda$. For example, the micro-F1 score for the target domain C with source domain A varies slightly between 78.56% and 79.61%. In summary, these tables collectively indicate that our proposed Trans-GCR method is not highly sensitive to the specific values of $M$ and $\lambda$.

Table 1: Averaged Micro F1 score (%) of various methods, over 10 replicates, with source training rate fixed at 0.75 and target training rate fixed at 0.03.

| Target | Source | Trans-GCR | GCR | AdaGCN | UDAGCN | Pre-trained GNNs | GPRGNN | GRAND | GCN | APPNP | SGC | GAT |
|--------|--------|-----------|-----|--------|--------|------------------|--------|-------|-----|-------|-----|-----|
|   | C | **76.53** | 72.30 | 75.14 | 69.52 | 73.15 | 74.48 | 67.23 | 71.59 | 73.10 | 72.12 | 71.74 |
| D | A | **75.16** | 69.75 | 74.52 | 58.24 | 70.03 | 72.07 | 66.78 | 69.12 | 70.82 | 68.64 | 67.34 |
|   | C&A | **76.61** | 70.09 | 74.87 | 71.15 | 72.89 | 75.11 | 69.33 | 64.35 | 70.94 | 71.31 | 62.48 |
|   | D | **78.99** | 72.82 | 77.85 | 61.63 | 75.32 | 75.36 | 67.13 | 72.97 | 75.26 | 77.56 | 73.17 |
| C | A | **80.37** | 77.16 | 79.29 | 71.85 | 67.48 | 75.02 | 73.24 | 73.85 | 75.86 | 76.73 | 73.39 |
|   | D&A | **80.58** | 77.23 | 78.91 | 73.35 | 76.21 | 75.72 | 69.67 | 70.53 | 74.81 | 77.31 | 70.52 |
|   | D | **72.61** | 69.54 | 72.35 | 53.35 | 66.78 | 71.67 | 66.63 | 66.87 | 66.56 | 69.26 | 66.67 |
| A | C | **73.56** | 71.17 | 73.32 | 55.52 | 68.94 | 72.95 | 56.74 | 66.10 | 67.33 | 72.52 | 67.79 |
|   | D&C | **73.78** | 71.32 | 73.26 | 65.85 | 69.61 | 72.91 | 67.31 | 63.19 | 64.81 | 70.18 | 63.80 |

## 7 DISCUSSION

In this paper, we introduce the GCR model to capture the relationship between node classification labels ($\mathbf{Y}$), network structure ($\mathbf{A}$), and node covariates ($\mathbf{X}$). We then propose Trans-GCR, a transfer learning method that enhances estimation in the target domain using knowledge from the source domain. Despite its strong empirical performance and theoretical benefits, our method has limitations that need further research. Firstly, our theoretical results are limited to the ER random graph model. While we show empirical success with other models like SBM and graphon, theoretical validation for these models is still needed. Secondly, the GCR model assumes a linear relationship between graph convolutional features and the log odds ratio; extending this to nonlinear models would be valuable. Thirdly, our source detection algorithm only provides a transferability score, and developing a hypothesis testing method for source domain transferability would be useful. Fourthly, our algorithm currently transfers only point estimations of model parameters, and extending it to include confidence interval estimation would provide a measure of uncertainty. Lastly, as $M$ increases, $\mathbf{S}^M$ converges to a matrix of ones, resulting in oversmoothing, where node representations become indistinguishable. Future work could explore adaptive propagation mechanisms, such as APPNP, to mitigate oversmoothing and preserve meaningful differentiation between node representations in deeper architectures.

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

In this Appendix, we show (1) the notation table to briefly summarize important notations in this manuscript, (2) proofs of theoretical guarantees, (3) additional experimental results, and (4) pseudo code of algorithms.

## A  NOTATION TABLE

We present the detailed expressions of notations widely used in the proposed model and algorithm in Table S1.

Table S1: Notations

| Notations | Description |
|---|---|
| $n$ | number of nodes in entire network |
| $\mathbf{A} \in \{0,1\}^{n \times n}$ | adjacency matrix of entire network without self-loops |
| $d$ | number of covariates |
| $\mathbf{X} \in \mathbb{R}^{n \times d}$ | entire covariate matrix |
| $\mathbf{X}_j \in \mathbb{R}^{1 \times d}$ | feature vector of node $j$, i.e., $j$th row in covariate matrix $\mathbf{X}$ |
| $C$ | number of classes |
| $\mathbf{Y} \in \{0,1\}^{n \times C}$ | entire classification label matrix |
| $\tilde{\mathbf{A}}$ | adjacency matrix of entire network with added self-connections |
| $\tilde{\mathbf{D}}$ | degree matrix of $\tilde{\mathbf{A}}$ |
| $\mathbf{S}$ | normalized adjacency matrix of $\tilde{\mathbf{A}}$ |
| $n_0$ | target domain sample size |
| $\mathbf{A}^{(0)} \in \{0,1\}^{n_0 \times n_0}$ | adjacency matrix of target network |
| $\mathbf{X}^{(0)} \in \mathbb{R}^{n_0 \times d}$ | covariate matrix of target domain |
| $\mathbf{Y}^{(0)} \in \{0,1\}^{n_0 \times C}$ | classification label matrix of target domain |
| $\boldsymbol{\beta}^{(0)} \in \mathbb{R}^{d \times (C-1)}$ | true coefficient matrix associated with target domain under GCR |
| $n_k$ | the $k$th source domain sample size |
| $\mathbf{A}^{(k)} \in \{0,1\}^{n_k \times n_k}$ | adjacency matrix of the $k$th source network |
| $\mathbf{X}^{(k)} \in \mathbb{R}^{n_k \times d}$ | covariate matrix of the $k$th source domain |
| $\mathbf{Y}^{(k)} \in \{0,1\}^{n_k \times C}$ | classification label matrix of the $k$th source domain |
| $\boldsymbol{\beta}^{(k)} \in \mathbb{R}^{d \times (C-1)}$ | true coefficient matrix associated with $k$th source domain under GCR |
| $\delta^{(k)}$ | difference between the $k$th source and the target domain's coefficient |
| $\mathcal{A}_h$ | the set of $h-$level transferable source data, abbreviated as $\mathcal{A}$ for brevity |
| $p_0$ | parameter of Erdős–Rényi (ER) random graph model for target domain |
| $p_k$ | parameter of ER random graph model for the $k$th source domain |
| $\mathbf{A}^{\mathcal{A}}$ | pooled adjacency matrix |
| $\mathbf{X}^{\mathcal{A}}$ | pooled covariate matrix |
| $\mathbf{Y}^{\mathcal{A}}$ | pooled label matrix |
| $n^{\mathcal{A}}$ | sample size in the pooled sample |
| $\boldsymbol{\beta}^{\mathcal{A}}$ | true underlying parameter related to the pooled sample $(\mathbf{A}^{\mathcal{A}}, \mathbf{X}^{\mathcal{A}}, \mathbf{Y}^{\mathcal{A}})$ |
| $\hat{\boldsymbol{\beta}}^{(0)}$ | estimate for $\boldsymbol{\beta}^{(0)}$ obtained using our algorithms |

## B  ROADMAP OF THE PROOF

The proof of our method comprises three key components:

1. Obtaining an optimal value of $\lambda$, the penalty parameter, that upper bounds the effect of the noise in the system (Lemma C.2).

2. Establishing restricted strong convexity of the sample covariance matrix (Lemma C.1).

3. Combining Step 1 and Step 2 along with the fact that $\hat{\beta}$ minimizes the penalized loss function to complete the proof via some algebraic manipulation.

Step 3 is similar to the analysis of the analysis of independent data as in Tian & Feng (2023) or Li et al. (2022). The key difference between the existing analysis and our analysis is Step 1 and Step 2. In this subsection, we highlight the differences. In Step 1, we need to provide an upper bound on

$$\|\nabla L_n(\boldsymbol{\beta}_0)\|_\infty = \max_{1 \le j \le d} \left| \frac{1}{n} \sum_{i=1}^n Z_{ij} \left\{ \mathbf{Y}_i - \psi'(\mathbf{Z}_i^\top \boldsymbol{\beta}^{(0)}) \right\} \right|$$

In the case of i.i.d. observations, the above expression is the maximum of $d$ sample averages, so one may use any standard concentration bound for the sample average (e.g., Chernoff bound, Hoeffding's inequality, etc.) along with a union bound would be sufficient. However, as our observations are dependent, we cannot use any standard concentration inequality directly. Hence, careful modifications are necessary, which are presented in the proof of Lemma C.2 in detail.

Step 3 is the most technically demanding part of our proof, where we demonstrate that the Bregman divergence of the loss function, defined as:

$$\delta L_n(u) = L_n(\boldsymbol{\beta}_0 + u) - L_n(\boldsymbol{\beta}_0) - \left\langle u, \nabla L_n(\boldsymbol{\beta}^{(0)}) \right\rangle$$

is lower bounded by $g(u)|u|_2^2$. This condition would be straightforward to establish with $g(u) \equiv C$ for some $C > 0$ if $L_n$ were strongly convex over the entire domain. However, in the high-dimensional setting, strong convexity on the entire domain is not feasible, making this condition challenging to prove, constructing a suitable $g(u)$.

For i.i.d. observations, this result is established in Negahban et al. (2009). However, as in Step 1, the proof becomes significantly more complex when the observations are dependent. To address this, we had to modify the proof technique of Negahban et al. (2009) to account for dependencies among observations. The details of this modification, as well as a comparison with the proof for i.i.d. data in Negahban et al. (2009), are provided in the proof of Lemma C.1.

## C  PROOF OF THEOREM 4.4

For notational simplicity, define $\psi(x) = \log(1 + e^x)$. Recall that our loss function is negative log-likelihood:

$$L_n(\boldsymbol{\beta}) = \frac{1}{n} \sum_i \left\{ -\mathbf{Y}_i \mathbf{Z}_i^\top \boldsymbol{\beta} + \phi(\mathbf{Z}_i^\top \boldsymbol{\beta}) \right\} ,$$

and our estimator is:

$$\hat{\boldsymbol{\beta}} = \arg\min_{\boldsymbol{\beta}} \left[ L_n(\boldsymbol{\beta}) + \lambda \|\boldsymbol{\beta}\|_1 \right] .$$

Furthermore, define the Bregman divergence $\delta L_n(\boldsymbol{\beta})$ as:

$$\delta L_n(\boldsymbol{\beta}) = L_n(\boldsymbol{\beta}) - L_n(\boldsymbol{\beta}^{(0)}) - \langle \boldsymbol{\beta} - \boldsymbol{\beta}^{(0)}, \nabla L_n(\boldsymbol{\beta}^{(0)}) \rangle . \tag{C.1}$$

As $L_n(\cdot)$ is a convex loss function, it is immediate that $\delta L_n(\boldsymbol{\beta}) \ge 0$ for all $\boldsymbol{\beta} \in \mathbb{R}$. Furthermore, define $\hat{v} = \hat{\boldsymbol{\beta}} - \boldsymbol{\beta}^{(0)}$ and the penalized excess risk function $R_n$ as:

$$R_n(v) = L_n(\boldsymbol{\beta}^{(0)} + v) + \lambda \|\boldsymbol{\beta}^{(0)} + v\|_1 - L_n(\boldsymbol{\beta}^{(0)}) - \lambda \|\boldsymbol{\beta}^{(0)}\|_1 .$$

As both the loss and penalty functions are convex, $R_n(\cdot)$ is a convex function. Furthermore, as $\hat{\boldsymbol{\beta}}$ minimizes the penalized loss function, we have:

$$R_n(\hat{v}) \le 0, \text{ or equivalently } L_n(\hat{\boldsymbol{\beta}}) + \lambda \|\hat{\boldsymbol{\beta}}\|_1 \le L_n(\boldsymbol{\beta}^{(0)}) + \lambda \|\boldsymbol{\beta}^{(0)}\|_1 . \tag{C.2}$$

Recall that we need to show $\|\hat{v}\|_2^2 = \|\hat{\boldsymbol{\beta}} - \boldsymbol{\beta}^{(0)}\|_2^2 \le C(s \log d)/n$. We prove this by reductio ad absurdum. As $s = o(\log d/n)$, we know $C(s \log d)/n < 1$. Suppose that the claim of the theorem is not true. Then there exists $t \in (0, 1)$ such that $C(s \log d)/n < \|t\hat{v}\|_2^2 \le 1$. Call $t\hat{v} = \tilde{v}$. By the convexity of $R_n(\cdot)$ we have:

$$R_n(\tilde{v}) = R_n(t\hat{v}) = R_n(t\hat{v} + (1-t)0) \le t \underbrace{R_n(\hat{v})}_{\le 0} + (1-t) \underbrace{R_n(0)}_{=0} \le 0 .$$

The above chain of inequalities provides an immediate upper bound on $R_n(\tilde{v})$. For the lower bound, we use Bregman divergence:

$$
\begin{aligned}
0 \geq R_n(\tilde{v}) &= L_n(\boldsymbol{\beta}^{(0)} + \tilde{v}) + \lambda\|\boldsymbol{\beta}^{(0)} + \tilde{v}\|_1 - L_n(\boldsymbol{\beta}^{(0)}) - \lambda\|\boldsymbol{\beta}^{(0)}\|_1 \\
&= \langle \tilde{v}, \nabla L_n(\boldsymbol{\beta}^{(0)}) \rangle + \delta L_n(\tilde{v}) + \lambda\|\boldsymbol{\beta}^{(0)} + \tilde{v}\|_1 - \lambda\|\boldsymbol{\beta}^{(0)}\|_1 \\
&\geq \delta L_n(\tilde{v}) - \|\nabla L_n(\boldsymbol{\beta}^{(0)})\|_\infty \|\tilde{v}\|_1 + \lambda\|\boldsymbol{\beta}^{(0)} + \tilde{v}\|_1 - \lambda\|\boldsymbol{\beta}^{(0)}\|_1
\end{aligned}
$$

Therefore, we have:

$$
\delta L_n(\tilde{v}) + \lambda\|\boldsymbol{\beta}^{(0)} + \tilde{v}\|_1 \leq \|\nabla L_n(\boldsymbol{\beta}^{(0)})\|_\infty \|\tilde{v}\|_1 + \lambda\|\boldsymbol{\beta}^{(0)}\|_1 .
$$

Now suppose we choose $\lambda$ such that $\|\nabla L_n(\boldsymbol{\beta}^{(0)})\|_\infty \leq \lambda/2$ with high probability. We will specify the choice of $\lambda$ later in the proof. Under this choice we have:

$$
\delta L_n(\tilde{v}) + \lambda\|\boldsymbol{\beta}^{(0)} + \tilde{v}\|_1 \leq \frac{\lambda}{2}\|\tilde{v}\|_1 + \lambda\|\boldsymbol{\beta}^{(0)}\|_1
$$

$$
\implies \delta L_n(\tilde{v}) + \lambda\|\tilde{v}_{S^c}\|_1 \leq \frac{3\lambda}{2}\|\tilde{v}_S\|_1 . \tag{C.3}
$$

As $\delta L_n(\tilde{v}) \geq 0$, we further have $\|\tilde{v}_{S^c}\|_1 \leq 3\|\tilde{v}_S\|_1$, i.e. $\tilde{v} \in \mathcal{C}(s, 3)$, where $\mathcal{C}(s, \alpha)$ is defined as:

$$
\mathcal{C}(s, \alpha) = \{v \in \mathbb{R}^p : \|v_{S^c}\|_1 \leq \alpha\|v_S\|_1, \text{ for some } S \text{ with } |S| = s\} .
$$

We next present a lemma, which establishes a lower bound on $\delta L_n(\cdot)$ with high probability:

**Lemma C.1.** *Under Assumptions 1 and 2, there exists some positive constants $\kappa_l$ and $C_4$ such that,*

$$
\delta L_n(u) \geq L_\psi(T)\|u\|_2^2 \left\{ \kappa_l - \left( C_4 \log d \sqrt{\frac{\Psi(p)}{n}} \right) \frac{\|u\|_1}{\|u\|_2} \right\}, \ \forall \ \|u\|_2 \leq 1 .
$$

*with probability at least $1 - (\exp\left(\frac{1}{2}\log d + 1 - c_4 \log^2 d\, \Psi(p)\right) + 2\exp(1 - (c_1 - 3/2)\log d) + 2\exp\left(\frac{3}{2}\log d + 1 - c_2 n\right))$, where*

$$
\Psi(p) = \frac{1 - 2p}{4p \log\left((1-p)/p\right)} ,
$$

*and $L_\psi(T)$ is a constant, same as in the proof of Proposition 2 of Negahban et al. (2009).*

We defer the proof the lemme to the end to maintain the flow. Consider the event when the upper bound of Lemma C.1 holds. As by definition $\|\tilde{v}\|_2 \leq 1$, applying Lemma C.1 on $\tilde{v}$ yields:

$$
\delta L_n(\tilde{v}) \geq L_\psi(T)\kappa_l \|\tilde{v}\|_2^2 - L_\psi(T) \left( C_4 \log d \sqrt{\frac{\Psi(p)}{n}} \right) \|\tilde{v}\|_1 \|\tilde{v}\|_2 .
$$

Using this lower bound on the Bregman divergence in equation equation C.3, we have:

$$
\begin{aligned}
L_\psi(T)\kappa_l \|\tilde{v}\|_2^2 + \lambda\|\tilde{v}_{S^c}\|_1 &\leq \frac{3\lambda}{2}\|\tilde{v}_S\|_1 + L_\psi(T) \left( C_4 \log d \sqrt{\frac{\Psi(p)}{n}} \right) \|\tilde{v}\|_1 \|\tilde{v}\|_2 \\
&\leq \frac{3\lambda\sqrt{s}}{2}\|\tilde{v}_S\|_2 + L_\psi(T) \left( C_4 \log d \sqrt{\frac{\Psi(p)}{n}} \right) (\|\tilde{v}_S\|_1 + \|\tilde{v}_{S^c}\|_1)\|\tilde{v}\|_2 \\
&\leq \frac{9}{8L_\psi(T)\kappa_l} s\lambda^2 + \frac{L_\psi(T)\kappa_l}{2}\|\tilde{v}_S\|_2 + 4L_\psi(T) \left( C_4 \log d \sqrt{\frac{\Psi(p)}{n}} \right) (\|\tilde{v}_S\|_1)\|\tilde{v}\|_2 .
\end{aligned}
$$

Changing sides we have:

$$
\begin{aligned}
\frac{L_\psi(T)\kappa_l}{2}\|\tilde{v}\|_2^2 + \lambda\|\tilde{v}_{S^c}\|_1 &\leq \frac{9}{8L_\psi(T)\kappa_l} s\lambda^2 + 4L_\psi(T) \left( C_4 \log d \sqrt{\frac{\Psi(p)}{n}} \right) (\|\tilde{v}_S\|_1)\|\tilde{v}\|_2 \\
&\leq \frac{9}{8L_\psi(T)\kappa_l} s\lambda^2 + 4L_\psi(T)\sqrt{s} \left( C_4 \log d \sqrt{\frac{\Psi(p)}{n}} \right) \|\tilde{v}\|_2^2
\end{aligned}
$$

As under Assumption 4.3, we have $s \log d = o_p(\sqrt{n/\Psi(p)})$, eventually it will be smaller than $L_\psi(T)\kappa_l/4$. Therefore, we have:

$$\frac{1}{4}L_\psi(T)\kappa_l\|\tilde{v}\|_2^2 + \lambda\|\tilde{v}_{S^c}\|_1 \leq \frac{9}{8L_\psi(T)\kappa_l}s\lambda^2 \implies \|\tilde{v}\|_2^2 \leq \frac{9}{2L_\psi^2(T)\kappa_l^2}s\lambda^2 . \tag{C.4}$$

If we can show that a proper choice of $\lambda$ is $C_2\sqrt{\log d/n}$, then we are done as in that we have:

$$\|\tilde{v}\|_2^2 \leq \frac{9C_2}{2L_\psi^2(T)\kappa_l^2}\frac{s\log d}{n} \triangleq C\frac{s\log d}{n}$$

which contradicts the assumption that $\|\tilde{v}\|_2 > C\sqrt{s\log d/n}$. This will complete the proof. In the following lemma, we show that, indeed $C_2\sqrt{\log d/n}$ is a valid choice for $\lambda$:

**Lemma C.2.** *Under Assumption 4.1, there are universal positive constants $(c_1, c_2, c_3)$ such that*

$$\|\nabla L_n(\boldsymbol{\beta}^{(0)})\|_\infty = \left\|\frac{1}{n}\sum_{i=1}^n \mathbf{Z}_i\left\{\mathbf{Y}_i - \psi'(\mathbf{Z}_i^\top\boldsymbol{\beta}^{(0)})\right\}\right\|_\infty \leq C\sqrt{\frac{\log d}{n}} ,$$

*with probability $1 - c_1\left(d^{-c_2} + n^{-1} + e^{\log n - np/c_3}\right)$.*

Therefore, under the above lemma, we replace $\lambda$ by $C\sqrt{\log d/n}$ in equation equation C.4 we complete the proof. Proof of Lemma C.1 and Lemma C.2 can be found in Appendix D

# D PROOF OF ADDITIONAL LEMMAS

## D.1 PROOF OF LEMMA C.1

The proof of Lemma C.1 follows the basic structure of the proof of Proposition 2 of Negahban et al. (2009). However, suitable modifications are necessary to incorporate network dependency. We first state Proposition 2 of Negahban et al. (2009) here for the convenience of the readers:

**Proposition D.1** (Proposition 2 of Negahban et al. (2009))**.** *Consider the logistic loss (negative log-likelihood) function:*

$$L_n(\boldsymbol{\beta}) = \frac{1}{n}\sum_i\left\{-\mathbf{Y}_i\mathbf{Z}_i^\top\boldsymbol{\beta} + \log\left(1 + e^{\mathbf{Z}_i^\top\boldsymbol{\beta}}\right)\right\}$$

*Define the Bregman divergence of $L_n$ as follows:*

$$\delta L_n(u) = L_n(\boldsymbol{\beta}^* + u) - L_n(\boldsymbol{\beta}^*) - \langle u, \nabla L_n(\boldsymbol{\beta}^*)\rangle = \frac{1}{n}\sum_i\psi''(\mathbf{Z}_i^\top\boldsymbol{\beta}^* + v\mathbf{Z}_i^\top u)(\mathbf{Z}_i^\top u)^2 .$$

*where $\psi(x) = \log\left(1 + e^x\right)$. Then we have:*

$$\delta L_n(u) \geq \kappa_1\|u\|_2\left\{\|u\|_2 - \kappa_2\sqrt{\frac{\log d}{n}}\|u\|_1\right\} \quad \forall \quad \|u\|_2 \leq 1$$

*with probability at least $1 - c_1 e^{-c_2 n}$ for some constant $\kappa_1, \kappa_2$.*

However, in their proof, the authors heavily used that $\mathbf{Z}_i$'s are i.i.d., which is not longer true in our situation, as $\mathbf{Z}_i$'s the rows of $\mathbf{AX}/\sqrt{np}$. Below, we highlight the modification of the steps that are needed for the proof:

**Modification 1:** We first need to show that there exists some constant $\kappa_l$ (not depending on $(n, p, d)$) such that

$$v^\top\mathbb{E}\left[\frac{1}{n}\sum_i\mathbf{Z}_i^\top\mathbf{Z}_i\right]v = v^\top\left(\frac{1}{n^2 p}\mathbb{E}[\mathbf{X}^\top\mathbf{A}^\top\mathbf{AX}]\right)v \geq \kappa_l\|v\|_2^2$$

for all $v \in \mathbb{R}^d$. It is easy to see that $\mathbb{E}[\mathbf{X}^\top\mathbf{A}^\top\mathbf{AX}]/(n^2 p) = \Sigma_X$. Therefore, as long as we assume that $\lambda_{\min}(\Sigma_X) \geq \kappa_l$, we are done. The lower bound on the minimum eigenvalue of $\Sigma_X$ follows from Assumption 4.1.

**Remark D.2.** *Suppose we do not know $p$. Then, we can estimate it from separate data (by data splitting) independent of current data. An application of Hoeffding's inequality yields:*

$$\mathbb{P}\left(|\hat{p} - p| \geq t/n\right) \leq Ce^{-ct^2}.$$

*Taking $t = np$ and $t = np/2$ (which goes to infinity as per Assumption 4.2), we have:*

$$\mathbb{P}(p/2 \leq \hat{p} \leq 2p) \geq 1 - C_1 e^{-cn^2 p^2} \uparrow 1.$$

*Therefore, we can perform the entire analysis conditioning on this event.*

**Remark D.3.** *As per Remark D.2, if we condition on the event $\hat{p} \leq 2p$, then we have on that event:*

$$v^\top \mathbb{E}\left[\left(\frac{1}{n^2 \hat{p}} \mathbb{E}[\mathbf{X}^\top \mathbf{A}^\top \mathbf{A} \mathbf{X}]\right)\right] v \geq \frac{\kappa_l}{2} \|v\|_2^2.$$

*Furthermore, if we add self-loop, it is easy to see that*

$$\mathbb{E}[\mathbf{X}^\top \mathbf{A}^\top \mathbf{A} \mathbf{X}] = \left(1 - \frac{1-p}{np}\right) \Sigma_X$$

*As the constant goes to $1$ as $n \uparrow \infty$, we have:*

$$v^\top \mathbb{E}\left[\left(\frac{1}{n^2 \hat{p}} \mathbb{E}[\mathbf{X}^\top \mathbf{A}^\top \mathbf{A} \mathbf{X}]\right)\right] v \geq \frac{\kappa_l}{4} \|v\|^2 \ \forall \ \text{large } n.$$

**Modification 2:** To conclude equation (72) of Negahban et al. (2009), we need i) an upper bound on $\mathbb{E}[(u^\top \mathbf{Z}_i)^4]$ and ii) a tail bound $\mathbb{P}(|u^\top \mathbf{Z}_i| \geq t)$. Getting the tail bound is easy, as we can apply the Cauchy-Schwarz inequality:

$$\mathbb{P}(|u^\top \mathbf{Z}_i| \geq t) \leq \frac{\mathbb{E}[(u^\top \mathbf{Z}_i)^2]}{t^2} \leq \frac{\lambda_{\max}(\Sigma_X)}{t^2}.$$

Now we need to bound the fourth moment (in fact, any $2 + \delta$ moment is sufficient):

$$\mathbb{E}[(u^\top \mathbf{Z}_i)^4]$$

$$= \mathbb{E}\left[\left(\frac{1}{\sqrt{np}} \sum_{j=1}^{n} a_{ij}(\mathbf{X}_j^\top u)\right)^4\right]$$

$$= \mathbb{E}\left[\left(\frac{1}{\sqrt{np}} \sum_{j=1}^{n} (a_{ij} - p + p)(\mathbf{X}_j^\top u)\right)^4\right]$$

$$\leq 8\left\{\mathbb{E}\left[\left(\frac{1}{\sqrt{np}} \sum_{j=1}^{n} (a_{ij} - p)(\mathbf{X}_j^\top u)\right)^4\right] + \mathbb{E}\left[\left(\frac{\sqrt{p}}{\sqrt{n}} \sum_{j=1}^{n} (\mathbf{X}_j^\top u)\right)^4\right]\right\}$$

$$= 8\left\{\frac{1}{n^2} \sum_{j} \mathbb{E}\left[\left(\frac{a_{ij} - p}{\sqrt{p}}\right)^4 (\mathbf{X}_j^\top u)^4\right] + \frac{1}{n^2} \sum_{j \neq j'} \mathbb{E}\left[\left(\frac{a_{ij} - p}{\sqrt{p}}\right)^2 (\mathbf{X}_j^\top u)^2\right] \mathbb{E}\left[\left(\frac{a_{ij} - p}{\sqrt{p}}\right)^2 (\mathbf{X}_j^\top u)^2\right]\right.$$

$$\left. + p^2 \frac{1}{n^2} \sum_{j} \mathbb{E}[(\mathbf{X}_j^\top u)^4] + p^2 \frac{1}{n^2} \sum_{j \neq j'} \mathbb{E}[(\mathbf{X}_j^\top u)^2]\mathbb{E}[(\mathbf{X}_{j'}^\top u)^2]\right\}$$

$$\leq \left\{\frac{1}{np} \mathbb{E}[(\mathbf{X}^\top u)^4] + \left(\mathbb{E}[(\mathbf{X}^\top u)^2]\right)^2 + \frac{p^2}{n} \mathbb{E}[(\mathbf{X}^\top u)^4] + p^2 \left(\mathbb{E}[(\mathbf{X}^\top u)^2]\right)^2\right\}$$

which is finite as $\mathbb{E}[(\mathbf{X}^\top u)^4]$ is finite ($\mathbf{X}$ is sub-gaussian) and $np \uparrow \infty$.

**Remark D.4.** *If $p$ is unknown, then we can modify the first step by conditioning on the event $\hat{p} \geq p/2$ and have:*

$$\mathbb{E}\left[\left(\frac{1}{\sqrt{n\hat{p}}} \sum_{j=1}^{n} a_{ij}(\mathbf{X}_j^\top u)\right)^4\right] \leq 4\mathbb{E}\left[\left(\frac{1}{\sqrt{np}} \sum_{j=1}^{n} a_{ij}(\mathbf{X}_j^\top u)\right)^4\right]$$

*The rest of the proof will remain the same. Furthermore, if we add self-loop, then we have to single the term $a_{ii}(\mathbf{X}_i^\top u)^2$ out as now $a_{ii} = 1$. However, it will add another term of other $1/(n^2 p^2)$ to the above bound, which is asymptotically negligible.*

**Modification 3:** We next show an analog of equation (76) of Negahban et al. (2009). Following the notations of Negahban et al. (2009), define a random process $Z(t)$ as:

$$Z(t) = \sup_{u \in \mathbb{S}_2(1) \cap \mathbb{B}_1(t)} \left| \frac{1}{n} \sum_{i=1}^{n} \{g_u(\mathbf{Z}_i) - \mathbb{E}[g_u(\mathbf{Z}_i)]\} \right| \triangleq F_t(Z_1, \ldots, Z_n) \,.$$

We will apply bounded difference inequality (Theorem 6.2 of Boucheron et al. (2003)). Note that conditional of $\mathbf{X}$, $\mathbf{Z}_i$'s are independent random vectors. Furthermore, for any $1 \leq i \leq n$ and for any $Z_i' \neq \mathbf{Z}_i$:

$$F_t(Z_1, \ldots, Z_{i-1}, \mathbf{Z}_i, \ldots, Z_n) - F_t(Z_1, \ldots, Z_{i-1}, Z_i', \ldots, Z_n)$$

$$\leq \frac{1}{n} \sup_{u \in \mathbb{S}_2(1) \cap \mathbb{S}_1(t)} |g_u(Z_i') - \mathbb{E}[g_u(Z_i')]| \leq \frac{\tau^2}{2n} \quad [\because g_u(\cdot) \leq \tau^2/4].$$

Therefore, by bounded difference inequality:

$$\mathbb{P}\left( Z(t) \geq \mathbb{E}[Z(t) \mid \mathbf{X}] + t \mid \mathbf{X} \right) \leq \exp\left( -\frac{8nt^2}{\tau^4} \right)$$

As the right-hand side does not depend on the value of $\mathbf{X}$, we can further conclude the following by taking expectations with respect to $\mathbf{X}$ on both sides:

$$\mathbb{P}\left( Z(t) \geq \mathbb{E}[Z(t) \mid \mathbf{X}] + t \right) \leq \exp\left( -\frac{8nt^2}{\tau^4} \right). \tag{D.1}$$

Next, using symmetrization and Rademacher complexity bounds, we bound $\mathbb{E}[Z(t) \mid \mathbf{X}]$. For notational simplicity let us define:

$$V_n = \max_{1 \leq j \leq p} \left| \frac{1}{\sqrt{n}} \sum_{k=1}^{n} \mathbf{X}_{kj} \right|$$

$$\Gamma_n = \max_{1 \leq j \leq d} \frac{1}{n} \sum_{k=1}^{n} \mathbf{X}_{kj}^2 \,.$$

Now, as we have already pointed out, conditional on $\mathbf{X}$, $\mathbf{Z}_i$'s are i.i.d. random vectors. Therefore, the symmetrization argument holds, and following the same line of argument as of Negahban et al. (2009), we can conclude an analog of their equation (78):

$$\mathbb{E}[Z(t) \mid \mathbf{X}] \leq 8K_3 t \mathbb{E}_{\epsilon, Z} \left[ \max_{1 \leq j \leq d} \left| \frac{1}{n} \sum_{i=1}^{n} \epsilon_i \mathbf{Z}_{ij} \mathbb{1}_{|\mathbf{Z}_i^\top \boldsymbol{\beta}_*| \leq T} \right| \mid \mathbf{X} \right]$$

$$= \frac{8K_3 t}{\sqrt{n}} \mathbb{E}_{\mathbf{Z}|\mathbf{X}} \left[ \mathbb{E}_{\epsilon|\mathbf{Z},\mathbf{X}} \left[ \max_{1 \leq j \leq d} \left| \frac{1}{\sqrt{n}} \sum_{i=1}^{n} \epsilon_i \mathbf{Z}_{ij} \mathbb{1}_{|\mathbf{Z}_i^\top \boldsymbol{\beta}_*| \leq T} \right| \right] \right]$$

First, observe that $\{\epsilon_1, \ldots, \epsilon_n\}$ are Rademacher random variables (which are also subgaussian with sub-gaussian constant being 1), and therefore, conditionally on $\mathbf{Z}$,

$$\frac{1}{\sqrt{n}} \sum_{i=1}^{n} \epsilon_i \mathbf{Z}_{ij} \mathbb{1}_{|\mathbf{Z}_i^\top \boldsymbol{\beta}_*| \leq T} \text{ is subgaussian with norm } \sqrt{\frac{1}{n} \sum_{i=1}^{n} \mathbf{Z}_{ij}^2 \mathbb{1}_{|\mathbf{Z}_i^\top \boldsymbol{\beta}_*| \leq T}} \leq \sqrt{\frac{1}{n} \sum_{i=1}^{n} \mathbf{Z}_{ij}^2} \,.$$

Therefore, from standard probability tail bound calculation, we have:

$$\mathbb{E}_{\epsilon|\mathbf{Z},\mathbf{X}} \left[ \max_{1 \leq j \leq d} \left| \frac{1}{\sqrt{n}} \sum_{i=1}^{n} \epsilon_i \mathbf{Z}_{ij} \mathbb{1}_{|\mathbf{Z}_i^\top \boldsymbol{\beta}_*| \leq T} \right| \right] \leq \sqrt{2 \log d} \max_{1 \leq j \leq d} \sqrt{\frac{1}{n} \sum_{i=1}^{n} \mathbf{Z}_{ij}^2} \,.$$

Therefore, we have:

$$\mathbb{E}[Z(t) \mid \mathbf{X}] \le 8\sqrt{2}K_3 t \sqrt{\frac{\log d}{n}} \mathbb{E}\left[\max_{1\le j\le d} \sqrt{\frac{1}{n}\sum_{i=1}^{n}\mathbf{Z}_{ij}^2} \mid \mathbf{X}\right] \tag{D.2}$$

Recall that by define $\mathbf{Z}_{ij} = \left(\sum_k A_{ik}\mathbf{X}_{kj}\right)/\sqrt{np}$, which is not centered conditional on $\mathbf{X}$. Therefore we first center it:

$$Z_{ij} = \frac{1}{\sqrt{np}}\sum_k A_{ik}\mathbf{X}_{kj} = \frac{1}{\sqrt{np}}\sum_k (A_{ik}-p)\mathbf{X}_{kj} + \sqrt{\frac{p}{n}}\sum_k \mathbf{X}_{kj} \triangleq \bar{\mathbf{Z}}_{ij} + \sqrt{\frac{p}{n}}\sum_k \mathbf{X}_{kj}.$$

Using this we have:

$$\mathbb{E}\left[\max_{1\le j\le d}\sqrt{\frac{1}{n}\sum_{i=1}^{n}\mathbf{Z}_{ij}^2} \mid \mathbf{X}\right] \le \mathbb{E}\left[\max_{1\le j\le d}\sqrt{\frac{1}{n}\sum_{i=1}^{n}\bar{\mathbf{Z}}_{ij}^2} \mid \mathbf{X}\right] + \sqrt{p}\max_{1\le j\le d}\left|\frac{1}{\sqrt{n}}\sum_k \mathbf{X}_{kj}\right|$$

$$= \mathbb{E}\left[\max_{1\le j\le d}\sqrt{\frac{1}{n}\sum_{i=1}^{n}\bar{\mathbf{Z}}_{ij}^2} \mid \mathbf{X}\right] + \sqrt{p}V_n$$

$$\le \sqrt{\mathbb{E}\left[\max_{1\le j\le d}\frac{1}{n}\sum_{i=1}^{n}\bar{\mathbf{Z}}_{ij}^2 \mid \mathbf{X}\right]} + \sqrt{p}V_n. \tag{D.3}$$

**Remark D.5.** *If we have $\hat{p}$, then again, at the bound in equation equation D.3, we have an additional factor of $\sqrt{2}$ for replacing $\hat{p}$ by $p$.*

We next establish an upper bound on the conditional expectation of the maximum of the mean of $\bar{\mathbf{Z}}_{ij}^2$. We first claim that $\bar{\mathbf{Z}}_{ij}$ is a SG($\sigma_j$) random variable with the value of $\sigma_j$ defined in equation equation D.4 below. To see this, first note that, from Theorem 2.1 of Ostrovsky & Sirota (2014), we know $(A_{ik}-p)$ is SG($\sqrt{2}Q(p)$). Therefore, we have:

$$\bar{Z}_{ij} = \frac{1}{\sqrt{np}}\sum_k (A_{ik}-p)\mathbf{X}_{kj} \in \text{SG}\left(\sqrt{\frac{2Q^2(p)}{p}\frac{1}{n}\sum_k \mathbf{X}_{kj}^2}\right) \triangleq \text{SG}(\sigma_j). \tag{D.4}$$

**Remark D.6.** *The same sub-gaussian bound continues to hold even under self-loop as $(1-p)$ is sub-gaussian with constant $\le 2Q^2(p)$.*

Let $\mu_j = \mathbb{E}[\bar{\mathbf{Z}}_{ij}^2|\mathbf{X}]$. Then, by equation (37) of Honorio & Jaakkola (2014), we know $\bar{\mathbf{Z}}_{ij}^2 - \mu_j$ is a sub-exponential random variable, in particular:

$$\bar{\mathbf{Z}}_{ij}^2 - \mu_j \in \text{SE}\left(\sqrt{32}\sigma_j, 4\sigma_j^2\right).$$

Hence we have, by equation (2.18) of Wainwright (2019) (we use the version for the two-sided bound here):

$$\mathbb{P}\left(\left|\frac{1}{n}\sum_{i=1}^{n}(\bar{\mathbf{Z}}_{ij}^2 - \mu_j)\right| \ge t\right) \le \exp\left(-\frac{1}{8\sigma_j^2}\min\left\{\frac{nt^2}{8}, nt\right\}\right). \tag{D.5}$$

Going back to equation D.3, we have:

$$\mathbb{E}\left[\max_{1\le j\le d}\frac{1}{n}\sum_{i=1}^{n}\bar{\mathbf{Z}}_{ij}^2 \mid \mathbf{X}\right] = \mathbb{E}\left[\max_{1\le j\le d}\left\{\left(\frac{1}{n}\sum_{i=1}^{n}(\bar{\mathbf{Z}}_{ij}^2 - \mu_j)\right) + \mu_j\right\} \mid \mathbf{X}\right]$$

$$\le \mathbb{E}\left[\max_{1\le j\le d}\left|\frac{1}{n}\sum_{i=1}^{n}(\bar{\mathbf{Z}}_{ij}^2 - \mu_j)\right| \mid \mathbf{X}\right] + \max_{1\le j\le d}\mu_j$$

Now, bound the first term using the concentration inequality equation D.5. Towards that end, define $\sigma_* = \max_j \sigma_j$ and observe that $\sigma_* = \sqrt{2Q^2(p)/p}\sqrt{\Gamma_n}$.

$$\mathbb{E}\left[\max_{1\le j\le d}\left|\frac{1}{n}\sum_{i=1}^{n}(\bar{\mathbf{Z}}_{ij}^2 - \mu_j)\right| \mid \mathbf{X}\right] \le 8\max\{\sigma_*\sqrt{\log d}, \sigma_*^2\log d\}.$$

Furthermore, observe that:

$$\mu_j = \mathbb{E}[\bar{\mathbf{Z}}_{ij}^2 \mid \mathbf{X}] = \mathbb{E}\left[\left(\frac{1}{\sqrt{np}}\sum_k (A_{ik}-p)X_{kj}\right)^2 \mid \mathbf{X}\right] = (1-p)\frac{1}{n}\sum_k X_{kj}^2,$$

which implies, $\max_{1\leq j\leq d}\mu_j = (1-p)\Gamma_n$. Using these bounds in equation equation D.3, we have:

$$\mathbb{E}\left[\max_{1\leq j\leq d}\sqrt{\frac{1}{n}\sum_{i=1}^n \mathbf{Z}_{ij}^2} \mid \mathbf{X}\right] \leq \sqrt{\max\{\sigma_*\sqrt{\log d},\sigma_*^2\log d\} + (1-p)\Gamma_n} + \sqrt{p}V_n \qquad \text{(D.6)}$$

**Remark D.7.** *Here also the constant will change by an additional factor of $\sqrt{2}$, see Remark D.5.*

This, along, with equation equation D.2,yields:

$$\mathbb{E}[Z(t) \mid \mathbf{X}] \leq Ct\sqrt{\frac{\log d}{n}}\left(\sqrt{\max\{\sigma_*\sqrt{\log d},\sigma_*^2\log d\} + (1-p)\Gamma_n} + \sqrt{p}V_n\right)$$

$$\triangleq Ct\sqrt{\frac{\log d}{n}}g(\mathbf{X},p,d). \qquad \text{(D.7)}$$

Using this in the inequality equation D.1 yields:

$$\mathbb{P}\left(Z(t) \geq Ct\sqrt{\frac{\log d}{n}}g(\mathbf{X},p,d) + y\right) \leq \exp\left(-\frac{8ny^2}{\tau^4}\right). \qquad \text{(D.8)}$$

We next provide an upper bound for $g(\mathbf{X},p,d)$ term. Note that in the expression of $g(\mathbf{X},p,d)$, there are two key terms: $\Gamma_n, V_n$. Therefore, if we can obtain an upper bound on them individually, we can obtain an upper bound on $g(\mathbf{X},p,d)$. We start with $V_n$; for any fixed $j$, $\mathbf{X}_{kj}$'s are i.i.d sub-gaussian random variable with constant $\sigma_X^2$. Therefore, we have:

$$\mathbb{P}\left(\left|\frac{1}{\sqrt{n}}\sum_{k=1}^n \mathbf{X}_{kj}\right| \geq t\right) \leq 2e^{-\frac{t^2}{2\sigma_X^2}}$$

As a consequence, by union bound:

$$\mathbb{P}(V_n \geq t) = \mathbb{P}\left(\max_j\left|\frac{1}{\sqrt{n}}\sum_{k=1}^n \mathbf{X}_{kj}\right| \geq t\right) \leq 2e^{\log d - \frac{t^2}{2\sigma_X^2}}$$

Therefore, choosing $t = \sigma_X\sqrt{2c_1\log d}$ (where $c_1 \geq 2$), we have:

$$V_n \leq \sigma_X\sqrt{2c_1\log d}) \quad \text{with probability } \geq 1 - 2\exp(-(c_1-1)\log d). \qquad \text{(D.9)}$$

Call this event $\Omega_{n,\mathbf{X},1}$. Our next target is $\Gamma_n$ which can be further upper bounded by:

$$\Gamma_n = \max_j\frac{1}{n}\sum_{k=1}^n \mathbf{X}_{kj}^2 \leq \max_j\frac{1}{n}\sum_{k=1}^n(\mathbf{X}_{kj}^2 - \Sigma_{X,jj}) + \max_j\Sigma_{X,jj} \triangleq \bar{\Gamma}_n + \max_j\Sigma_{X,jj}.$$

As we have assumed $\max_j\Sigma_{X,jj} \leq C_1$ for some constant $C_1$, we need to bound $\bar{\Gamma}_n$. Here, we also use the fact that $\mathbf{X}_{jk}^2 - \Sigma_{X,jj} \in \text{SE}(\sqrt{32}\sigma_X, 4\sigma_X^2)$. Therefore, by equation (2.18) of Wainwright (2019) we have:

$$\mathbb{P}\left(\left|\frac{1}{n}\sum_{k=1}^n(\mathbf{X}_{kj}^2 - \Sigma_{X,jj})\right| \geq t\right) \leq 2\exp\left(-\frac{n}{8\sigma_X^2}\min\left\{\frac{t^2}{8},t\right\}\right)$$

Therefore, by union bound:

$$\mathbb{P}\left(\max_{1\leq j\leq d}\left|\frac{1}{n}\sum_{k=1}^n(\mathbf{X}_{kj}^2 - \Sigma_{X,jj})\right| \geq t\right) \leq 2\exp\left(\log d - \frac{n}{8\sigma_X^2}\min\left\{\frac{t^2}{8},t\right\}\right)$$

Choosing $t = \max_j \Sigma_{X,jj}$, we have:

$$\Gamma_n \leq 2\max_j \Sigma_{X,jj} \leq 2C_1 \quad \text{with probability} \geq 1 - 2\exp(\log d - c_2 n). \tag{D.10}$$

Call this event $\Omega_{n,\mathbf{X},2}$. Now, going back to the definition of $g(\mathbf{X}, p, d)$ in equation equation D.7, we first note that, on the event $\Omega_{n,\mathbf{X},1} \cap \Omega_{n,\mathbf{X},2}$:

$$\sigma_* = \sqrt{\frac{2Q^2(p)}{p}\Gamma_n} \leq 2\sqrt{\frac{C_1 Q^2(p)}{p}} \triangleq 2\sqrt{C_1 \Psi(p)}.$$

It is immediate from the definition of $Q(p)$ that $\Psi(p) \sim 1/(-4p \log p)$ for $p$ close to 0. Therefore for all small $p$ and large $d$, $\sigma_*^2 \log d \geq 1$ and consequently $\max\{\sigma_* \sqrt{\log d}, \sigma_*^2 \log d\} = \sigma_*^2 \log d$. Hence, we have on the event $\Omega_{n,\mathbf{X},1} \cap \Omega_{n,\mathbf{X},2}$:

$$g(\mathbf{X}, p, d) \leq C_2\sqrt{\Psi(p) \log d} + 2C_1 + \sigma_X\sqrt{2c_1 p \log d}.$$

It is immediate that the dominating term is the first term, which implies:

$$g(\mathbf{X}, p, d) \leq 3C_2\sqrt{\Psi(p) \log d}.$$

We now use this bound in equation equation D.1. Note that:

$$\mathbb{P}\left(Z(t) \geq \mathbb{E}[Z(t) \mid \mathbf{X}] + y\right)$$
$$\geq \mathbb{P}\left(Z(t) \geq \mathbb{E}[Z(t) \mid \mathbf{X}] + y, \Omega_{n,\mathbf{X},1} \cap \Omega_{n,\mathbf{X},2}\right)$$
$$\geq \mathbb{P}\left(Z(t) \geq 3CC_2 t\log d\sqrt{\frac{\Psi(p)}{n}} + y, \Omega_{n,\mathbf{X},1} \cap \Omega_{n,\mathbf{X},2}\right)$$
$$\geq \mathbb{P}\left(Z(t) \geq 3CC_2 t\log d\sqrt{\frac{\Psi(p)}{n}} + y\right) + \mathbb{P}(\Omega_{n,\mathbf{X},1} \cap \Omega_{n,\mathbf{X},2}) - 1$$

Therefore,

$$\mathbb{P}\left(Z(t) \geq 3CC_2 t\log d\sqrt{\frac{\Psi(p)}{n}} + y\right) \leq \exp\left(-\frac{8ny^2}{\tau^4}\right) + \mathbb{P}((\Omega_{n,\mathbf{X},1} \cap \Omega_{n,\mathbf{X},2})^c)$$
$$\leq \exp\left(-\frac{8ny^2}{\tau^4}\right) + 2\exp(-(c_1 - 1)\log d) + 2\exp(\log d - c_2 n). \tag{D.11}$$

Choosing $y = C_3 t\log d\sqrt{\Psi(p)/n}$, we have:

$$\mathbb{P}\left(Z(t) \geq 3CC_2 t\log d\sqrt{\frac{\Psi(p)}{n}} + C_3 t\log d\sqrt{\frac{\Psi(p)}{n}}\right)$$
$$\leq \exp\left(-\frac{8C_3^2 t^2 \log^2 d\, \Psi(p)}{\tau^4}\right) + 2\exp(-(c_1 - 1)\log d) + 2\exp(\log d - c_2 n). \tag{D.12}$$

**Modification 4:** Our last modification, not modification per se, but an application of peeling argument. Infact we want an upper bound on the event $\mathcal{E}$ defined as:

$$\mathcal{E} = \left\{Z(t) \geq 3eCC_2 t\log d\sqrt{\frac{\Psi(p)}{n}} + C_3 et\log d\sqrt{\frac{\Psi(p)}{n}} \quad \text{for some } t \in [1, \sqrt{d}]\right\}.$$

Note that $t$ denotes the $\ell_1$ norm of a a vector $u$ such that $\|u\|_2 = 1$. Therefore, $t \in [1, \sqrt{d}]$. Also recall that $Z(t)$ is the suprema of the empirical process over all vectors $u$ such that $\|u\|_2 = 1$ and $\|u\|_1 \leq t$. In peeling, we write $\mathcal{E}$ as union of disjoint events. Define $\mathcal{E}_j$ as:

$$\mathcal{E}_j = \left\{Z(t) \geq 3eCC_2 t\log d\sqrt{\frac{\Psi(p)}{n}} + C_3 et\log d\sqrt{\frac{\Psi(p)}{n}} \quad \text{for some } t \in [\sqrt{d}/e^j, \sqrt{d}/e^{j-1}]\right\}.$$

Therefore,

$$\mathcal{E} \subseteq \cup_{j=1}^{\lceil \frac{1}{2} \log d \rceil} \mathcal{E}_j \implies \mathbb{P}(\mathcal{E}) \leq \sum_{j=1}^{\lceil \frac{1}{2} \log d \rceil} \mathbb{P}(\mathcal{E}_j).$$

Now observe that, for any $t \in [\sqrt{d}/e^j, \sqrt{d}/e^{j-1}]$, we have $Z(t) \leq Z(\sqrt{d}/e^{j-1})$ and also

$$3eCC_2 t \log d \sqrt{\frac{\Psi(p)}{n}} + C_3 et \log d \sqrt{\frac{\Psi(p)}{n}} \geq 3eCC_2 \frac{\sqrt{d}}{e^j} \log d \sqrt{\frac{\Psi(p)}{n}} + C_3 e \frac{\sqrt{d}}{e^j} \log d \sqrt{\frac{\Psi(p)}{n}}$$

$$\geq 3CC_2 \frac{\sqrt{d}}{e^{j-1}} \log d \sqrt{\frac{\Psi(p)}{n}} + C_3 \frac{\sqrt{d}}{e^{j-1}} \log d \sqrt{\frac{\Psi(p)}{n}}.$$

Therefore:

$$\mathbb{P}(\mathcal{E}_j) \leq \mathbb{P}\left( Z\left( \frac{\sqrt{d}}{e^{j-1}} \right) \geq 3CC_2 \frac{\sqrt{d}}{e^{j-1}} \log d \sqrt{\frac{\Psi(p)}{n}} + C_3 \frac{\sqrt{d}}{e^{j-1}} \log d \sqrt{\frac{\Psi(p)}{n}} \right)$$

$$\leq \exp\left( -\frac{8C_3^2 d \log^2 d \Psi(p)}{e^{2j-2} \tau^4} \right) + 2\exp(-(c_1 - 1) \log d) + 2\exp(\log d - c_2 n)$$

$$\leq \exp\left( -c_4 \log^2 d \Psi(p) \right) + 2\exp(-(c_1 - 1) \log d) + 2\exp(\log d - c_2 n)$$

Hence:

$$\mathbb{P}(\mathcal{E}) \leq \exp\left( \frac{1}{2} \log d + 1 - c_4 \log^2 d \Psi(p) \right) + 2\exp(1 - (c_1 - 3/2) \log d) + 2\exp\left( \frac{3}{2} \log d + 1 - c_2 n \right).$$

On the event $\mathcal{E}^c$ (which is a high probability event):

$$Z(t) \leq 3eCC_2 t \log d \sqrt{\frac{\Psi(p)}{n}} + C_3 et \log d \sqrt{\frac{\Psi(p)}{n}} \quad \text{for all } t \in [1, \sqrt{d}].$$

Now let us conclude with the entire roadmap of the proof. First, following the same line of argument as of Negahban et al. (2009) we show that

$$\delta L_n(u) \geq L_\psi(T) \frac{1}{n} \sum_i \phi_\tau \left( (u^\top \mathbf{Z}_i)^2 \mathbb{1}_{|\mathbf{z}_i^\top \boldsymbol{\beta}_*| \leq T} \right)$$

$$= L_\psi(T) \|u\|^2 \frac{1}{n} \sum_i \phi_\tau \left( \left( u^\top \mathbf{Z}_i / \|u\| \right)^2 \mathbb{1}_{|\mathbf{z}_i^\top \boldsymbol{\beta}_*| \leq T} \right)$$

$$= L_\psi(T) \|u\|^2 \mathbb{P}_n(g_{u/\|u\|}(Z))$$

$$= L_\psi(T) \|u\|_2^2 \left\{ P(g_{u/\|u\|}(Z)) + (\mathbb{P}_n - P) g_{u/\|u\|}(Z) \right\}$$

We have proved in Modification 1 that $P(g_{u/\|u\|}(Z)) \geq \kappa_l$. Therefore,

$$\delta L_n(u) \geq L_\psi(T) \|u\|_2^2 \left\{ \kappa_l + (\mathbb{P}_n - P) g_{u/\|u\|}(Z) \right\}$$

Now for any $u$,

$$(\mathbb{P}_n - P) g_{u/\|u\|}(Z) \leq Z\left( \left\| \frac{u}{\|u\|_2} \right\|_1 \right)$$

$$\leq \left( 3eCC_2 \log d \sqrt{\frac{\Psi(p)}{n}} + C_3 e \log d \sqrt{\frac{\Psi(p)}{n}} \right) \frac{\|u\|_1}{\|u\|_2}.$$

Hence, we conclude that:

$$\delta L_n(u) \geq L_\psi(T) \|u\|_2^2 \left\{ \kappa_l - \left( C_4 \log d \sqrt{\frac{\Psi(p)}{n}} \right) \frac{\|u\|_1}{\|u\|_2} \right\}.$$

## D.2   PROOF OF LEMMA C.2

Recall that we have:

$$\nabla L_n(\boldsymbol{\beta}^{(0)}) = \frac{1}{n} \sum_i \mathbf{Z}_i \left\{ \mathbf{Y}_i - \psi'(\mathbf{Z}_i^\top \boldsymbol{\beta}^{(0)}) \right\} .$$

Now consider the $j^{th}$ element of $\nabla L_n(\boldsymbol{\beta}^{(0)})$, i.e.,

$$\nabla L_n(\boldsymbol{\beta}^{(0)})_j = \frac{1}{n} \sum_i \mathbf{Z}_{ij} \left\{ \mathbf{Y}_i - \psi'(\mathbf{Z}_i^\top \boldsymbol{\beta}^{(0)}) \right\} .$$

First, we show that conditional on $Z_{1j}, \ldots, Z_{nj}$, the terms are mean 0 (which is true from the definition of $\mathbf{Y}_i$), independent subgaussian random variable. The subgaussianity follows from the fact that:

$$\left\| \sum_i \mathbf{Z}_{ij} \left\{ \mathbf{Y}_i - \psi'(\mathbf{Z}_i^\top \boldsymbol{\beta}^{(0)}) \right\} \right\|_{\psi_2}^2 \le C \sum_i \mathbf{Z}_{ij}^2 \|\mathbf{Y}_i - \psi'(\mathbf{Z}_i^\top \boldsymbol{\beta}^{(0)})\|_{\psi_2}^2 \le C \sum_i \mathbf{Z}_{ij}^2 .$$

Here $C$ is some absolute constant. Therefore, we have:

$$\mathbb{P}\left( \max_{1 \le j \le d} \left| \frac{1}{\sqrt{n}} \sum_i \mathbf{Z}_{ij} \left\{ \mathbf{Y}_i - \psi'(\mathbf{Z}_i^\top \boldsymbol{\beta}^{(0)}) \right\} \right| \ge t \mid \mathbf{Z} \right) \le c_1 \exp\left( \log d - c_2 \frac{t^2}{\max_{1 \le j \le d} \frac{1}{n} \sum_{i=1}^n \mathbf{Z}_{ij}^2} \right)$$
(D.13)

We next bound the term in the tail bound $\max_{1 \le j \le d}(\sum_{i=1}^n \mathbf{Z}_{ij}^2)/n$. Towards that end, first observe that:

$$\max_{1 \le j \le d} \frac{1}{n} \sum_{i=1}^n \mathbf{Z}_{ij}^2 = \max_{1 \le j \le d} \frac{1}{n} \mathbf{Z}_{*j}^\top \mathbf{Z}_{*j} = \max_{1 \le j \le d} \frac{1}{n^2 p} e_j^\top \mathbf{X}^\top \mathbf{A}^\top \mathbf{A} \mathbf{X} e_j$$

$$= \max_{1 \le j \le d} \frac{1}{n} e_j^\top \mathbf{X} \left( \frac{\mathbf{A}^\top \mathbf{A}}{np} \right) \mathbf{X}^\top e_j .$$

**Remark D.8.** *If $p$ is unknown, i.e., we have $\hat{p}$, then, conditional on the event $p/2 \le \hat{p} \le 2p$, the above equality will be replaced by an inequality with an additional factor of 2.*

As we know $\mathbb{E}[\mathbf{X}^\top \mathbf{A}^\top \mathbf{A} \mathbf{X}]/(n^2 p) = \Sigma_X$ and if we define $\sigma_+ = \max_j \Sigma_{X,jj}$, we have:

$$\max_{1 \le j \le d} \frac{1}{n} \sum_{i=1}^n \mathbf{Z}_{ij}^2 \le \max_{1 \le j \le d} \left[ \frac{1}{n} e_j^\top \mathbf{X} \left( \frac{\mathbf{A}^\top \mathbf{A}}{np} \right) \mathbf{X}^\top e_j - \Sigma_{X,jj} \right] + \sigma_+ .$$

From Hanson-Wright inequality, we have for any matrix $\mathbf{Q}$ (independent of $\mathbf{X}$):

$$\mathbb{P}\left( \max_{1 \le j \le d} \left| \mathbf{X}_{*j}^\top \mathbf{Q} \mathbf{X}_{*j} - \mathbb{E}[\mathbf{X}_{*j}^\top \mathbf{Q} \mathbf{X}_{*j}] \right| \ge t \right) \le \exp\left( -c \min\left( \frac{t^2}{\kappa_u^4 \|\mathbf{Q}\|_F^2}, \frac{t}{\kappa_u^2 \|\mathbf{Q}\|_2} \right) \right) .$$

Here $\mathbf{Q} = (\mathbf{A}^\top \mathbf{A})/n^2 p$. We use some concentration results on $\mathbf{Q}$ in the rest of the proof. For notational convenience, set $\tilde{\mathbf{A}} = \mathbf{A}/\sqrt{np}$. We have the following concentration bound:

**Lemma D.9.** *For the Frobenous norm, we have with probability $\ge 1 - n^{-1}$:*

$$\|\tilde{\mathbf{A}}^\top \tilde{\mathbf{A}}\|_F^2 \le \mathbb{E}[\|\tilde{\mathbf{A}}^\top \tilde{\mathbf{A}}\|_F^2] + n + n^2 p^2 \le 2(n + n^2 p^2) .$$

*For the operator norm, we have with probability $\ge 1 - e^{\log n - \frac{np}{c}}$:*

$$\|\tilde{\mathbf{A}} - \mathbb{E}[\tilde{\mathbf{A}}]\|_{\mathrm{op}} \le 1 + 3\sqrt{2} \implies \|\tilde{\mathbf{A}}\|_{\mathrm{op}} \le 2\sqrt{np} .$$

**Remark D.10.** *This lemma remains the same for the self-loop.*

First, assume the above lemma is true, and consider the event so that the upper bound holds. Call that event $\mathcal{E}$. On this event we have:

$$\mathbb{P}\left( \max_{1 \le j \le d} \frac{1}{n} \left| \mathbf{X}_{*j}^\top \mathbf{Q} \mathbf{X}_{*j} - \mathbb{E}[\mathbf{X}_{*j}^\top \mathbf{Q} \mathbf{X}_{*j}] \right| \ge t \mid \mathcal{E} \right) \le \exp\left( \log d - c \min\left( \frac{n^2 t^2}{2\kappa_u^4(n + n^2 p^2)}, \frac{t}{2\kappa_u^2 p} \right) \right) .$$

Choosing

$$t = K \max \left\{ \sqrt{\frac{\log d}{n} + p^2 \log d}, p \log d \right\}$$

we conclude:

$$\max_{1 \le j \le d} \frac{1}{n} \left| \mathbf{X}_{*j}^\top \mathbf{Q} \mathbf{X}_{*j} - \mathbb{E}[\mathbf{X}_{*j}^\top \mathbf{Q} \mathbf{X}_{*j}] \right| \le K \max \left\{ \sqrt{\frac{\log d}{n} + p^2 \log d}, p \log d \right\} \le K \max \left\{ \sqrt{\frac{\log d}{n}}, p \log d \right\} .$$

Therefore, we have

$$\max_{1 \le j \le d} \frac{1}{n} \sum_{i=1}^{n} \mathbf{Z}_{ij}^2 \le \sigma_+ + K \max \left\{ \sqrt{\frac{\log d}{n}}, p \log d \right\} \le 2\sigma_+$$

with probability $\ge 1 - n^{-1} - e^{\log n - \frac{np}{c}}$. Call this event $\mathcal{E}_1$. Therefore, we have:

$$\mathbb{P} \left( \max_{1 \le j \le d} \left| \frac{1}{\sqrt{n}} \sum_i \mathbf{Z}_{ij} \left\{ \mathbf{Y}_i - \psi'(\mathbf{Z}_i^\top \boldsymbol{\beta}^{(0)}) \right\} \right| \ge t \right)$$

$$\le \mathbb{P} \left( \max_{1 \le j \le d} \left| \frac{1}{\sqrt{n}} \sum_i \mathbf{Z}_{ij} \left\{ \mathbf{Y}_i - \psi'(\mathbf{Z}_i^\top \boldsymbol{\beta}^{(0)}) \right\} \right| \ge t \mid \mathbf{Z} \in \mathcal{E}_1 \right) + \mathbb{P}(\mathcal{E}_1^c)$$

$$\le c_1 \exp \left( \log d - \frac{c_2 t^2}{2\sigma_+} \right) + \frac{1}{n} + \exp \left( \log n - \frac{np}{c} \right) .$$

Choosing $t = K\sqrt{\log d}$ we complete the proof.

### D.3 Proof of Lemma D.9

**Upper bound on $\|\tilde{\mathbf{A}}\|_{\mathrm{op}}$:** To establish a bound on $\|\tilde{\mathbf{A}}\|_{\mathrm{op}}$, we first center it:

$$\|\tilde{\mathbf{A}}\|_{\mathrm{op}} = \|\tilde{\mathbf{A}} - \mathbb{E}[\tilde{\mathbf{A}}]\|_{\mathrm{op}} + \|\mathbb{E}[\tilde{\mathbf{A}}]\|_{\mathrm{op}} .$$

A bound on $\|\mathbb{E}[\tilde{\mathbf{A}}]\|_{\mathrm{op}}$ directly follows from the definition:

$$\|\mathbb{E}[\tilde{\mathbf{A}}]\|_{\mathrm{op}} = \frac{1}{\sqrt{np}} \|\mathbb{E}[\mathbf{A}]\|_{\mathrm{op}} = \frac{1}{\sqrt{np}} \|p(\mathbf{1}\mathbf{1}^\top - I)\|_{\mathrm{op}} \le \sqrt{np} . \tag{D.14}$$

Now we bound $\|\tilde{\mathbf{A}} - \mathbb{E}[\tilde{\mathbf{A}}]\|_{\mathrm{op}}$. As $\tilde{\mathbf{A}} = \mathbf{A}/\sqrt{np}$, it is enough to bound $\|\mathbf{A} - \mathbb{E}[\mathbf{A}]\|_{\mathrm{op}}$. Using Corollary 3.12 and Remark 3.13 of Bandeira & Van Handel (2016) (with $\epsilon = 1/2$), which implies:

$$\mathbb{P} \left( \|\mathbf{A} - \mathbb{E}[\mathbf{A}]\|_{\mathrm{op}} \ge 3\sqrt{2}\tilde{\sigma} + t \right) \le e^{\log n - \frac{t^2}{c\sigma_*^2}}$$

where

$$\tilde{\sigma} = \max_i \sqrt{\sum_j \mathrm{Var}(\tilde{\mathbf{A}}_{ij})} = \sqrt{np(1-p)} \le \sqrt{np}, \quad \sigma_* = \max_{i,j} |\mathbf{A}_{ij}| \le 1 .$$

Therefore, we obtain:

$$\mathbb{P} \left( \|\mathbf{A} - \mathbb{E}[\mathbf{A}]\|_{\mathrm{op}} \ge 3\sqrt{2}\sqrt{np} + t \right) \le e^{\log n - \frac{t^2}{c}}$$

Taking $t = \sqrt{np}$, we get:

$$\mathbb{P} \left( \|\mathbf{A} - \mathbb{E}[\mathbf{A}]\|_{\mathrm{op}} \ge (1 + 3\sqrt{2})\sqrt{np} \right) \le e^{\log n - \frac{np}{c}}$$

As $\tilde{\mathbf{A}} = \mathbf{A}/\sqrt{np}$, we have:

$$\mathbb{P} \left( \|\tilde{\mathbf{A}} - \mathbb{E}[\tilde{\mathbf{A}}]\|_{\mathrm{op}} \ge (1 + 3\sqrt{2}) \right) \le e^{\log n - \frac{np}{c}} \tag{D.15}$$

Combining the bound on equation equation D.14 and equation D.15 we have with probability $\geq 1 - e^{\log n - \frac{np}{c}}$:

$$\|\tilde{\mathbf{A}}\|_{\mathrm{op}} \leq \sqrt{np} + (1 + 3\sqrt{2}) \leq 2\sqrt{np}. \tag{D.16}$$

**Finding a bound on** $\|\tilde{\mathbf{A}}^\top \tilde{\mathbf{A}}\|_F^2$**:** As before, we first find the expected value of $\|\tilde{\mathbf{A}}^\top \tilde{\mathbf{A}}\|_F^2$. For any $1 \leq i \neq j \leq n$:

$$\mathbb{E}[(\tilde{\mathbf{A}}^\top \tilde{\mathbf{A}})_{ij}^2] = \frac{1}{n^2 p^2} \mathbb{E}[(\mathbf{A}^\top \mathbf{A})_{ij}^2] = \frac{1}{n^2 p^2} \mathbb{E}\left[\left(\sum_{k=1}^n \mathbf{A}_{ki} \mathbf{A}_{kj}\right)^2\right]$$

$$= \frac{1}{n^2 p^2} \left(\sum_{k=1}^n \mathbb{E}[(\mathbf{A}_{ki} \mathbf{A}_{kj})^2] + \sum_{k \neq l} \mathbb{E}[(\mathbf{A}_{ki} \mathbf{A}_{kj})(\mathbf{A}_{li} \mathbf{A}_{lj})]\right)$$

$$\leq \frac{1}{n^2 p^2} (np^2 + n^2 p^4) = \frac{1}{n} + p^2.$$

Now for $1 \leq i = j \leq n$:

$$\mathbb{E}[(\tilde{\mathbf{A}}^\top \tilde{\mathbf{A}})_{ii}^2] = \frac{1}{n^2 p^2} \mathbb{E}\left[\left(\sum_{k=1}^n \mathbf{A}_{ki}^2\right)^2\right]$$

$$= \frac{1}{n^2 p^2} \left(\sum_k \mathbb{E}[\mathbf{A}_{ki}^4] + \sum_{k \neq l} \mathbb{E}[\mathbf{A}_{ki}^2 \mathbf{A}_{li}^2]\right)$$

$$\leq \frac{1}{n^2 p^2} (np + n^2 p^2) = \frac{1}{np} + 1.$$

Therefore, we have:

$$\mathbb{E}[\|\tilde{\mathbf{A}}^\top \tilde{\mathbf{A}}\|_F^2] = \sum_i \mathbb{E}[(\tilde{\mathbf{A}}^\top \tilde{\mathbf{A}})_{ii}^2] + \sum_{i \neq j} \mathbb{E}[(\tilde{\mathbf{A}}^\top \tilde{\mathbf{A}})_{ij}^2]$$

$$\leq n \left(\frac{1}{np} + 1\right) + n^2 \left(\frac{1}{n} + p^2\right)$$

$$\leq n + \frac{1}{p} + n^2 p^2 \leq n + n^2 p^2.$$

The last inequality follows from $p \geq n^{-1}$. Next, we establish a bound on the variance:

$$\mathrm{Var}\left(\|\tilde{\mathbf{A}}^\top \tilde{\mathbf{A}}\|_F^2\right) = \frac{1}{n^4 p^4} \mathrm{Var}(\|\mathbf{A}^\top \mathbf{A}\|_F^2)$$

$$= \frac{1}{n^4 p^4} \mathrm{Var}\left(\sum_{i,j} (\mathbf{A}^\top \mathbf{A})_{i,j}^2\right)$$

$$= \frac{1}{n^4 p^4} \left[\sum_{i,j} \mathrm{Var}\left((\mathbf{A}^\top \mathbf{A})_{i,j}^2\right) + \sum_{(i,j) \neq (k,l)} \mathrm{Cov}((\mathbf{A}^\top \mathbf{A})_{i,j}^2, (\mathbf{A}^\top \mathbf{A})_{k,l}^2)\right]$$

$$\triangleq \frac{1}{n^4 p^4} (T_1 + T_2).$$

We bound $T_1$ and $T_2$ separately. For that, we use some basic bounds on the raw moments of a binomial random variable; if $X \sim \mathrm{Ber}(n, p)$, then $\mathbb{E}[X^k] \leq C n^k p^k$ for all $k \in \{1, 2, 3, 4\}$, for some universal constant $C$ as long as $np \uparrow \infty$. Observe that $(\mathbf{A}^\top \mathbf{A})_{ii} \sim \mathrm{Ber}(n-1, p)$ and $(\mathbf{A}^\top \mathbf{A})_{ij} \sim \mathrm{Ber}(n-2, p^2)$ for $i \neq j$. For $T_1$ we have:

$$\sum_{i,j} \mathrm{Var}\left((\mathbf{A}^\top \mathbf{A})_{i,j}^2\right) = \sum_{i=1}^n \mathrm{Var}\left((\mathbf{A}^\top \mathbf{A})_{ii}^2\right) + \sum_{i \neq j} \mathrm{Var}\left((\mathbf{A}^\top \mathbf{A})_{ij}^2\right)$$

$$\leq \sum_i \mathbb{E}[(\mathbf{A}^\top \mathbf{A})_{ii}^4] + \sum_{i \neq j} \mathbb{E}[(\mathbf{A}^\top \mathbf{A})_{ij}^4]$$

$$\leq C(n^5 p^4 + n^6 p^8).$$

Next, we bound $T_2$, i.e., the covariance term. Note that if $(i, j, k, l)$ all are distinct, then covariance is $0$ as the terms are independent. Therefore, we only consider the cases when there are three or two distinct indices. We first deal with the terms of two distinct indices, i.e., $\text{Cov}((\mathbf{A}^\top \mathbf{A})_{ii}^2, (\mathbf{A}^\top \mathbf{A})_{ij}^2)$ where $i \neq j$. There are almost $n^2$ many terms of this form. For each of these types of terms:

$$\text{Cov}((\mathbf{A}^\top \mathbf{A})_{ii}^2, (\mathbf{A}^\top \mathbf{A})_{ij}^2) = \mathbb{E}\left[(\mathbf{A}^\top \mathbf{A})_{ii}^2 (\mathbf{A}^\top \mathbf{A})_{ij}^2\right] - \mathbb{E}\left[(\mathbf{A}^\top \mathbf{A})_{ii}^2\right] \mathbb{E}\left[(\mathbf{A}^\top \mathbf{A})_{ij}^2\right]$$

$$\leq \mathbb{E}\left[(\mathbf{A}^\top \mathbf{A})_{ii}^2 (\mathbf{A}^\top \mathbf{A})_{ij}^2\right]$$

$$= \mathbb{E}\left[\left(\sum_{k,k'=1}^n \mathbf{A}_{ki}^2 \mathbf{A}_{k'i} \mathbf{A}_{k'j}\right)^2\right]$$

$$= \mathbb{E}\left[\left(\sum_k \mathbf{A}_{ki}^3 \mathbf{A}_{kj} + \sum_{k \neq k'} \mathbf{A}_{ki}^2 \mathbf{A}_{k'i} \mathbf{A}_{k'j}\right)^2\right]$$

$$= \mathbb{E}\left[\left(\sum_k \mathbf{A}_{ki} \mathbf{A}_{kj} + \sum_{k \neq k'} \mathbf{A}_{ki} \mathbf{A}_{k'i} \mathbf{A}_{k'j}\right)^2\right]$$

$$\leq 2\left(\mathbb{E}\left[\left(\sum_k \mathbf{A}_{ki} \mathbf{A}_{kj}\right)^2\right] + \mathbb{E}\left[\left(\sum_{k \neq k'} \mathbf{A}_{ki} \mathbf{A}_{k'i} \mathbf{A}_{k'j}\right)^2\right]\right)$$

$$\leq 2C(n^2 p^4 + n^4 p^6).$$

Therefore, we have:

$$\sum_{i \neq j} \text{Cov}((\mathbf{A}^\top \mathbf{A})_{ii}^2, (\mathbf{A}^\top \mathbf{A})_{ij}^2) \leq 2C(n^4 p^4 + n^6 p^6). \tag{D.17}$$

Next, we bound the covariance terms of the form $\text{Cov}((\mathbf{A}^\top \mathbf{A})_{ij}^2, (\mathbf{A}^\top \mathbf{A})_{jk}^2)$, i.e. two terms share an index with $i \neq j \neq k$. There are almost $n^3$ such terms. For each term:

$$\text{Cov}((\mathbf{A}^\top \mathbf{A})_{ij}^2, (\mathbf{A}^\top \mathbf{A})_{jk}^2) \leq \mathbb{E}\left[(\mathbf{A}^\top \mathbf{A})_{ij}^2 (\mathbf{A}^\top \mathbf{A})_{jk}^2\right]$$

$$= \mathbb{E}\left[\left(\sum_{l,l'} \mathbf{A}_{li} \mathbf{A}_{lj} \mathbf{A}_{l'i} \mathbf{A}_{l'k}\right)^2\right]$$

$$= \mathbb{E}\left[\left(\sum_l \mathbf{A}_{li}^2 \mathbf{A}_{lj} \mathbf{A}_{lk} + \sum_{l \neq l'} \mathbf{A}_{li} \mathbf{A}_{lj} \mathbf{A}_{l'i} \mathbf{A}_{l'k}\right)^2\right]$$

$$= \mathbb{E}\left[\left(\sum_l \mathbf{A}_{li} \mathbf{A}_{lj} \mathbf{A}_{lk} + \sum_{l \neq l'} \mathbf{A}_{li} \mathbf{A}_{lj} \mathbf{A}_{l'i} \mathbf{A}_{l'k}\right)^2\right]$$

$$= 2\left(\mathbb{E}\left[\left(\sum_l \mathbf{A}_{li} \mathbf{A}_{lj} \mathbf{A}_{lk}\right)^2\right] + \mathbb{E}\left[\left(\sum_{l \neq l'} \mathbf{A}_{li} \mathbf{A}_{lj} \mathbf{A}_{l'i} \mathbf{A}_{l'k}\right)^2\right]\right)$$

$$\leq 2C(n^2 p^6 + n^4 p^8).$$

As there are almost $n^3$ such terms, we have:

$$\sum_{i \neq j \neq k} \text{Cov}((\mathbf{A}^\top \mathbf{A})_{ij}^2, (\mathbf{A}^\top \mathbf{A})_{jk}^2) \leq 2C(n^5 p^6 + n^7 p^8) \tag{D.18}$$

Therefore, combining equation D.17 and equation D.18, we have:

$$T_2 \leq C_1 \left( n^4 p^4 + n^6 p^6 + n^5 p^6 + n^7 p^8 \right) .$$

Combining the bounds on the variance and the covariance term, we conclude:

$$\text{Var}(\|\mathbf{A}^\top \mathbf{A}\|_F^2) \leq C_2(n^5 p^4 + n^6 p^8 + n^4 p^4 + n^6 p^6 + n^5 p^6 + n^7 p^8) \leq C_3(n^5 p^4 + n^6 p^6 + n^7 p^8).$$

Here the last equality follows from the fact that $n^5 p^4 \geq n^4 p^4$, $n^6 p^6 \geq n^6 p^8$ and $n^6 p^6 \geq n^4 p^4$ (as $np \geq 1$). As a consequence, we have:

$$\text{Var}(\|\tilde{\mathbf{A}}^\top \tilde{\mathbf{A}}\|_F^2) = \frac{1}{n^4 p^4} \text{Var}(\|\mathbf{A}^\top \mathbf{A}\|_F^2) \leq C_3(n + n^2 p^2 + n^3 p^4) .$$

The last step involves an application of Chebychev's inequality:

$$\mathbb{P}\left( \|\tilde{\mathbf{A}}^\top \tilde{\mathbf{A}}\|_F^2 - \mathbb{E}[\|\tilde{\mathbf{A}}^\top \tilde{\mathbf{A}}\|_F^2] \geq t \right) \leq \frac{\text{Var}(\|\tilde{\mathbf{A}}^\top \tilde{\mathbf{A}}\|_F^2)}{t^2} .$$

Taking $t = n + n^2 p^2$, we have:

$$\mathbb{P}\left( \|\tilde{\mathbf{A}}^\top \tilde{\mathbf{A}}\|_F^2 - \mathbb{E}[\|\tilde{\mathbf{A}}^\top \tilde{\mathbf{A}}\|_F^2] \geq n + n^2 p^2 \right) \leq \frac{\text{Var}(\|\tilde{\mathbf{A}}^\top \tilde{\mathbf{A}}\|_F^2)}{(n + n^2 p^2)^2} \leq C_3 \frac{n(1 + np^2 + n^2 p^4)}{n^2(1 + 2np^2 + n^2 p^4)} \leq \frac{C_3}{n} .$$

Therefore, with probability $\geq 1 - n^{-1}$:

$$\|\tilde{\mathbf{A}}^\top \tilde{\mathbf{A}}\|_F^2 \leq \mathbb{E}[\|\tilde{\mathbf{A}}^\top \tilde{\mathbf{A}}\|_F^2] + n + n^2 p^2 \leq 2(n + n^2 p^2) .$$

# E  EXTENSION OF THEOREM 4.4 UNDER MULTIPLE SOURCE

In Theorem 4.4, we have established the convergence guarantee of $\hat{\beta}$ on a domain under network dependency. This section presents some ideas for extending our analysis when we have data from multiple related source domains. We have conjectured a theorem (Theorem E.10) and lay down the steps needed to prove it. There is one conjecture (Conjecture E.8), which, if true, will lead to a complete proof of the theorem.

We start a simple setting with one source domain $\mathcal{A}$ and one target domain. Consider the transfer learning setup, in which we have $n_1$ observations from the source domain and $n_0$ observations from the target domain. We assume that $p_0 = p_1 = p$. Define $\mathbf{Z} = (\mathbf{AX})/\sqrt{np} \in \mathbb{R}^{n \times d}$, and $\mathbf{Z}_i$ is the $i$th row of $\mathbf{Z}$. Given the logistic regression, the inverse link function (McCullagh, 2019) for logit link is $\psi'(u) = \text{logistic}(u)$, where $\psi(u) = \log(1 + e^u)$. The method is similar to the proposed method, i.e., we have a two-step estimator:

1. Step 1: First estimate $\hat{\boldsymbol{\beta}}^{\mathcal{A}}$ as:

$$\hat{\boldsymbol{\beta}}^{\mathcal{A}} = \arg\min_{\boldsymbol{\beta}} \ - \frac{1}{n_{\mathcal{A}} + n_0} \sum_{k \in \{0, \mathcal{A}\}} \left\{ (\mathbf{Y}^{(k)})^\top \mathbf{Z}^{(k)} \boldsymbol{\beta} - \log\left(1 + e^{\mathbf{Z}_i^\top \boldsymbol{\beta}}\right) \right\} + \lambda_{\boldsymbol{\beta}} \|\boldsymbol{\beta}\|_1$$

2. Step 2: Then estimate the correction $\hat{\boldsymbol{\delta}}^{\mathcal{A}}$ only based on the target observations:

$$\hat{\boldsymbol{\delta}}^{\mathcal{A}} = \arg\min_{\boldsymbol{\delta}} \ - \frac{1}{n_0} \left\{ (\mathbf{Y}^{(0)})^\top \mathbf{Z}^{(0)} (\hat{\boldsymbol{\beta}}^{\mathcal{A}} + \boldsymbol{\delta}) - \log\left(1 + e^{\mathbf{Z}^\top (\hat{\boldsymbol{\beta}}^{\mathcal{A}} + \boldsymbol{\delta})}\right) \right\} + \lambda_{\delta} \|\boldsymbol{\delta}\|_1 .$$

Our final estimator for the target coefficient $\boldsymbol{\beta}^{(0)}$ is $\hat{\boldsymbol{\beta}}^{(0)} = \hat{\boldsymbol{\beta}}^{\mathcal{A}} + \hat{\boldsymbol{\delta}}^{\mathcal{A}}$. We must extend Theorem 1 of Tian & Feng (2023) to handle the network dependency. There are two key steps in the proof: i) to establish the rate of convergence of $\hat{\boldsymbol{\beta}}^{\mathcal{A}}$ (which is the estimator of $\boldsymbol{\beta}^{\mathcal{A}}$ obtained by combining all the observations from both the set of transferable source $\mathcal{A}$ and the target domain, and ii) then establish the rate of convergence of the $\boldsymbol{\beta}^{(0)}$ is $\hat{\boldsymbol{\beta}}^{(0)} = \hat{\boldsymbol{\beta}}^{\mathcal{A}} + \hat{\boldsymbol{\delta}}^{\mathcal{A}}$, where $\hat{\boldsymbol{\delta}}^{\mathcal{A}}$ is obtained using only the target observations.

We will use bolded $\boldsymbol{\psi}'$ hereafter to denote the vector with each component from the scalar function $\psi'$ with corresponding variables. Define $\hat{\boldsymbol{u}}^{\mathcal{A}} = \hat{\boldsymbol{\beta}}^{\mathcal{A}} - \boldsymbol{\beta}^{\mathcal{A}}$, $\mathcal{D} = \left\{ \left( \mathbf{Z}^{(k)}, \mathbf{Y}^{(k)} \right) \right\}_{k \in \{0, \mathcal{A}\}}$, and $L(\boldsymbol{\beta}, \mathcal{D})$

is the negative log likelihood on the combined sample $\mathcal{D}$:

$$L(\boldsymbol{\beta}, \mathcal{D}) = -\frac{1}{n_{\mathcal{A}} + n_0} \sum_{k \in \{0, \mathcal{A}\}} \left(\mathbf{Y}^{(k)}\right)^T \mathbf{Z}^{(k)} \boldsymbol{\beta} + \frac{1}{n_{\mathcal{A}} + n_0} \sum_{k \in \{0, \mathcal{A}\}} \sum_{i=1}^{n_k} \psi\left(\boldsymbol{\beta}^T \mathbf{Z}_i^{(k)}\right)$$

$$\nabla L(\boldsymbol{\beta}, \mathcal{D}) = -\frac{1}{n_{\mathcal{A}} + n_0} \sum_{k \in \{0, \mathcal{A}\}} \left(\mathbf{Z}^{(k)}\right)^T \mathbf{Y}^{(k)} + \frac{1}{n_{\mathcal{A}} + n_0} \sum_{k \in \{0, \mathcal{A}\}} \left(\mathbf{Z}^{(k)}\right)^T \boldsymbol{\psi}'\left(\boldsymbol{\beta}^T \mathbf{Z}_i^{(k)}\right)$$

$$\delta L(\boldsymbol{u}, \mathcal{D}) = L\left(\boldsymbol{\beta}^{\mathcal{A}} + \boldsymbol{u}, \mathcal{D}\right) - L\left(\boldsymbol{\beta}^{\mathcal{A}}\right) - \nabla L\left(\boldsymbol{\beta}^{\mathcal{A}}\right)^T \boldsymbol{u}.$$

We present useful assumptions and lemmas first.

**Assumption E.1.** *(SubGaussian Assumption.) For any $\boldsymbol{a} \in \mathbb{R}^p$, $\boldsymbol{a}^T \mathbf{X}^{(k)}$ are $\kappa_u \|\boldsymbol{a}\|_2^2$-subGaussian variables with zero mean for all $k \in \{0, \mathcal{A}\}$, where $\kappa_u$ is a positive constant.*

**Assumption E.2.** *(Positive Definite Covariance Assumption.) Denote the covariance matrix of $\mathbf{X}^{(k)}$ as $\boldsymbol{\Sigma}_{\mathbf{X}}^{(k)}$, $k \in \{0, \mathcal{A}\}$, we require that $\lambda_{\min}\left(\boldsymbol{\Sigma}_{\mathbf{X}}^{(k)}\right) \geq \kappa_l > 0$, where $\kappa_l$ is a positive constant.*

**Assumption E.3.** *(Connectivity Bound of Network.) $p_k > \frac{\log n_k}{n_k}$, $k \in \{0, \mathcal{A}\}$.*

**Assumption E.4.** *Denote*

$$\widetilde{\boldsymbol{\Sigma}}_h = \sum_{k \in \{0, \mathcal{A}\}} \alpha_k \mathbb{E}\left[\mathbf{S}^{(k)}\left(\mathbf{S}^{(k)}\right)^T \int_0^1 \psi''\left(\left(\mathbf{S}^{(k)}\right)^T \boldsymbol{\beta}^{(0)} + t\left(\mathbf{S}^{(k)}\right)^T \left(\boldsymbol{\beta}^{\mathcal{A}} - \boldsymbol{\beta}^{(0)}\right)\right) dt\right]$$

*and $\widetilde{\boldsymbol{\Sigma}}_h^{(k)} = \mathbb{E}\left[\int_0^1 \psi''\left(\left(\mathbf{S}^{(k)}\right)^T \boldsymbol{\beta}^{(0)} + t\left(\mathbf{Z}^{(k)}\right)^T \left(\boldsymbol{\beta}^{(k)} - \boldsymbol{\beta}^{(0)}\right)\right) dt \cdot \mathbf{S}^{(k)}\left(\mathbf{S}^{(k)}\right)^T\right]$. It holds that $\sup_{k \in \{0, \mathcal{A}\}} \left\|\widetilde{\boldsymbol{\Sigma}}_h^{-1} \widetilde{\boldsymbol{\Sigma}}_h^{(k)}\right\|_1 < \infty$.*

**Lemma E.5.** *Under Assumptions E.1 and E.4,*

$$\left\|\boldsymbol{\delta}^{\mathcal{A}}\right\|_1 = \left\|\boldsymbol{\beta}^{\mathcal{A}} - \boldsymbol{\beta}^{(0)}\right\|_1 \leq C_l h$$

*where $\boldsymbol{\beta}^{\mathcal{A}}$ is the true coefficient of Step 1, $\boldsymbol{\delta}^{\mathcal{A}}$ is the true coefficient of Step 2, and $\boldsymbol{\beta}^{(0)}$ is the true coefficient of target domain. And $C_l := \sup_{k \in \mathcal{T} \cup \mathcal{A}} \left\|\widetilde{\boldsymbol{\Sigma}}_h^{-1} \widetilde{\boldsymbol{\Sigma}}_h^{(k)}\right\|_1 < \infty$.*

**Lemma E.6.** *Under Assumptions E.1 and E.2, there exists some positive constants $\kappa_l$ and $C_4$ such that,*

$$\delta L(\boldsymbol{u}, \mathcal{D}) \geq L_{\psi}(T) \|\boldsymbol{u}\|_2^2 \left\{\kappa_l - \left(C_4 \log d \sqrt{\frac{\Psi(p)}{n}}\right) \frac{\|\boldsymbol{u}\|_1}{\|\boldsymbol{u}\|_2}\right\}$$

*with probability at least $1 - (\exp\left(\frac{1}{2}\log d + 1 - c_4 \log^2 d \Psi(p)\right) + 2\exp(1 - (c_1 - 3/2)\log d) + 2\exp\left(\frac{3}{2}\log d + 1 - c_2 n\right))$, where $T$ is some constant, $L_{\psi}(T) = \min_{u \leq |2T|} \psi''(u)$, and $\Psi(p) \sim 1/(-4p \log p)$.*

**Lemma E.7.** *Under Assumption E.1, there are universal positive constants $(c_6, c_7, c_8)$ such that*

$$\frac{1}{n_{\mathcal{A}} + n_0} \left\|\sum_{k \in \{0, \mathcal{A}\}} \left(\mathbf{Z}^{(k)}\right)^T \left[\mathbf{Y}^{(k)} - \boldsymbol{\psi}'\left(\mathbf{Z}^{(k)} \boldsymbol{\beta}^{(k)}\right)\right]\right\|_{\infty} \lesssim \sqrt{\frac{\log d}{n_{\mathcal{A}} + n_0}}$$

*with probability $1 - (c_6 \left(d^{-c_7} + \sum_k n_k^{-1} + \sum_k e^{\log n_k - n_k p_k / c_8}\right))$.*

**Conjecture E.8.** *Under Assumption E.1, there are universal positive constants $(c_9, c_{10}, c_{11})$ such that*

$$\frac{1}{n_{\mathcal{A}} + n_0} \left\|\sum_{k \in \{0, \mathcal{A}\}} \left(\mathbf{Z}^{(k)}\right)^T \left[\boldsymbol{\psi}'\left(\mathbf{Z}^{(k)} \boldsymbol{\beta}^{(k)}\right) - \boldsymbol{\psi}'\left(\mathbf{Z}^{(k)} \boldsymbol{\beta}^{\mathcal{A}}\right)\right]\right\|_{\infty} \lesssim \sqrt{\frac{\log d}{n_{\mathcal{A}} + n_0}}$$

*with probability $1 - c_9 d^{-c_{10}} + \exp\left[-c_{11}\left(n_{\mathcal{A}} + n_0\right)\right]$.*

**Lemma E.9.** *With high probability of at least $1 - d^{-\tilde{K}} - n^{-1} - e^{\log n - \frac{np}{c}}$, where $\tilde{K}$ and $c$ are some constants, there exists some constant $C$ such that:*

$$\frac{\|\mathbf{AXv}\|_2^2}{n^2 p \|\mathbf{v}\|_2^2} = \frac{\mathbf{v}^\top \mathbf{X}^\top \mathbf{A}^\top \mathbf{AXv}}{n^2 p \|\mathbf{v}\|_2^2} \leq C \qquad \forall \, \mathbf{v} : \|\mathbf{v}_{S^c}\|_1 \leq \kappa \|\mathbf{v}_S\|_1 \, .$$

**Theorem E.10.** *(Convergence rate of Trans-GCR). Under Assumptions E.1, E.2, E.3, E.4, suppose $h \ll \sqrt{\frac{n_0}{\log d}}, h \leq c\sqrt{s}, n_0 \geq c \log d$ and $n_{\mathcal{A}} \geq cs \log d$, where $c > 0$ is a constant, we have*

$$\sup\nolimits_{\xi \in \Xi(s,h)} \mathbb{P}\left( \|\hat{\boldsymbol{\beta}}^{(0)} - \boldsymbol{\beta}^{(0)}\|_2 \lesssim h \log d \sqrt{\frac{\Psi(p)}{n_{\mathcal{A}} + n_0}} + \sqrt{\frac{s \log d}{n_{\mathcal{A}} + n_0}} + \left(\frac{\log d}{n_{\mathcal{A}} + n_0}\right)^{1/4} \sqrt{h} \right) \geq 1 - n_0^{-1}$$

## E.1 PROOF OF LEMMA E.5

By definition in (E.1),

$$\sum_{k \in \{0, \mathcal{A}\}} \alpha_k \mathbb{E}\left\{ \left[ \psi'\left(\left(\boldsymbol{\beta}^{\mathcal{A}}\right)^T \mathbf{Z}^{(k)}\right) - \psi'\left(\left(\boldsymbol{\beta}^{(k)}\right)^T \mathbf{Z}^{(k)}\right) \right] \mathbf{Z}^{(k)} \right\} = \mathbf{0}_p$$

which implies

$$\sum_{k \in \{0, \mathcal{A}\}} \alpha_k \mathbb{E}\left\{ \left[ \psi'\left(\left(\boldsymbol{\beta}^{\mathcal{A}}\right)^T \mathbf{Z}^{(k)}\right) - \psi'\left(\left(\boldsymbol{\beta}^{(0)}\right)^T \mathbf{Z}^{(k)}\right) \right] \mathbf{Z}^{(k)} \right\}$$

$$= \sum_{k \in \{0, \mathcal{A}\}} \alpha_k \mathbb{E}\left\{ \left[ \psi'\left(\left(\boldsymbol{\beta}^{(k)}\right)^T \mathbf{Z}^{(k)}\right) - \psi'\left(\left(\boldsymbol{\beta}^{(0)}\right)^T \mathbf{Z}^{(k)}\right) \right] \mathbf{Z}^{(k)} \right\}$$

By Taylor expansion,

$$\sum_{k \in \{0, \mathcal{A}\}} \alpha_k \mathbb{E}\left[ \int_0^1 \psi''\left(\left(\boldsymbol{\beta}^{\mathcal{A}}\right)^T \mathbf{Z}^{(k)} + t\left(\boldsymbol{\beta}^{\mathcal{A}} - \boldsymbol{\beta}^{(0)}\right)^T \mathbf{Z}^{(k)}\right) \mathbf{Z}^{(k)} \left(\mathbf{Z}^{(k)}\right)^T \right] \left(\boldsymbol{\beta}^{\mathcal{A}} - \boldsymbol{\beta}^{(0)}\right)$$

$$= \sum_{k \in \{0, \mathcal{A}\}} \alpha_k \mathbb{E}\left[ \int_0^1 \psi''\left(\left(\boldsymbol{\beta}^{(k)}\right)^T \mathbf{Z}^{(k)} + t\left(\boldsymbol{\beta}^{(k)} - \boldsymbol{\beta}^{(0)}\right)^T \mathbf{Z}^{(k)}\right) \mathbf{Z}^{(k)} \left(\mathbf{Z}^{(k)}\right)^T \right] \left(\boldsymbol{\beta}^{(k)} - \boldsymbol{\beta}^{(0)}\right)$$

Therefore, by Assumption E.4, $\left\|\boldsymbol{\beta}^{\mathcal{A}} - \boldsymbol{\beta}^{(0)}\right\|_1 \leq \sum_{k \in \mathcal{A}} \alpha_k \left\|\widetilde{\boldsymbol{\Sigma}}_h^{-1} \widetilde{\boldsymbol{\Sigma}}_h^{(k)}\right\|_1 \cdot \left\|\boldsymbol{\beta}^{(k)} - \boldsymbol{\beta}^{(0)}\right\|_1 \leq C_l h$.

## E.2 PROOF OF LEMMA E.6

See proof of Lemma C.1.

## E.3 PROOF OF LEMMA E.7

Here, we define $\mathbf{B}_i$ as the $i$-th row of matrix $\mathbf{B}$, and $\mathbf{B}_{(j)}$ as the $j$-th column of matrix $\mathbf{B}$. For a fixed index $j \in \{1, 2, \ldots, p\}$, we denote $R_{ij}^{(k)} := \mathbf{Z}_{ij}^{(k)}\left(Y_i^{(k)} - \psi'\left(\left\langle \boldsymbol{\beta}^{(k)}, \mathbf{Z}_i^{(k)}\right\rangle\right)\right)$, and the $j$-th element of $\left(\mathbf{Z}^{(k)}\right)^T \left[\mathbf{Y}^{(k)} - \psi'\left(\mathbf{Z}^{(k)}\boldsymbol{\beta}^{(k)}\right)\right]$ can be written as $\sum_{i=1}^{n_k} R_{ij}^{(k)}$. Given the condition $\left\{\mathbf{Z}_i^{(k)}\right\}_{i=1}^{n_k}$, $y_i^{(k)}$ follow a Bernoulli distribution with parameter $\frac{\exp\left(\left\langle \boldsymbol{\beta}^{(k)}, \mathbf{Z}_i^{(k)}\right\rangle\right)}{1 + \exp\left(\left\langle \boldsymbol{\beta}^{(k)}, \mathbf{Z}_i^{(k)}\right\rangle\right)}$. For any $t \in \mathbb{R}$, we compute

$$\log \mathbb{E}\left[\exp\left(t R_{ij}^{(k)}\right) \mid \mathbf{Z}_i^{(k)}\right] = \log\left\{\mathbb{E}\left[\exp\left(t \mathbf{Z}_{ij}^{(k)} Y_i^{(k)}\right) \mid \mathbf{Z}_i^{(k)}\right] \exp\left(-t \mathbf{Z}_{ij}^{(k)} \psi'\left(\left\langle \boldsymbol{\beta}^{(k)}, \mathbf{Z}_i^{(k)}\right\rangle\right)\right)\right\}$$

$$= \psi\left(t \mathbf{Z}_{ij}^{(k)} + \left\langle \boldsymbol{\beta}^{(k)}, \mathbf{Z}_i^{(k)}\right\rangle\right) - \psi\left(\left\langle \boldsymbol{\beta}^{(k)}, \mathbf{Z}_i^{(k)}\right\rangle\right) - t \mathbf{Z}_{ij}^{(k)} \psi'\left(\left\langle \boldsymbol{\beta}^{(k)}, \mathbf{Z}_i^{(k)}\right\rangle\right)$$

By second-order Taylor series expansion, we have

$$\log \mathbb{E}\left[\exp\left(tR_{ij}^{(k)}\right) \mid \mathbf{Z}_i^{(k)}\right] = \frac{t^2}{2}\mathbf{Z}_{ij}^2\psi''\left(\left\langle\boldsymbol{\beta}^{(k)}, \mathbf{Z}_i^{(k)}\right\rangle + v_i t\mathbf{Z}_{ij}^{(k)}\right) \quad \text{for some } v_i \in [0,1]$$

Since this upper bound holds for each $i = 1, 2, \ldots, n_k$, we have shown that

$$\sum_{i=1}^{n_k}\log \mathbb{E}\left[\exp\left(tR_{ij}^{(k)}\right) \mid \mathbf{Z}_i^{(k)}\right] \leq \frac{t^2}{2}\left\{\sum_{i=1}^{n_k}\left(\mathbf{Z}_{ij}^{(k)}\right)^2\psi''\left(\left\langle\boldsymbol{\beta}^{(k)}, \mathbf{Z}_i^{(k)}\right\rangle + v_i t\mathbf{Z}_{ij}^{(k)}\right)\right\}$$

For the link function $\psi(x) = \log\{1 + \exp(x)\}$, it is easy to know that its second derivative $\psi''(x) = \exp(x)/(1 + \exp(x))^2$ takes values between 0 and 1, therefore the aforementioned equation can be simply bounded by an upper bound:

$$\sum_{i=1}^{n_k}\log \mathbb{E}\left[\exp\left(tR_{ij}^{(k)}\right) \mid \mathbf{Z}_i^{(k)}\right] \leq \frac{t^2}{2}\sum_{i=1}^{n_k}\left(\mathbf{Z}_{ij}^{(k)}\right)^2$$

and

$$\sum_{k\in\{0,\mathcal{A}\}}\sum_{i=1}^{n_k}\log \mathbb{E}\left[\exp\left(tR_{ij}^{(k)}\right) \mid \mathbf{Z}_i^{(k)}\right] \leq \frac{t^2}{2}\sum_{k\in\{0,\mathcal{A}\}}\sum_{i=1}^{n_k}\left(\mathbf{Z}_{ij}^{(k)}\right)^2$$

To control $\sum_{k\in\{0,\mathcal{A}\}}\sum_{i=1}^{n_k}\left(\mathbf{Z}_{ij}^{(k)}\right)^2$, it is easy to observe that it is a kind of quadratic forms respect to $\mathbf{X}_{(j)}^{(k)}$ so we will use Hanson-Wright inequality:

$$\sum_{k\in\{0,\mathcal{A}\}}\sum_{i=1}^{n_k}\left(\mathbf{Z}_{ij}^{(k)}\right)^2$$

$$= \sum_{k\in\{0,\mathcal{A}\}}\mathbf{Z}_{(j)}^{(k)^T}\mathbf{Z}_{(j)}^{(k)}$$

$$= \sum_{k\in\{0,\mathcal{A}\}}\left(\widetilde{\mathbf{A}}^{(k)}\mathbf{X}_{(j)}^{(k)}\right)^T\widetilde{\mathbf{A}}^{(k)}\mathbf{X}_{(j)}^{(k)}$$

$$\triangleq \sum_{k\in\{0,\mathcal{A}\}}\mathbf{X}_{(j)}^{(k)^T}\mathbf{Q}^{(k)}\mathbf{X}_{(j)}^{(k)}$$

$$= \mathbf{X}_{(j)}^T\mathbf{Q}\mathbf{X}_{(j)}$$

where $\mathbf{X}_{(j)} \in \mathbb{R}^{(n_{\mathcal{A}}+n_0)}$ represents the vector obtained by vertically concatenating $\mathbf{X}_{(j)}^{(k)}$, and $\mathbf{Q} \in \mathbb{R}^{(n_{\mathcal{A}}+n_0)\times(n_{\mathcal{A}}+n_0)}$ represents the block diagonal matrix with diagonal elements $\mathbf{Q}^{(k)}$. Hanson-Wright inequality tell us:

$$\mathbb{P}\left\{\left|\mathbf{X}_{(j)}^T\mathbf{Q}\mathbf{X}_{(j)} - \mathbb{E}\left[\mathbf{X}_{(j)}^T\mathbf{Q}\mathbf{X}_{(j)}\right]\right| > t\right\}$$

$$\leq 2\exp\left[-c\min\left(\frac{t^2}{\kappa_u^4\|\mathbf{Q}\|_{\text{F}}^2}, \frac{t}{\kappa_u^2\|\mathbf{Q}\|_2}\right)\right]$$

In our article on linear regression, we have already proven that

$$\|\mathbf{Q}\|_F^2 = \sum_{k \in \{0, \mathcal{A}\}} \|\mathbf{Q}^{(k)}\|_F^2 \leq \sum_{k \in \{0, \mathcal{A}\}} 2(n_k + (n_k p_k)^2)$$

$$\|\mathbf{Q}\|_2 = \max_{k \in \{0, \mathcal{A}\}} \|\mathbf{Q}^{(k)}\|_2 \leq 4 \max_{k \in \{0, \mathcal{A}\}} \{n_k p_k\}$$

with high probability $1 - \sum_k n_k^{-1} - \sum_k e^{\log n_k - \frac{n_k p_k}{c}}$ converge to 1, and
$\mathrm{E}\left[\mathbf{X}_{(j)}{}^T \mathbf{Q} \mathbf{X}_{(j)}\right] = \sigma_{jj}^2 (n_\mathcal{A} + n_0)$

So we have the tail bound

$$\mathbb{P}\left[\frac{1}{n_\mathcal{A} + n_0} \sum_{k \in \{0, \mathcal{A}\}} \sum_{i=1}^{n_k} \left(\mathbf{Z}_{ij}^{(k)}\right)^2 \geq C\right]$$

$$\leq 2\exp(-\frac{n_\mathcal{A} + n_0}{2}) + \sum_k n_k^{-1} + \sum_k e^{\log n_k - \frac{n_k p_k}{c}}$$

Define the event $\mathcal{E} = \left\{\max_{j=1,\ldots,p} \frac{1}{n_\mathcal{A} + n_0} \sum_{k \in \{0, \mathcal{A}\}} \sum_{i=1}^{n_k} \left(\mathbf{Z}_{ij}^{(k)}\right)^2 \leq C\right\}$, we have

$$\mathbb{P}\left[\mathcal{E}^c\right] \leq 2\exp\left(-\frac{n_\mathcal{A} + n_0}{2} + \log d\right) + \sum_k n_k^{-1} + \sum_k e^{\log n_k - \frac{n_k p_k}{c}}$$

$$\leq 2\exp(-c(n_\mathcal{A} + n_0)) + \sum_k n_k^{-1} + \sum_k e^{\log n_k - \frac{n_k p_k}{c}}$$

where we have used the fact that $n_\mathcal{A} \gg \log d$.

Given that $\left\{\mathbf{Z}_i^{(k)}\right\} \in \mathcal{E}$, using the independence between $R_{ij}^{(k)}$ given $\mathbf{Z}_i^{(k)}$, we have

$$\frac{1}{n_\mathcal{A} + n_0} \sum_{k \in \{0, \mathcal{A}\}} \sum_{i=1}^{n_k} \log \mathbb{E}\left[\exp\left(tR_{ij}^{(k)}\right) \mid \mathbf{Z}_i^{(k)}\right]$$

$$= \frac{1}{n_\mathcal{A} + n_0} \sum_{k \in \{0, \mathcal{A}\}} \sum_{i=1}^{n_k} \mathbb{E}\left[tR_{ij}^{(k)} \mid \mathbf{Z}_i^{(k)}\right]$$

$$\leq ct^2 \quad \text{for each } j = 1, 2, \ldots, d$$

By the Chernoff bound, we obtain

$$\mathbb{P}\left[\left|\frac{1}{n_\mathcal{A} + n_0} \sum_{k \in \{0, \mathcal{A}\}} \sum_{i=1}^{n_k} R_{ij}^{(k)}\right| \geq \delta \mid \mathbf{Z}_i\right] \leq 2\exp\left(-c(n_\mathcal{A} + n_0)\delta^2\right)$$

Combining this bound with the union bound yields

$$\mathbb{P}\left[\max_{j=1,\ldots,d} \left|\frac{1}{n_\mathcal{A} + n_0} \sum_{k \in \{0, \mathcal{A}\}} \sum_{i=1}^{n_k} R_{ij}^{(k)}\right| \geq t \mid \mathcal{E}\right] \leq 2\exp\left(-c(n_\mathcal{A} + n_0)t^2 + \log d\right)$$

Setting $t = c\sqrt{\frac{\log d}{n_\mathcal{A} + n_0}}$, and putting together the pieces yields

$$\mathbb{P}\left[\max_{j=1,\ldots,d}\left|\frac{1}{n_{\mathcal{A}}+n_0}\sum_{k\in\{0,\mathcal{A}\}}\sum_{i=1}^{n_k}R_{ij}^{(k)}\right|\geq c\sqrt{\frac{\log d}{n_{\mathcal{A}}+n_0}}\right]$$

$$\leq\mathbb{P}\left[\mathcal{E}^c\right]+\mathbb{P}\left[\max_{j=1,\ldots,d}\left|\frac{1}{n_{\mathcal{A}}+n_0}\sum_{k\in\{0,\mathcal{A}\}}\sum_{i=1}^{n_k}R_{ij}^{(k)}\right|\geq t\mid\mathcal{E}\right]$$

$$\leq c_6 d^{-c_7}+\sum_k n_k^{-1}+\sum_k e^{\log n_k-\frac{n_k p_k}{c_8}}$$

### E.4 PROOF OF LEMMA E.9

In this section, we prove that with high probability, there exists some constant $c_{12}$ such that:

$$\frac{\|\mathbf{Z}\mathbf{v}\|_2^2}{n\|\mathbf{v}\|_2^2}=\frac{\|\mathbf{A}\mathbf{X}\mathbf{v}\|_2^2}{n^2p\|\mathbf{v}\|_2^2}=\frac{\mathbf{v}^\top\mathbf{X}^\top\mathbf{A}^\top\mathbf{A}\mathbf{X}\mathbf{v}}{n^2p\|\mathbf{v}\|_2^2}\leq c_{12}\ \forall\ \mathbf{v}:\|\mathbf{v}_{S^c}\|_1\leq\kappa\|\mathbf{v}_S\|_1\,.$$

Using our previous notation, we define $\mathbf{Z}=(\mathbf{A}\mathbf{X})/\sqrt{np}$. Define events $\Omega_{n,1}=\left\{\|\mathbf{A}^\top\mathbf{A}\|_{\mathrm{op}}/np\leq 4np\right\}$ and $\Omega_{n,2}=\left\{\|\mathbf{A}^\top\mathbf{A}\|_F^2/np\leq 2(n+n^2p^2)\right\}$

Therefore, we have:

$$\mathbb{P}\left(\left\|\frac{(\mathbf{X}^\top\mathbf{A}^\top\mathbf{A}\mathbf{X})}{n^2p}-\Sigma_X\left(1-\frac{1}{n}\right)\right\|_\infty\geq t\right)$$

$$\leq\mathbb{P}\left(\left\|\frac{(\mathbf{X}^\top\mathbf{A}^\top\mathbf{A}\mathbf{X})}{n^2p}-\Sigma_X\left(1-\frac{1}{n}\right)\right\|_\infty\geq t\mid\mathbf{A}\in\Omega_{n,1}\cap\Omega_{n,2}\right)+\mathbb{P}(\mathbf{A}\in(\Omega_{n,1}\cap\Omega_{n,2})^c)$$

$$\leq 2\exp\left(2\log d-c'\min\left(\frac{n^2t^2}{n+n^2p^2},\frac{t}{p}\right)\right)+e^{\log n-\frac{np}{c}}+\frac{1}{n}\,.$$

The last step comes from Hanson Wright inequality.

Therefore, choosing

$$t=K\max\left\{\sqrt{\frac{\log d}{n}+p^2\log d},p\log d\right\}$$

we conclude:

$$\boxed{\left\|\frac{(\mathbf{X}^\top\mathbf{A}^\top\mathbf{A}\mathbf{X})}{n^2p}-\Sigma_X\left(1-\frac{1}{n}\right)\right\|_\infty\leq K\max\left\{\sqrt{\frac{\log d}{n}+p^2\log d},p\log d\right\}\leq K\max\left\{\sqrt{\frac{\log d}{n}},p\log d\right\}\,.}$$

which means

$$\left\|\frac{\mathbf{X}^\top\mathbf{A}^\top\mathbf{A}\mathbf{X}}{n^2p}-\Sigma_X\right\|_\infty\leq K\max\left\{\sqrt{\frac{\log d}{n}},p\log d\right\}\triangleq\epsilon_n\,.$$

with probability $\geq 1-d^{-\tilde{K}}-n^{-1}-e^{\log n-\frac{np}{c}}$ for some constant $c$ and $\tilde{K}$.

Using this, we have:

$$\frac{\mathbf{v}^\top\mathbf{Z}^\top\mathbf{Z}\mathbf{v}}{n\|\mathbf{v}\|^2}=\frac{\mathbf{v}^\top\Sigma_X\mathbf{v}}{\|\mathbf{v}\|^2}+\frac{\mathbf{v}^\top(\mathbf{Z}^\top\mathbf{Z}/n-\Sigma_Z)\mathbf{v}}{\|\mathbf{v}\|^2}$$

$$\leq\lambda_{\max}(\Sigma_X)+\left\|\frac{\mathbf{Z}^\top\mathbf{Z}}{n}-\Sigma_X\right\|_\infty\frac{\|\mathbf{v}\|_1^2}{\|\mathbf{v}\|_2^2}$$

$$\leq\lambda_{\max}(\Sigma_X)+\epsilon_n\frac{(1+\kappa)^2s\|\mathbf{v}\|_2^2}{\|\mathbf{v}\|_2^2}$$

$$\leq \lambda_{\max}(\Sigma_X) + (1 + \kappa)^2 s\epsilon_n.$$

Here the penultimate inequality follows from the fact:

$$\|\mathbf{v}\|_1 = \|\mathbf{v}_S\|_1 + \|\mathbf{v}_{S^c}\|_1 \leq (1 + k)\|\mathbf{v}_S\|_1 \leq (1 + k)\sqrt{s}\|\mathbf{v}_S\|_2.$$

Hence as soon as we assume $s\epsilon_n$ is bounded or goes to 0 we are good.

### E.5 PROOF OF THEOREM E.10

We follow the proof of Theorem 1 in Tian & Feng (2023) and extend and modify the results to allow for network dependency. Notice that we assume both source and target domains share the same ER graph probability, denoted as $p$.

**Step 1:**

Step 1 aims to solve the following equation w.r.t. $\boldsymbol{\beta} \in \mathbb{R}^d$ :

$$\sum_{k\in\{0,\mathcal{A}\}} \left[ \left(\mathbf{Z}^{(k)}\right)^T \mathbf{Y}^{(k)} - \sum_{i=1}^{n_k} \psi'\left(\boldsymbol{\beta}^T \mathbf{Z}_i^{(k)}\right) \mathbf{Z}_i^{(k)} \right] = \mathbf{0}_p$$

converging to its population version's solution under certain conditions with $\alpha_k = \frac{n_k}{n_\mathcal{A}+n_0}$ :

$$\sum_{k\in\{0,\mathcal{A}\}} \alpha_k \mathbb{E}\left\{ \left[ \psi'\left(\left(\boldsymbol{\beta}^\mathcal{A}\right)^T \mathbf{Z}^{(k)}\right) - \psi'\left(\left(\boldsymbol{\beta}^{(k)}\right)^T \mathbf{Z}^{(k)}\right) \right] \mathbf{Z}^{(k)} \right\} = \mathbf{0}_p \tag{E.1}$$

As we define before, $\hat{\boldsymbol{u}}^\mathcal{A} = \hat{\boldsymbol{\beta}}^\mathcal{A} - \boldsymbol{\beta}^\mathcal{A}$ and $\mathcal{D} = \left\{ \left(\mathbf{Z}^{(k)}, \mathbf{Y}^{(k)}\right) \right\}_{k\in\{0,\mathcal{A}\}}$. Firstly, we claim that when $\lambda_{\boldsymbol{\beta}} \geq 2\left\|\nabla L\left(\boldsymbol{\beta}^\mathcal{A}, \mathcal{D}\right)\right\|_\infty$, it holds that with probability of at least $1 - \left(\exp\left(\frac{1}{2}\log d + 1 - c_4\log^2 d\Psi(p)\right) + 2\exp(1 - (c_1 - 3/2)\log d) + 2\exp\left(\frac{3}{2}\log d + 1 - c_2 n\right)\right)$ that

$$\left\|\hat{\boldsymbol{u}}^\mathcal{A}\right\|_2 \leq 8\frac{C_4}{\kappa_l}C_l h\log d\sqrt{\frac{\Psi(p)}{n_\mathcal{A}+n_0}} + 3\frac{\sqrt{s}}{\kappa_1}\lambda_\omega + 2\sqrt{\frac{C_l}{\kappa_1}h\lambda_\omega} \tag{E.2}$$

According to the definition of $\hat{\omega}^\mathcal{A}$, Hölder inequality and Lemma 1, we will have

$$\begin{aligned}
\delta\hat{L}\left(\hat{\boldsymbol{u}}^\mathcal{A}, \mathcal{D}\right) &\leq \lambda_{\boldsymbol{\beta}}\left(\left\|\boldsymbol{\beta}_S^\mathcal{A}\right\|_1 + \left\|\boldsymbol{\beta}_{S^c}^\mathcal{A}\right\|_1\right) - \lambda_{\boldsymbol{\beta}}\left(\left\|\hat{\boldsymbol{\beta}}_S^\mathcal{A}\right\|_1 + \left\|\hat{\boldsymbol{\beta}}_{S^c}^\mathcal{A}\right\|_1\right) + \nabla\hat{L}(\boldsymbol{\beta}^\mathcal{A}, \mathcal{D})^T\hat{\boldsymbol{u}}^\mathcal{A} \\
&\leq \lambda_{\boldsymbol{\beta}}\left(\left\|\boldsymbol{\beta}_S^\mathcal{A}\right\|_1 + \left\|\boldsymbol{\beta}_{S^c}^\mathcal{A}\right\|_1\right) - \lambda_{\boldsymbol{\beta}}\left(\left\|\hat{\boldsymbol{\beta}}_S^\mathcal{A}\right\|_1 + \left\|\hat{\boldsymbol{\beta}}_{S^c}^\mathcal{A}\right\|_1\right) + \frac{1}{2}\lambda_{\boldsymbol{\beta}}\left\|\hat{\boldsymbol{u}}^\mathcal{A}\right\|_1 \\
&\leq \frac{3}{2}\lambda_{\boldsymbol{\beta}}\left\|\hat{\boldsymbol{u}}_S^\mathcal{A}\right\|_1 - \frac{1}{2}\lambda_{\boldsymbol{\beta}}\left\|\hat{\boldsymbol{u}}_{S^c}^\mathcal{A}\right\|_1 + 2\lambda_{\boldsymbol{\beta}}\left\|\boldsymbol{\beta}_{S^c}^\mathcal{A}\right\|_1 \\
&\leq \frac{3}{2}\lambda_{\boldsymbol{\beta}}\left\|\hat{\boldsymbol{u}}_S^\mathcal{A}\right\|_1 - \frac{1}{2}\lambda_{\boldsymbol{\beta}}\left\|\hat{\boldsymbol{u}}_{S^c}^\mathcal{A}\right\|_1 + 2\lambda_{\boldsymbol{\beta}}C_l h \tag{E.3}
\end{aligned}$$

If we assume that the claim we stated does not hold, we consider $\mathbb{C} = \left\{\boldsymbol{u} : \frac{3}{2}\|\boldsymbol{u}_S\|_1 - \frac{1}{2}\|\boldsymbol{u}_{S^c}\|_1 + 2C_l h \geq 0\right\}$. By (E.3) and the convexity of $\hat{L}$, we conclude $\hat{\boldsymbol{u}}^\mathcal{A} \in \mathbb{C}$. Then for any $t \in (0, 1)$, we can see that

$$\frac{1}{2}\left\|t\hat{\boldsymbol{u}}_{S^c}^\mathcal{A}\right\|_1 = t \cdot \frac{1}{2}\left\|\hat{\boldsymbol{u}}_{S^c}^\mathcal{A}\right\|_1 \leq t \cdot \left(\frac{3}{2}\left\|\hat{\boldsymbol{u}}_S^\mathcal{A}\right\|_1 + 2C_{\boldsymbol{\beta}}h\right) \leq \frac{3}{2}\left\|t\hat{\boldsymbol{u}}_S^\mathcal{A}\right\|_1 + 2C_l h$$

which also implies that $t\hat{u}^{\mathcal{A}} \in \mathbb{C}$. There exists certain $t$ satisfying that $\left\|t\hat{u}^{\mathcal{A}}\right\|_2 > 8\kappa_2 C_l h \sqrt{\frac{\log d}{n_{\mathcal{A}}+n_0}} + 3\frac{\sqrt{s}}{\kappa_1}\lambda_\omega + 2\sqrt{\frac{C_l}{\kappa_1}h\lambda_\omega}$ and $\left\|t\hat{u}^{\mathcal{A}}\right\|_2 \leq 1$. We denote $\tilde{u}^{\mathcal{A}} = t\hat{u}^{\mathcal{A}}$ and $F(u) = \hat{L}\left(\beta^{\mathcal{A}} + u, \mathcal{D}\right) - \hat{L}\left(\beta^{\mathcal{A}}\right) + \lambda_{\boldsymbol{\beta}}\left(\left\|\beta^{\mathcal{A}} + u\right\|_1 - \left\|\beta^{\mathcal{A}}\right\|_1\right)$. As $F(\mathbf{0}) = 0$ and $F\left(\hat{u}^{\mathcal{A}}\right) \leq 0$, by convexity, we establish

$$F\left(\tilde{u}^{\mathcal{A}}\right) = F\left(t\hat{u}^{\mathcal{A}} + (1-t)\mathbf{0}\right) \leq tF\left(\hat{u}^{\mathcal{A}}\right) \leq 0 \tag{E.4}$$

However, by Lemma E.6 and the same trick we use for (E.3),

$$F\left(\tilde{u}^{\mathcal{A}}\right) \geq \delta\hat{L}\left(\hat{u}^{\mathcal{A}}, \mathcal{D}\right) + \nabla\hat{L}\left(\beta^{\mathcal{A}}\right)^T \tilde{u}^{\mathcal{A}} - \lambda_{\boldsymbol{\beta}}\left\|\beta^{\mathcal{A}}\right\|_1 + \lambda_{\boldsymbol{\beta}}\left\|\beta^{\mathcal{A}} + \tilde{u}^{\mathcal{A}}\right\|_1$$

$$\geq L_\psi(T)\kappa_l\left\|\tilde{u}^{\mathcal{A}}\right\|_2^2 - L_\psi(T)\left(C_4 \log d\sqrt{\frac{\Psi(p)}{n_{\mathcal{A}}+n_0}}\right)\left\|\tilde{u}^{\mathcal{A}}\right\|_1\left\|\tilde{u}^{\mathcal{A}}\right\|_2$$

$$- \frac{3}{2}\lambda_{\boldsymbol{\beta}}\left\|\tilde{u}_S^{\mathcal{A}}\right\|_1 + \frac{1}{2}\lambda_{\boldsymbol{\beta}}\left\|\tilde{u}_{S^c}^{\mathcal{A}}\right\|_1 - 2\lambda_{\boldsymbol{\beta}}C_l h$$

$$\geq L_\psi(T)\kappa_l\left\|\tilde{u}^{\mathcal{A}}\right\|_2^2 - L_\psi(T)C_4 \log d\sqrt{\frac{\Psi(p)}{n_{\mathcal{A}}+n_0}}\left\|\tilde{u}^{\mathcal{A}}\right\|_1\left\|\tilde{u}^{\mathcal{A}}\right\|_2$$

$$- \frac{3}{2}\lambda_{\boldsymbol{\beta}}\left\|\tilde{u}_S^{\mathcal{A}}\right\|_1 - 2\lambda_{\boldsymbol{\beta}}C_l h$$

Due to $\tilde{u}^{\mathcal{A}} \in \mathbb{C}$, it holds that

$$\frac{1}{2}\left\|\tilde{u}^{\mathcal{A}}\right\|_1 \leq 2\left\|\tilde{u}_S^{\mathcal{A}}\right\|_1 + 2C_{\boldsymbol{\beta}}h \leq 2\sqrt{s}\left\|\tilde{u}^{\mathcal{A}}\right\|_2 + 2C_l h$$

Here, we denote $\kappa_1 = L_\psi(T)\kappa_l$, $\kappa_2 = C_4 \log d/\kappa_l$, when $n_{\mathcal{A}} + n_0 > 16\kappa_2^2 s\Psi(p)$, we have $2\frac{C_4}{\kappa_l}\log d\sqrt{\frac{s\Psi(p)}{n_{\mathcal{A}}+n_0}} \leq \frac{1}{2}$.Then it follows

$$F\left(\tilde{u}^{\mathcal{A}}\right) \geq \frac{1}{2}\kappa_1\left\|\tilde{u}^{\mathcal{A}}\right\|_2^2 - \left[2\kappa_1\kappa_2\sqrt{\frac{\Psi(p)}{n_{\mathcal{A}}+n_0}}C_{\boldsymbol{\beta}}h + \frac{3}{2}\lambda_{\boldsymbol{\beta}}\sqrt{s}\right]\left\|\tilde{u}^{\mathcal{A}}\right\|_2 - 2\lambda_{\boldsymbol{\beta}}C_l h =$$

$$\frac{1}{2}\kappa_1\left\|\tilde{u}^{\mathcal{A}}\right\|_2^2 - \left[2\kappa_1\frac{C_4}{\kappa_l}\log d\sqrt{\frac{\Psi(p)}{n_{\mathcal{A}}+n_0}}C_{\boldsymbol{\beta}}h + \frac{3}{2}\lambda_{\boldsymbol{\beta}}\sqrt{s}\right]\left\|\tilde{u}^{\mathcal{A}}\right\|_2 - 2\lambda_{\boldsymbol{\beta}}C_l h > 0$$

that conflicts with (E.4). Thus our claim at the beginning holds.

Next, we will prove $\left\|\nabla\hat{L}\left(\beta^{\mathcal{A}}\right)\right\|_\infty \lesssim \sqrt{\frac{\log d}{n_{\mathcal{A}}+n_0}}$ with probability at least $1 - (c_{12}d^{-c_{13}} + \sum_k n_k^{-1} + \sum_k e^{\log n_k - \frac{n_k p_k}{c_{14}}})$. To see this, we notice that

$$\nabla\hat{L}\left(\beta^{\mathcal{A}}\right) = \frac{1}{n_{\mathcal{A}}+n_0}\sum_{k\in\{0,\mathcal{A}\}}\left(\mathbf{Z}^{(k)}\right)^T\left[-\mathbf{Y}^{(k)} + \psi'\left(\mathbf{Z}^{(k)}\beta^{\mathcal{A}}\right)\right]$$

$$= \frac{1}{n_{\mathcal{A}}+n_0}\sum_{k\in\{0,\mathcal{A}\}}\left(\mathbf{Z}^{(k)}\right)^T\left[-\mathbf{Y}^{(k)} + \psi'\left(\mathbf{Z}^{(k)}\beta^{(k)}\right)\right]$$

$$+ \frac{1}{n_{\mathcal{A}}+n_0}\sum_{k\in\{0,\mathcal{A}\}}\left(\mathbf{Z}^{(k)}\right)^T\left[-\psi'\left(\mathbf{Z}^{(k)}\beta^{(k)}\right) + \psi'\left(\mathbf{Z}^{(k)}\beta^{\mathcal{A}}\right)\right] \tag{E.5}$$

By extending Lemma 6 of Negahban et al. (2009) for network dependency in our settings, under Assumptions E.1 and the fact $n_{\mathcal{A}} \geq Cs\log d$, we have shown in Lemma E.7 that

$$\frac{1}{n_{\mathcal{A}} + n_0} \left\| \sum_{k \in \{0, \mathcal{A}\}} \left( \mathbf{Z}^{(k)} \right)^T \left[ -\mathbf{Y}^{(k)} + \psi' \left( \mathbf{Z}^{(k)} \boldsymbol{\beta}^{(k)} \right) \right] \right\|_\infty \lesssim \sqrt{\frac{\log d}{n_{\mathcal{A}} + n_0}}$$

with probability at least $1 - (c_6 d^{-c_7} + \sum_k n_k^{-1} + \sum_k e^{\log n_k - \frac{n_k p_k}{c_8}})$.

The remaining work aims to bound the infinity norm of the second term in (E.5). We denote $U_{ij}^{(k)} = \mathbf{Z}_{ij}^{(k)} \left[ -\psi' \left( \left( \mathbf{Z}_i^{(k)} \right)^T \boldsymbol{\beta}^{(k)} \right) + \psi' \left( \left( \mathbf{Z}_i^{(k)} \right)^T \boldsymbol{\beta}^{\mathcal{A}} \right) \right]$. Under Assumption E.1, we have shown in Conjecture E.8 that:

$$\frac{1}{n_{\mathcal{A}} + n_0} \sup_{j=1,\ldots,d} \left| \sum_{k \in \{0, \mathcal{A}\}} \sum_{i=1}^{n_k} U_{ij}^{(k)} \right| \lesssim \sqrt{\frac{\log d}{n_{\mathcal{A}} + n_0}}$$

with probability at least $1 - (c_9 d^{-c_{10}} + \sum_k n_k^{-1} + \sum_k e^{\log n_k - \frac{n_k p_k}{c_{11}}})$. Hence $\left\| \nabla \hat{L} \left( \boldsymbol{\beta}^{\mathcal{A}} \right) \right\|_\infty \lesssim \sqrt{\frac{\log d}{n_{\mathcal{A}} + n_0}}$ holds with probability at least $1 - (c_{12} d^{-c_{13}} + \sum_k n_k^{-1} + \sum_k e^{\log n_k - \frac{n_k p_k}{c_{14}}})$. We plug this rate into (E.2), and get

$$\left\| \hat{\boldsymbol{u}}^{\mathcal{A}} \right\|_2 \lesssim h \log d \sqrt{\frac{\Psi(p)}{n_{\mathcal{A}} + n_0}} + \sqrt{\frac{s \log d}{n_{\mathcal{A}} + n_0}} + \left( \frac{\log d}{n_{\mathcal{A}} + n_0} \right)^{1/4} \sqrt{h} \tag{E.6}$$

with probability at least $1 - (c_{12} d^{-c_{13}} + \sum_k n_k^{-1} + \sum_k e^{\log n_k - \frac{n_k p_k}{c_{14}}})$ when $\lambda_{\boldsymbol{\beta}} \asymp C_{\boldsymbol{\beta}} \sqrt{\frac{\log d}{n_{\mathcal{A}} + n_0}}$ with $C_{\boldsymbol{\beta}} > 0$ sufficiently large. As $\hat{\boldsymbol{u}}^{\mathcal{A}} \in \mathbb{C}$, (E.6) implies

$$\left\| \hat{\boldsymbol{u}}^{\mathcal{A}} \right\|_1 \lesssim s \sqrt{\frac{\log d}{n_{\mathcal{A}} + n_0}} + \left( \frac{\log d}{n_{\mathcal{A}} + n_0} \right)^{1/4} \sqrt{sh} + h \left( 1 + \log d \sqrt{\frac{s \Psi(p)}{n_{\mathcal{A}} + n_0}} \right) \tag{E.7}$$

with probability at least $1 - (c_{12} d^{-c_{13}} + \sum_k n_k^{-1} + \sum_k e^{\log n_k - \frac{n_k p_k}{c_{14}}})$.

**Step 2:**

For our convenience, we state the notation again here: $\mathcal{D}^{(0)} = \left( \mathbf{Z}^{(0)}, \mathbf{Y}^{(0)} \right)$, $\hat{L}^{(0)} \left( \boldsymbol{\beta}, \mathcal{D}^{(0)} \right) = -\frac{1}{n_0} \left( \mathbf{Y}^{(0)} \right)^T \mathbf{Z}^{(0)} \boldsymbol{\beta} + \frac{1}{n_0} \sum_{i=1}^{n_0} \psi \left( \left( \mathbf{Z}_i^{(0)} \right)^T \boldsymbol{\beta} \right)$, $\nabla \hat{L}^{(0)} \left( \boldsymbol{\beta}, \mathcal{D}^{(0)} \right) = -\frac{1}{n_0} \left( \mathbf{Z}^{(0)} \right)^T \mathbf{Y}^{(0)} + \frac{1}{n_0} \left( \mathbf{Z}^{(0)} \right)^T \psi' \left( \mathbf{Z}^{(0)} \boldsymbol{\beta} \right)$, $\delta^{\mathcal{A}} = \boldsymbol{\beta}^{(0)} - \boldsymbol{\beta}^{\mathcal{A}}$, $\hat{\boldsymbol{\beta}}^{(0)} = \hat{\boldsymbol{\beta}}^{\mathcal{A}} + \hat{\boldsymbol{\delta}}^{\mathcal{A}}$, $\hat{\mathbf{v}}^{\mathcal{A}} = \hat{\boldsymbol{\delta}}^{\mathcal{A}} - \delta^{\mathcal{A}}$, and $\delta \hat{L}^{(0)}(\boldsymbol{\delta}, \mathcal{D}) = \hat{L}^{(0)} \left( \hat{\boldsymbol{\beta}}^{\mathcal{A}} + \boldsymbol{\delta}, \mathcal{D}^{(0)} \right) - \hat{L}^{(0)} \left( \hat{\boldsymbol{\beta}}^{\mathcal{A}} + \delta^{\mathcal{A}}, \mathcal{D}^{(0)} \right) - \nabla \hat{L}^{(0)} \left( \hat{\boldsymbol{\beta}}^{\mathcal{A}} + \delta^{\mathcal{A}}, \mathcal{D}^{(0)} \right)^T \hat{\mathbf{v}}^{\mathcal{A}}$.

Following similar derivations for (E.3), when $\lambda_\delta \geq 2 \left\| \nabla \hat{L}^{(0)} \left( \boldsymbol{\beta}^{(0)}, \mathcal{D}^{(0)} \right) \right\|_\infty$, we establish

$$
\begin{aligned}
\delta \hat{L}^{(0)} \left( \hat{\boldsymbol{\delta}}^{\mathcal{A}}, \mathcal{D} \right) \leq {} & \lambda_\delta \left( \left\| \boldsymbol{\delta}^{\mathcal{A}} \right\|_1 - \left\| \hat{\boldsymbol{\delta}}^{\mathcal{A}} \right\|_1 \right) - \nabla \hat{L}^{(0)} \left( \hat{\boldsymbol{\beta}}^{\mathcal{A}} + \delta^{\mathcal{A}}, \mathcal{D}^{(0)} \right)^T \hat{\mathbf{v}}^{\mathcal{A}} \\
\leq {} & \lambda_{\boldsymbol{\delta}} \left( 2 \left\| \boldsymbol{\delta}^{\mathcal{A}} \right\|_1 - \left\| \hat{\mathbf{v}}^{\mathcal{A}} \right\|_1 \right) + \left\| \nabla \hat{L}^{(0)} \left( \boldsymbol{\beta}^{(0)}, \mathcal{D}^{(0)} \right) \right\|_\infty \left\| \hat{\mathbf{v}}^{\mathcal{A}} \right\|_1 \\
& - \left[ \nabla \hat{L}^{(0)} \left( \hat{\boldsymbol{\beta}}^{\mathcal{A}} + \delta^{\mathcal{A}}, \mathcal{D}^{(0)} \right) - \nabla \hat{L}^{(0)} \left( \boldsymbol{\beta}^{(0)}, \mathcal{D}^{(0)} \right) \right]^T \hat{\mathbf{v}}^{\mathcal{A}} \\
\leq {} & 2 \lambda_\delta \left\| \boldsymbol{\delta}^{\mathcal{A}} \right\|_1 - \frac{1}{2} \lambda_\delta \left\| \hat{\mathbf{v}}^{\mathcal{A}} \right\|_1 \\
& - \frac{1}{n_0} \left[ \psi' \left( \left( \mathbf{Z}^{(0)} \right)^T \left( \hat{\boldsymbol{\beta}}^{\mathcal{A}} + \delta^{\mathcal{A}} \right) \right) - \psi' \left( \left( \mathbf{Z}^{(0)} \right)^T \boldsymbol{\beta}^{(0)} \right) \right]^T \hat{\mathbf{v}}^{\mathcal{A}}
\end{aligned}
$$

$$\leq 2\lambda_\delta \left\| \boldsymbol{\delta}^{\mathcal{A}} \right\|_1 - \frac{1}{2}\lambda_\delta \left\| \hat{\mathbf{v}}^{\mathcal{A}} \right\|_1 + \frac{1}{4c_0}M_\psi^2 \cdot \frac{1}{n_0} \left\| \mathbf{Z}^{(0)}\hat{\boldsymbol{u}}^{\mathcal{A}} \right\|_2^2$$

$$+ c_0 \cdot \frac{1}{n_0} \left\| \mathbf{Z}^{(0)}\hat{\mathbf{v}}^{\mathcal{A}} \right\|_2^2 \tag{E.8}$$

with $c_0 > 0$ being a constant that is enough small. The last inequality holds according to:

$$-\frac{1}{n_0} \left[ \boldsymbol{\psi}' \left( \left( \mathbf{Z}^{(0)} \right)^T \left( \hat{\boldsymbol{\beta}}^{\mathcal{A}} + \boldsymbol{\delta}^{\mathcal{A}} \right) \right) - \boldsymbol{\psi}' \left( \left( \mathbf{Z}^{(0)} \right)^T \boldsymbol{\beta}^{(0)} \right) \right]^T \hat{\mathbf{v}}^{\mathcal{A}}$$

$$= \frac{1}{n_0} \left( \hat{\boldsymbol{u}}^{\mathcal{A}} \right)^T \left( \mathbf{Z}^{(0)} \right)^T \Lambda^{(0)} \mathbf{Z}^{(0)} \hat{\mathbf{v}}^{\mathcal{A}}$$

$$\leq \frac{1}{4c_0} M_\psi^2 \cdot \frac{1}{n_0} \left\| \mathbf{Z}^{(0)}\hat{\boldsymbol{u}}^{\mathcal{A}} \right\|_2^2 + c_0 \cdot \frac{1}{n_0} \left\| \mathbf{Z}^{(0)}\hat{\mathbf{v}}^{\mathcal{A}} \right\|_2^2$$

where $\Lambda^{(0)} = \text{diag}\left( \left\{ \psi'' \left( \left( \mathbf{Z}_i^{(0)} \right)^T \boldsymbol{\beta}^{(0)} + t_i \left( \mathbf{Z}_i^{(0)} \right)^T \hat{\boldsymbol{u}}^{\mathcal{A}} \right) \right\}_{i=1}^{n_0} \right)$ is a $n_0 \times n_0$ diagonal matrix and $\left\| \Lambda^{(0)} \right\|_{\max} \leq M_\psi$.

We denote $\tilde{\mathbf{v}}^{\mathcal{A}} = t\hat{\mathbf{v}}^{\mathcal{A}}$ and similar to what we defined before, let $F^{(0)}(\mathbf{v}) = \hat{L}^{(0)}\left( \hat{\boldsymbol{\beta}}^{\mathcal{A}} + \boldsymbol{\delta}^{\mathcal{A}} + \mathbf{v}, \mathcal{D}^{(0)} \right) - \hat{L}^{(0)}\left( \hat{\boldsymbol{\beta}}^{\mathcal{A}} + \boldsymbol{\delta}^{\mathcal{A}}, \mathcal{D}^{(0)} \right) + \lambda_\delta \left( \left\| \boldsymbol{\delta}^{\mathcal{A}} + \mathbf{v} \right\|_1 - \left\| \boldsymbol{\delta}^{\mathcal{A}} \right\|_1 \right)$. As $F(\mathbf{0}) = 0$ and $F^{(0)}\left( \hat{\mathbf{v}}^{\mathcal{A}} \right) \leq 0$, by convexity, for any $t \in (0, 1]$, we establish

$$F^{(0)}\left( \tilde{\mathbf{v}}_{\mathcal{A}} \right) = F^{(0)}\left( t\hat{\mathbf{v}}_{\mathcal{A}} + (1-t)\mathbf{0} \right) \leq tF^{(0)}\left( \hat{\boldsymbol{u}}^{\mathcal{A}} \right) \leq 0 \tag{E.9}$$

Setting $t \in (0, 1]$ ensures that $\left\| \tilde{\mathbf{v}}^{\mathcal{A}} \right\|_2 \leq 1$. By noticing the fact that $\left\| \tilde{\mathbf{v}}^{\mathcal{A}} \right\|_2 \leq \left\| \tilde{\mathbf{v}}^{\mathcal{A}} \right\|_1$, we can apply Lemma E.6 on $\tilde{\mathbf{v}}^{\mathcal{A}}$ with minor modifications. Also by (E.9) and (E.8), we establish:

$$\kappa_1 \left\| \tilde{\mathbf{v}}^{\mathcal{A}} \right\|_2^2 - \kappa_1\kappa_3 \left( \log d \sqrt{\frac{\Psi(p)}{n_0}} \right) \cdot \left\| \tilde{\mathbf{v}}^{\mathcal{A}} \right\|_1^2 \leq$$

$$F^{(0)}\left( \tilde{\mathbf{v}}^{\mathcal{A}} \right) - \nabla \hat{L}^{(0)}\left( \hat{\boldsymbol{\beta}}^{\mathcal{A}} + \boldsymbol{\delta}^{\mathcal{A}}, \mathcal{D}^{(0)} \right)^T \tilde{\mathbf{v}}^{\mathcal{A}} \leq 2\lambda_\delta \left\| \boldsymbol{\delta}^{\mathcal{A}} \right\|_1 - \frac{1}{2}\lambda_\delta \left\| \tilde{\mathbf{v}}^{\mathcal{A}} \right\|_1 +$$

$$\frac{1}{4c_0}M_\psi^2 \cdot \frac{1}{n_0} \left\| \mathbf{Z}^{(0)}\hat{\boldsymbol{u}}^{\mathcal{A}} \right\|_2^2 + c_0 \cdot \frac{1}{n_0} \left\| \mathbf{Z}^{(0)}\tilde{\mathbf{v}}^{\mathcal{A}} \right\|_2^2 \tag{E.10}$$

with $\kappa_1 = L_\psi(T)\kappa_l$, $\kappa_3 = C_4/\kappa_l$.

We showed in the proof of Step 1 that $\left\| \hat{\boldsymbol{u}}_{S^c}^{\mathcal{A}} \right\|_1 \leq 3 \left\| \hat{\boldsymbol{u}}_S^{\mathcal{A}} \right\|_1 + 4C_l h$. Next we discuss about bounding $\frac{1}{n_0} \left\| \mathbf{Z}^{(0)}\hat{\boldsymbol{u}}^{\mathcal{A}} \right\|_2^2$ by $\left\| \hat{\boldsymbol{u}}^{\mathcal{A}} \right\|_2^2$ using this fact.

If $3 \left\| \hat{\boldsymbol{u}}_S^{\mathcal{A}} \right\|_1 \geq 4C_l h$, then $\left\| \hat{\boldsymbol{u}}_{S^c}^{\mathcal{A}} \right\|_1 \leq 6 \left\| \hat{\boldsymbol{u}}_S^{\mathcal{A}} \right\|_1$. Then by Lemma E.9 (the extension on Theorem 1.6 of Zhou (2009) for network dependency), we have

$$\frac{1}{n_0} \left\| \mathbf{Z}^{(0)}\hat{\boldsymbol{u}}^{\mathcal{A}} \right\|_2^2 \lesssim \left\| \hat{\boldsymbol{u}}^{\mathcal{A}} \right\|_2^2 \lesssim \frac{s \log d}{n_{\mathcal{A}} + n_0} + h \cdot \sqrt{\frac{\log d}{n_{\mathcal{A}} + n_0}} \tag{E.11}$$

with probability at least $1 - d^{-\tilde{K}} - n_0^{-1} - e^{\log n_0 - \frac{n_0 p}{c}}$.

If $3 \left\| \hat{\boldsymbol{u}}_S^{\mathcal{A}} \right\|_1 < 4C_l h$, then $\left\| \hat{\boldsymbol{u}}_{S^c}^{\mathcal{A}} \right\|_1 \leq 8C_l h \leq \sqrt{s}$. Also $\left\| \hat{\boldsymbol{u}}^{\mathcal{A}} \right\|_2 \leq 1$ with probability $1 - (c_{12} d^{-c_{13}} + \sum_k n_k^{-1} + \sum_k e^{\log n_k - \frac{n_k p_k}{c_{14}}})$. We denote

$$\Pi_0(s) = \{\boldsymbol{u} \in \mathbb{R}^p : \|\boldsymbol{u}\|_2 \leq 1, \|\boldsymbol{u}\|_0 \leq s\}$$
$$\Pi_1(s) = \left\{\boldsymbol{u} \in \mathbb{R}^p : \|\boldsymbol{u}\|_2 \leq 1, \|\boldsymbol{u}\|_1 \leq \sqrt{s}\right\}$$

Due to Lemma 3.1 of Plan & Vershynin (2013), $\Pi_1(s) \subseteq 2\overline{\mathrm{conv}}\,(\Pi_0(s))$, where $\overline{\mathrm{conv}}\,(\Pi_0(s))$ is the closure of convex hull of $\Pi_0(s)$. Similarly, an extension for network dependency on the proof of Theorem 2.4 in Mendelson et al. (2008) will also conclude (E.11).

Next we bound $\frac{1}{n_0}\left\|\mathbf{Z}^{(0)}\tilde{\mathbf{v}}^{\mathcal{A}}\right\|_2^2$ by $\|\tilde{\mathbf{v}}^{\mathcal{A}}\|_2^2$. From basic inequality, we establish

$$
\begin{aligned}
0 &\leq \hat{L}^{(0)}\left(\hat{\boldsymbol{\beta}}^{(0)}, \mathcal{D}^{(0)}\right) - \hat{L}^{(0)}\left(\boldsymbol{\beta}^{(0)}, \mathcal{D}^{(0)}\right) - \nabla \hat{L}^{(0)}\left(\boldsymbol{\beta}^{(0)}, \mathcal{D}^{(0)}\right)^T (\hat{\boldsymbol{\beta}}^{(0)} - \boldsymbol{\beta}^{(0)}) \\
&\leq \lambda_\delta \left(\left\|\boldsymbol{\beta}^{(0)} - \hat{\boldsymbol{\beta}}^{\mathcal{A}}\right\|_1 - \left\|\hat{\boldsymbol{\delta}}^{\mathcal{A}}\right\|_1\right) + \left\|\nabla \hat{L}^{(0)}\left(\boldsymbol{\beta}^{(0)}, \mathcal{D}^{(0)}\right)\right\|_\infty \|\hat{\boldsymbol{\beta}}^{(0)} - \boldsymbol{\beta}^{(0)}\|_1 \\
&\leq \lambda_\delta \left(\left\|\boldsymbol{\beta}^{(0)} - \hat{\boldsymbol{\beta}}^{\mathcal{A}}\right\|_1 - \left\|\hat{\boldsymbol{\delta}}^{\mathcal{A}}\right\|_1\right) + \frac{1}{2}\lambda_\delta \|\hat{\boldsymbol{\beta}}^{(0)} - \boldsymbol{\beta}^{(0)}\|_1 \\
&\leq \frac{3}{2}\lambda_\delta \left\|\boldsymbol{\beta}^{(0)} - \hat{\boldsymbol{\beta}}^{\mathcal{A}}\right\|_1 - \frac{1}{2}\lambda_\delta \left\|\hat{\boldsymbol{\delta}}^{\mathcal{A}}\right\|_1 \\
&\leq \frac{3}{2}\lambda_\delta C_l h + \frac{3}{2}\lambda_\delta \left\|\hat{\boldsymbol{u}}^{\mathcal{A}}\right\|_1 - \frac{1}{2}\lambda_\delta \left\|\hat{\boldsymbol{\delta}}^{\mathcal{A}}\right\|_1
\end{aligned}
\tag{E.12}
$$

implying

$$\left\|\hat{\mathbf{v}}^{\mathcal{A}}\right\|_1 \leq \left\|\hat{\boldsymbol{\delta}}^{\mathcal{A}}\right\|_1 + C_l h \leq 3\left\|\hat{\boldsymbol{u}}^{\mathcal{A}}\right\|_1 + 4C_l h$$

Combined with results by (E.7), we have $\left\|\tilde{\mathbf{v}}^{\mathcal{A}}\right\|_1 \leq \left\|\hat{\mathbf{v}}^{\mathcal{A}}\right\|_1 \leq \sqrt{s}$ when $s\log d/\,(n_{\mathcal{A}} + n_0)$ and $h$ are small enough. We can see $\delta\hat{L}^{(0)}\left(\hat{\boldsymbol{\delta}}^{\mathcal{A}}, \mathcal{D}\right) > 0$ from the strict convexity, which leads to $\left\|\hat{\boldsymbol{\delta}}^{\mathcal{A}}\right\|_1 \leq 3\left\|\hat{\boldsymbol{u}}^{\mathcal{A}}\right\|_1 + 3h$. Then we get

$$\left\|\hat{\mathbf{v}}^{\mathcal{A}}\right\|_1 \leq \left\|\boldsymbol{\beta}^{(0)} - \hat{\boldsymbol{\beta}}^{\mathcal{A}}\right\|_1 + \left\|\hat{\boldsymbol{\delta}}^{\mathcal{A}}\right\|_1 \leq 4\left\|\hat{\boldsymbol{u}}^{\mathcal{A}}\right\|_1 + 4h \leq \sqrt{s} \tag{E.13}$$

Similar to the analysis considering $3\left\|\hat{\boldsymbol{u}}_S^{\mathcal{A}}\right\|_1 < 4C_1 h$ above, we establish

$$c_0 \cdot \frac{1}{n_0}\left\|\mathbf{Z}^{(0)}\tilde{\mathbf{v}}^{\mathcal{A}}\right\|_2^2 \leq c_0 \cdot C \left\|\tilde{\mathbf{v}}^{\mathcal{A}}\right\|_2^2$$

holds with probability at least $1 - d^{-\tilde{K}} - n_0^{-1} - e^{\log n_0 - \frac{n_0 p}{c}}$. As long as $c_0 C < c_9/2$, by (E.10), we have

$$
\begin{aligned}
&\kappa_1 \left\|\tilde{\mathbf{v}}^{\mathcal{A}}\right\|_2^2 - \kappa_1 \kappa_3 \left(\log d\sqrt{\frac{\Psi(p)}{n_0}}\right) \cdot \left\|\tilde{\mathbf{v}}^{\mathcal{A}}\right\|_1^2 \\
&\leq 2\lambda_\delta \left\|\boldsymbol{\delta}^{\mathcal{A}}\right\|_1 - \frac{1}{2}\lambda_\delta \left\|\tilde{\mathbf{v}}^{\mathcal{A}}\right\|_1 + C\frac{s\log d}{n_{\mathcal{A}} + n_0} + Ch\sqrt{\frac{\log d}{n_{\mathcal{A}} + n_0}} + c_3/2\left\|\tilde{\mathbf{v}}^{\mathcal{A}}\right\|_2^2
\end{aligned}
\tag{E.14}
$$

with probability at least $1 - C'n_0^{-1}$, $\kappa_1 = L_\psi(T)\kappa_l$, $\kappa_3 = C_4/\kappa_l$.

If satisfying $\lambda_\delta \left\|\boldsymbol{\delta}^{\mathcal{A}}\right\|_1 \leq C\frac{s\log d}{n_{\mathcal{A}}+n_0} + Ch\sqrt{\frac{\log d}{n_{\mathcal{A}}+n_0}}$, then

$$\left\|\tilde{\mathbf{v}}^{\mathcal{A}}\right\|_1 \lesssim \left[\frac{s\log d}{n_{\mathcal{A}} + n_0} + h\sqrt{\frac{\log d}{n_{\mathcal{A}} + n_0}}\right] \cdot \sqrt{\frac{1}{\log d}\sqrt{\frac{n_0}{\Psi(p)}}} + \left\|\tilde{\mathbf{v}}^{\mathcal{A}}\right\|_2^2$$

Because $\left\|\tilde{\mathbf{v}}^{\mathcal{A}}\right\|_2 \leq 1$, by (E.14), the following inequality holds

$$\left\|\tilde{\mathbf{v}}^{\mathcal{A}}\right\|_2^2 \lesssim \frac{s \log d}{n_{\mathcal{A}} + n_0} + h\sqrt{\frac{\log d}{n_{\mathcal{A}} + n_0}} \lesssim \frac{s \log d}{n_{\mathcal{A}} + n_0} + \left[h\sqrt{\frac{\log d}{n_0}}\right] \wedge h^2$$

with probability at least $1 - C'n_0^{-1}$.

If $\lambda_\delta \left\|\boldsymbol{\delta}^{\mathcal{A}}\right\|_1 > C\frac{s \log d}{n_{\mathcal{A}} + n_0} + Ch\sqrt{\frac{\log d}{n_{\mathcal{A}} + n_0}}$, then $\left\|\tilde{\mathbf{v}}^{\mathcal{A}}\right\|_1 \lesssim h + \left\|\tilde{\mathbf{v}}^{\mathcal{A}}\right\|_2^2$, leading to

$$\left\|\tilde{\mathbf{v}}^{\mathcal{A}}\right\|_2^2 \lesssim 2\lambda_\delta \left\|\boldsymbol{\delta}^{\mathcal{A}}\right\|_1 - \frac{1}{2}\lambda_\delta \left\|\tilde{\mathbf{v}}^{\mathcal{A}}\right\|_1$$

which implies $\left\|\tilde{\mathbf{v}}^{\mathcal{A}}\right\|_1 \leq 4\left\|\boldsymbol{\delta}^{\mathcal{A}}\right\|_1 \leq 4C_l h$. By plugging this result into (E.14), we obtain

$$\left\|\tilde{\mathbf{v}}^{\mathcal{A}}\right\|_2^2 \lesssim \frac{s \log d}{n_{\mathcal{A}} + n_0} + \left[h\sqrt{\frac{\log d}{n_0}}\right] \wedge h^2 \tag{E.15}$$

with probability at least $1 - C'n_0^{-1}$.

When $s \log d / (n_{\mathcal{A}} + n_0)$ and $h$ is small enough, due to $h\sqrt{\frac{\log d}{n_0}} = o(1)$, the right side of (E.15) can be very small, implying $\left\|\tilde{\mathbf{v}}^{\mathcal{A}}\right\|_2 \leq c < 1$ with probability at least $1 - C'n_0^{-1}$. We should notice that this result holds for any $t \in (0, 1]$ such that $\left\|\tilde{\mathbf{v}}^{\mathcal{A}}\right\|_2 \leq 1$. Finally let's consider the vector of interest: $\hat{\mathbf{v}}^{\mathcal{A}}$. Suppose $\left\|\tilde{\mathbf{v}}^{\mathcal{A}}\right\|_2^2 \geq \frac{s \log d}{n_{\mathcal{A}} + n_0} + \left[h\left(\sqrt{\frac{\log d}{n_0}}\right)\right] \wedge h^2$ for some constant $C > 0$ with probability at least $C'n_0^{-1}$, then there exists $t \in (0, 1]$ such that $c < \left\|\tilde{\mathbf{v}}^{\mathcal{A}}\right\|_2 \leq 1$. This contradicts with the fact $\left\|\tilde{\mathbf{v}}^{\mathcal{A}}\right\|_2 \leq c$ with probability at least $1 - C'n_0^{-1}$. Hence we establish

$$\left\|\hat{\mathbf{v}}^{\mathcal{A}}\right\|_2^2 \lesssim \frac{s \log d}{n_{\mathcal{A}} + n_0} + \left[h\sqrt{\left(\frac{\log d}{n_0}\right)}\right] \wedge h^2 \lesssim 1$$

with probability at least $1 - C'n_0^{-1}$.

Similarly, the $\ell_1$-bound on $\hat{\mathbf{v}}^{\mathcal{A}}$ will be obtained by going over the analysis procedure of $\tilde{\mathbf{v}}^{\mathcal{A}}$

$$\left\|\hat{\mathbf{v}}^{\mathcal{A}}\right\|_1 \lesssim \left[s\sqrt{\frac{\log d}{n_{\mathcal{A}} + n_0}} + h\right] \cdot \sqrt{\sqrt{\frac{1}{\Psi(p)}}}$$

with probability at least $1 - C'n_0^{-1}$.

Lastly, we combine the conclusions in this Step 2 with the upper bounds on $\left\|\hat{\boldsymbol{u}}^{\mathcal{A}}\right\|_2$ and $\left\|\hat{\boldsymbol{u}}^{\mathcal{A}}\right\|_1$ in Step 1, to complete the proof.

Combining the above inequalities, we obtain:

$$\left\|\hat{\boldsymbol{\beta}}^{(0)} - \boldsymbol{\beta}^{(0)}\right\|_2 \leq \|\hat{\boldsymbol{u}}\|_2 + \|\hat{\mathbf{v}}\|_2 \lesssim h \log d\sqrt{\frac{\Psi(p)}{n_{\mathcal{A}} + n_0}} + \sqrt{\frac{s \log d}{n_{\mathcal{A}} + n_0}} + \left(\frac{\log d}{n_{\mathcal{A}} + n_0}\right)^{1/4}\sqrt{h}.$$

And

$$\left\| \hat{\boldsymbol{\beta}}^{(0)} - \boldsymbol{\beta}^{(0)} \right\|_1 \le \|\hat{\boldsymbol{u}}\|_1 + \|\hat{\mathbf{v}}\|_1 \lesssim s\sqrt{\frac{\log d}{n_{\mathcal{A}} + n_0}} + \left(\frac{\log d}{n_{\mathcal{A}} + n_0}\right)^{1/4} \sqrt{sh} +$$

$$h\left(1 + \log d\sqrt{\frac{s\Psi(p)}{n_{\mathcal{A}} + n_0}}\right) + \left[s\sqrt{\frac{\log d}{n_{\mathcal{A}} + n_0}} + h\right] \cdot \sqrt{\sqrt{\frac{1}{\Psi(p)}}}.$$

## F ADDITIONAL EXPERIMENTAL RESULTS

In this section, we first conduct additional simulation studies considering other network models such as SBM and graphon models, and also consider multiple convolution layers $M = 2$ (see in Appendix F.1). Subsequently, we present additional results of real data analyses for transfer learning tasks, sensitivity analyses to hyperparameters, as well as the performance considering varying source and target data training rate in Appendix F.2.

### F.1 ADDITIONAL SIMULATION RESULTS

Here, we conduct simulation studies considering similar settings in Section 5.1 but generating the adjacency matrices from SBM or graphon models.

Figure S1 presents the results for SBM models. Figures S1 (a)(b) were performed when SBM generated the adjacency matrices of both target and source domains with between-community connection probability as 0.08 and the within-community probability as 0.1. Figures S1 (c)(d) were performed when the adjacency matrices of both target and source domains were generated by SBM with between-community connection probability as 0.08 and within-community probability as 0.04.

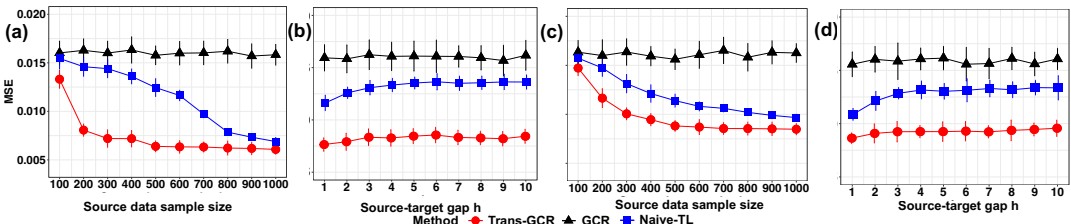

Figure S1: Performance comparison (MSE) of Trans-GCR (red), GCR (black), Naive TL (blue) across varying (a)(c) Source sample size, (b)(d) Source-target gap $h$, for two additional SBM models.

Figure S2 presents the results for graphon models. Figures S2 (a)(b), and (c)(d) were performed with the adjacency matrices of both target and source domains generated by two types of graphons, respectively.

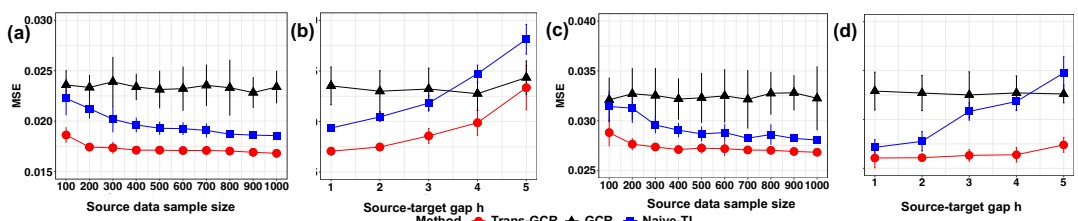

Figure S2: Performance comparison (MSE) of Trans-GCR (red), GCR (black), Naive TL (blue) across varying (a)(c) Source sample size, (b)(d) Source-target gap $h$, for two additional graphon models.

We also show the performance comparisons when we consider multiple convolution layers such as $M = 2$ in Figure S3.

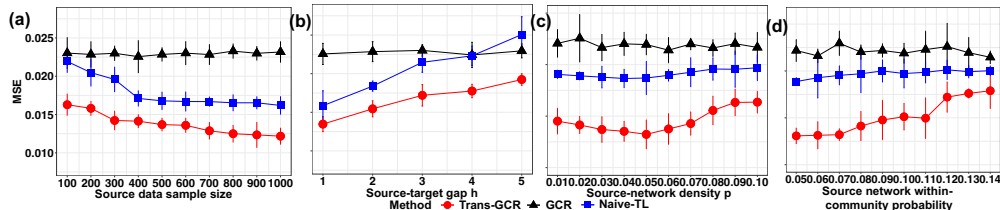

Figure S3: Performance comparison (MSE) of Trans-GCR (red), GCR (black), Naive TL (blue) across varying (a) Source sample size, (b) Source-target gap $h$, (c) Source network density (0.05 means identical densities) (d) Source network within-community probability (higher value means more discrepancy) when convolution layers $M = 2$.

We show performance comparison on averaged true positive rate (TPR) and false discovery proportion (FDP) for varying source data sample size in Table S2 as an additional metric to evaluate the estimate of the coefficients.

Table S2: Performance comparison on averaged true positive rate (TPR) and false discovery proportion (FDP) for varying source data sample size, over 10 replicates. TPRs and FDPs were calculated overall for coefficient matrices.

| Method | Metric | 100 | 200 | 300 | 400 | 500 | 600 | 700 | 800 | 900 | 1000 |
|---|---|---|---|---|---|---|---|---|---|---|---|
| Trans-GCR | TPR | 0.478 | 0.496 | 0.556 | 0.572 | 0.578 | 0.582 | 0.758 | 0.762 | 0.712 | 0.714 |
| | FDP | 0.378 | 0.342 | 0.275 | 0.263 | 0.133 | 0.123 | 0.237 | 0.134 | 0.164 | 0.112 |
| GCR | TPR | 0.072 | 0.054 | 0.032 | 0.050 | 0.072 | 0.042 | 0.050 | 0.022 | 0.066 | 0.026 |
| | FDP | 0.521 | 0.739 | 0.813 | 0.613 | 0.452 | 0.625 | 0.523 | 0.546 | 0.567 | 0.715 |
| Naive TL | TPR | 0.660 | 0.664 | 0.904 | 0.930 | 0.972 | 0.976 | 0.982 | 0.984 | 0.962 | 0.986 |
| | FDP | 0.717 | 0.715 | 0.722 | 0.717 | 0.606 | 0.677 | 0.667 | 0.668 | 0.662 | 0.669 |

## F.2 ADDITIONAL REAL DATA RESULTS

Here we present additional results from the real data analysis. Table S3 shows the averaged Micro-F1 scores for additional naive transfer learning methods as a supplement to Table 1. The performance of naive transfer learning methods when trained solely on target data is provided in Table S4. Table S5 and S6 show the results of sensitivity analyses to hyperparameter $M$ and $\lambda$. We also present the effect of source training rate and target training rate on Micro-F1 in Figure S4 for the transfer learning tasks $D \to C$, $C \to D$, and $A \to D$, in Figure S5 for the transfer learning tasks $D \to A$, $C \to A$, and $A \to C$, respectively.

We additionally consider two large-scale graphs and present in Table S7. The ogbn-arxiv dataset is a directed citation graph of Computer Science arXiv papers, where nodes represent papers and directed edges indicate citations. Each paper has a 128-dimensional feature vector derived from its title and abstract, and the task is to classify papers into 40 subject areas based on these features. Papers are also associated with publication years. We split the dataset into two subsets, including ogbn-arxiv1 (n=58,970, papers published up to 2017), and ogbn-arxiv2 (n=78,402, papers published since 2018).

Table S3: Averaged Micro F1 score (%) of additional naive transfer learning methods, over 10 replicates, with source training rate fixed at 0.75 and target training rate fixed at 0.03.

| Target | Source | node2vec | GraphSAGE | attri2vec |
|---|---|---|---|---|
| D | C | 66.55 | 69.78 | 67.64 |
| | A | 56.66 | 65.82 | 63.15 |
| | C&A | 52.09 | 56.22 | 65.75 |
| C | D | 62.34 | 69.63 | 73.33 |
| | A | 63.17 | 70.64 | 69.91 |
| | D&A | 50.93 | 60.77 | 71.85 |
| A | D | 54.36 | 62.59 | 62.92 |
| | C | 61.53 | 65.20 | 64.23 |
| | D&C | 49.27 | 57.11 | 63.42 |

Table S4: Averaged Micro F1 score (%) for comparisions with different methods, over 10 replicates, with only target training rate fixed at 0.03 (The third column is showing our proposed method's performance for transfer learning as a reference).

| Target | Source | Trans-GCR | node2vec | GraphSAGE | GCN | APPNP | attri2vec | SGC | GAT | GPRGNN | GRAND |
|---|---|---|---|---|---|---|---|---|---|---|---|
| D | C or A | **76.53 or 75.16** | 60.18 | 67.22 | 63.59 | 70.03 | 67.28 | 70.78 | 62.47 | 68.44 | 44.37 |
| C | D or A | **78.99 or 80.37** | 57.10 | 60.91 | 69.71 | 75.75 | 71.86 | 77.19 | 69.25 | 73.89 | 55.34 |
| A | D or C | **72.61 or 73.56** | 51.31 | 57.07 | 65.02 | 63.84 | 65.20 | 70.78 | 61.70 | 66.47 | 33.41 |

Table S5: Averaged Micro F1 score (%) of our proposed Trans-GCR for varying $M$, over 10 replicates, with source training rate fixed at 0.75 and target training rate fixed at 0.03.

| Target | Source | M=2 | M=3 | M=4 | **M=5** | M=6 | M=7 |
|---|---|---|---|---|---|---|---|
| D | C | 75.24 | 76.03 | 76.05 | **76.53** | 75.21 | 75.94 |
| | A | 73.14 | 74.36 | 72.03 | **75.16** | 74.79 | 76.69 |
| | C&A | 76.17 | 75.71 | 78.36 | **76.61** | 77.41 | 75.09 |
| C | D | 77.99 | 77.11 | 79.73 | **78.99** | 79.21 | 78.94 |
| | A | 76.88 | 79.07 | 80.16 | **80.37** | 79.85 | 80.32 |
| | D&A | 77.29 | 80.31 | 79.04 | **80.58** | 81.14 | 81.31 |
| A | D | 70.68 | 72.91 | 72.19 | **72.61** | 72.13 | 74.86 |
| | C | 69.42 | 73.18 | 73.17 | **73.56** | 74.49 | 74.21 |
| | D&C | 71.81 | 74.01 | 74.59 | **73.78** | 76.34 | 75.87 |

Table S6: Averaged Micro F1 score (%) of our proposed Trans-GCR for varying $\lambda$, over 10 replicates, with source training rate fixed at 0.75 and target training rate fixed at 0.03, while fixing $M = 5$.

| Target | Source | $\lambda$=0.00005 | $\lambda$=0.0001 | $\lambda$=0.0005 | $\lambda$=0.001 | $\lambda$=0.0015 | $\lambda = 0.01$ | $\lambda = 0.1$ |
|---|---|---|---|---|---|---|---|---|
| D | C | 73.19 | 75.32 | 73.76 | 76.73 | 75.12 | 59.71 | 21.57 |
| | A | 72.45 | 73.82 | 75.14 | 74.57 | 75.38 | 56.17 | 21.96 |
| | C&A | 71.32 | 74.63 | 77.08 | 74.67 | 73.82 | 58.20 | 21.47 |
| C | D | 78.26 | 79.18 | 79.96 | 78.91 | 78.68 | 65.53 | 25.62 |
| | A | 77.16 | 80.24 | 81.17 | 79.27 | 79.39 | 67.63 | 25.01 |
| | D&A | 77.64 | 80.45 | 80.85 | 80.39 | 79.98 | 65.61 | 25.47 |
| A | D | 70.69 | 67.91 | 70.64 | 72.41 | 72.97 | 58.15 | 20.33 |
| | C | 70.02 | 73.16 | 72.37 | 74.18 | 74.08 | 59.43 | 20.23 |
| | D&C | 70.44 | 72.59 | 73.82 | 73.18 | 72.01 | 58.76 | 20.34 |

Table S7: Averaged Micro F1 score (%) of various methods for additional datasets over 10 replicates, with source training rate fixed at 0.75 and target training rate fixed at 0.03. We were unable to obtain results for AdaGCN due to its high computational cost on such large datasets.

| Target | Source | Trans-GCR | GCR | AdaGCN | UDAGCN | GPRGNN | GRAND | GCN | APPNP | SGC | GAT |
|---|---|---|---|---|---|---|---|---|---|---|---|
| ogbn-arxiv2 | ogbn-arxiv1 | **62.95** | 59.76 | NA | 60.24 | 34.03 | 28.19 | 58.56 | 27.89 | 56.81 | 53.36 |
| ogbn-arxiv1 | ogbn-arxiv2 | **60.48** | 57.99 | NA | 57.58 | 27.78 | 29.96 | 54.35 | 20.61 | 47.93 | 41.33 |

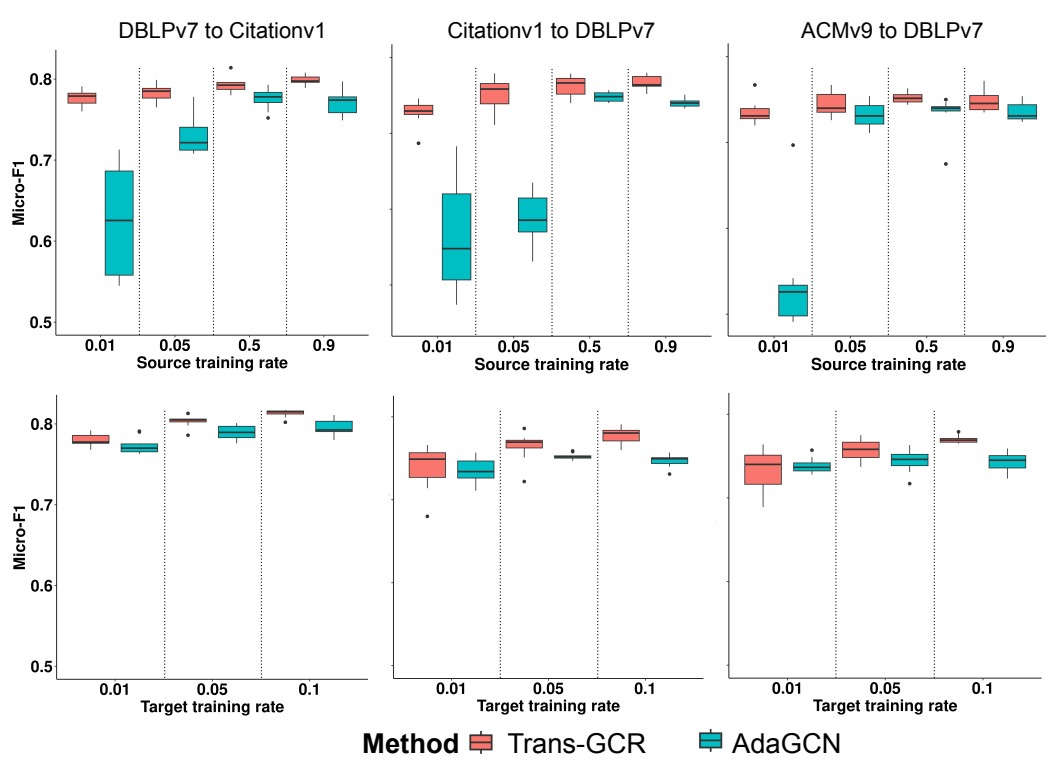

Figure S4: Multi-label classification with varying source training rates (first column, with target training rate fixed to be 0.03), with varying target training rate (second column, with source training rate fixed to be 0.75).

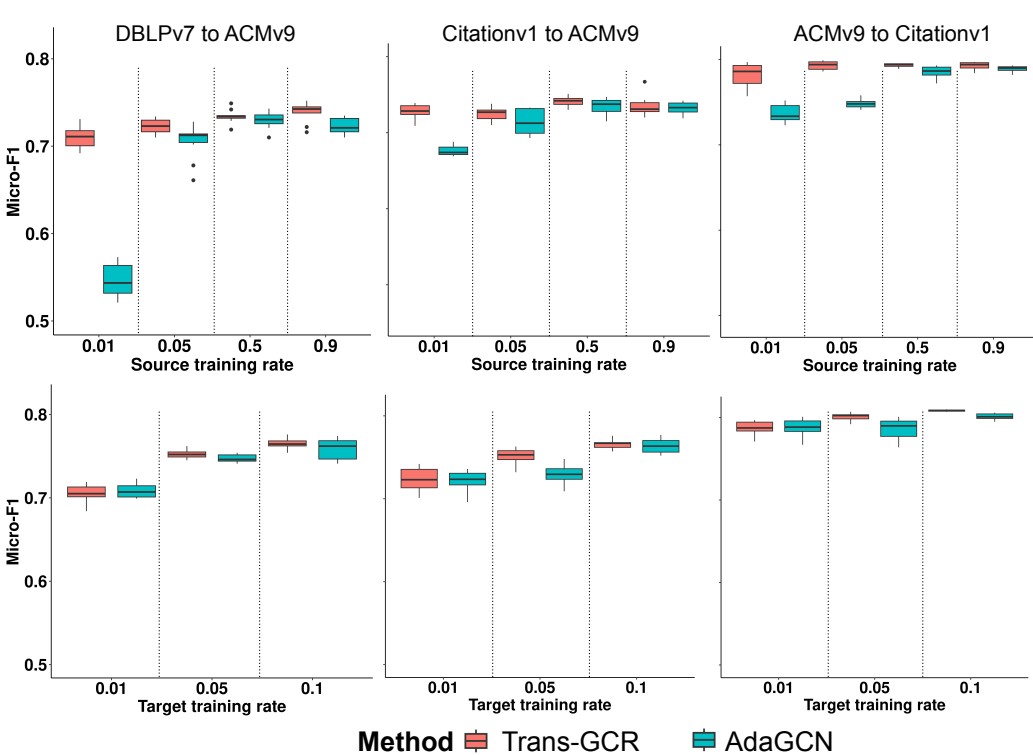

Figure S5: Multi-label classification with varying source training rates (first column, with target training rate fixed to be 0.03), with varying target training rate (second column, with source training rate fixed to be 0.75).

# G   PSEUDO CODE FOR ALGORITHMS

Here, we summarize the procedure of Trans-GCR in Algorithm 1, and the procedure of source domain selection in Algorithm 2.

---

**Algorithm 1** Trans-GCR Algorithm

---

**Require:** Target data $(\mathbf{A}^{(0)}, \mathbf{X}^{(0)}, \mathbf{Y}^{(0)})$; Source data $(\mathbf{A}^{(k)}, \mathbf{X}^{(k)}, \mathbf{Y}^{(k)}), k \in \mathcal{A}$; Hyperparameter $M$ and $\lambda$.

   Step 1. **Preprocessing.** Calculate the normalized adjacency matrix $\mathbf{S}^{(k)}$ based on $\mathbf{A}^{(k)}$, $k \in \{0, \mathcal{A}\}$.

   Step 2. **Pooled source samples.** Get pooled source sample with $\mathbf{S}^{\mathcal{A}} \in \mathbb{R}^{n_{\mathcal{A}} \times n_{\mathcal{A}}}$, $\mathbf{X}^{\mathcal{A}} \in \mathbb{R}^{n_{\mathcal{A}} \times d}$, $\mathbf{Y}^{\mathcal{A}} \in \{0,1\}^{n_{\mathcal{A}} \times C}$.

   Step 3. **Source domain parameter estimation.** Get $\hat{\boldsymbol{\beta}}^{\mathcal{A}}$ by Eq. 2.3, using the pooled source samples $(\mathbf{S}^{\mathcal{A}}, \mathbf{X}^{\mathcal{A}}, \mathbf{Y}^{\mathcal{A}})$.

   Step 4. **Domain shift estimation.** Obtain domain shift estimate $\hat{\delta}^{\mathcal{A}}$ using Eq. 3.1 and target data.

   Step 5. **Target domain parameter estimation.** Obtain $\hat{\boldsymbol{\beta}}^{(0)} = \hat{\boldsymbol{\beta}}^{\mathcal{A}} + \hat{\delta}^{\mathcal{A}}$.

   **Output**: $\hat{\boldsymbol{\beta}}^{(0)}$.

---

**Algorithm 2** Source Domain Transferability Score Calculation Algorithm

---

**Require:** Data: Target data $(\mathbf{A}^{(0)}, \mathbf{X}^{(0)}, \mathbf{Y}^{(0)})$; Source data $(\mathbf{A}^{(k)}, \mathbf{X}^{(k)}, \mathbf{Y}^{(k)}), k = 1, \ldots, K$; Hyperparameters: Number of layers $M$; Cross-validation folds $V$; Number of selected source data $L$.

   **Step 1. Target data partition.** Randomly partition data points in the target domain $\{1, \ldots, n_0\}$ into $V$ subsets of approximately equal size $s_1, \ldots, s_V$.

   **Step 2. Training and testing target data construction.** Construct testing target data $(\mathbf{A}^{(0)}, \mathbf{X}^{(0)}, \mathbf{Y}^{(0)}_{s_v})$, where we only use the label information of nodes in $s_v$. Similarly, construct training target data $(\mathbf{A}^{(0)}, \mathbf{X}^{(0)}, \mathbf{Y}^{(0)}_{-s_v})$ by excluding the label information of nodes in $s_v$.

   **Step 3. Cross-validation based score.** For $k$th source data, $k = 1, \ldots, K$, repeat the following procedure.

   **For** $v = 1, \ldots, V$,

   • **Model estimation.** Apply the transfer learning Algorithm 1 using the source data $\{\mathbf{A}^{(k)}, \mathbf{X}^{(k)}, \mathbf{Y}^{(k)}\}$ and training target data $\{\mathbf{A}^{(0)}, \mathbf{X}^{(0)}, \mathbf{Y}^{(0)}_{-s_v}\}$ to obtain the estimate $\hat{\boldsymbol{\beta}}^{(0)}_{vk}$ for the target data after transfer learning.

   • **Model evaluation** Using the learned $\hat{\boldsymbol{\beta}}^{(0)}_{vk}$ from the previous step, evaluate its prediction performance in the target domain testing data $\{\mathbf{A}^{(0)}, \mathbf{X}^{(0)}, \mathbf{Y}^{(0)}_{s_v}\}$ by calculating the negative log-likelihood value $\mathrm{NL}^{(k)}_v$ for nodes in $s_v$.

   **Averaged score over folds.** Calculate averaged negative log-likelihood over $V$ folds for each $k$ source data, $\mathrm{NL}^{(k)} = \frac{1}{V} \sum_{v=1}^{V} \mathrm{NL}^{(k)}_v$

   **Step 4. Selection.** Rank the $K$ sources according to $\mathrm{NL}^{(k)}$ and select among the top $L$ lowest sources as $\hat{\mathcal{A}}$.

   **Output**: $\hat{\mathcal{A}}$ and Transferability score $\mathrm{NL}^{(k)}, k = 1, \ldots, K$.

---

