# OpenReview forum: "Transfer Learning Under High-Dimensional Graph Convolutional Regression Model for Node Classification"
_ICLR.cc/2025/Conference — ICLR 2025 Conference Withdrawn Submission_

### Official Review · Reviewer_wstE · 2024-10-28

**Soundness:** 3
**Presentation:** 3
**Contribution:** 3
**Rating:** 6
**Confidence:** 3

**Summary:**

This paper proposed a novel transfer learning approaches to tackle transfer learning on graph domain data. Specifically, according to the SGC in 2019, the author remove the non-linearity and reformulate the GCN to be GRC model which first aggregate the graph features using symmetric normalized adjacency and then feed it into multinomial logistic lasso regression model assuming a linear relation between graph features and labels. Loss is a commonly used L1 regularized negative log likelihood loss for sparse coefficient learning. For input, pooling is used to combine source and target domain data for training. The training stage involves estimation of source domain coefficient and estimation of domain shift. Finally, estimation of target domain coefficient is learned through the addition of source domain coefficient and domain shift. Additionally, the paper provides a simple way to evaluate a score for each source domain dataset to select dataset that are close related to the target domain data.
For theoretical analysis, the author provides a theoretical analysis on the gap between the estimation and the true target coefficient with several assumptions.

**Strengths:**

1. Overall the presentation and logic is clear and sound. I find most part easy to understand and follow.
2. The model removes non-linearity and therefore is generally efficient and computationally inexpensive, it is more like a simple machine learning model than a generally deep learning approach.
3. The theoretical analysis provides a good estimate on how well the model can achieve given the number of nodes, the dimensionality of the feature, and the sparsity of the learned coefficient. Assuming the theorem is proper suitable for common cases, the high dimensionality problem seems to be alleviated with log terms.
4. The paper provides experiments on both simulated data and real world dataset.

**Weaknesses:**

1. The paper mentions several previous transfer learning approaches in GNN in the introduction, However when compared in the baseline in real datasets, only two adaptive based approach has been used and the rest is naive transfer learning with different GNN models. As the problem is focused on transfer learning scheme, It therefore doesn't seem to be convincing that the approach is fairly compared with existing STOA.
2. In the experiment, the hyperparameter in the proposed method is selected through cross validation, but the other methods are fixed with the same hyperparameter settings. For many GNN methods, performance is sensitive to the selection of the hyperparameters, the compared result is therefore not a rather fair comparison from my understanding. When checking on the sensitive analysis on the lambda and M, it is shown that the results in the proposed model have moderate fluctuations with different values.

**Questions:**

1. For the theoretical analysis, how do we obtain the assumption 4.3 (sparsity), is it assumed directly from the conclusions and why is it equal to O(1)?
2. If the author could provide a comparison results with baselines and the averaged results of the proposed results for different hyperparameter settings with some more recent transfer learning baselines on GCN, I think the experimental results will be much more convincing.

---

> ### Author Response · Authors · 2024-11-24
> **Response to reviewer wstE: 1/2**
>
> We thank the reviewer for the dedicated and insightful review. Please see below for our response.
>
> >  **Comment:** (1) Comparison with more transfer learning methods. (2) Hyperparameter tuning for each method.
>
> **Response:** Thank you for your great suggestions. In response, we have made the following improvements to our manuscript: (1) We have added an additional transfer learning approach to our experiments: the Pretrained GNN method as described in [1].  This method involves pretraining a Graph Neural Network on the source domain and then fine-tuning it on the target domain, serving as a strong baseline in graph transfer learning tasks. (2) We have employed a cross-validation strategy for each method to select the optimal hyperparameters. Specifically, we performed grid searches over key hyperparameters such as learning rates, regularization parameters, and the number of layers, using a validation set to assess performance.
>
> The updated results, incorporating the additional method and refined hyperparameter tuning, are presented in Table R4 below. As shown in Table R4, our proposed method, Trans-GCR, consistently outperforms the added Pretrained GNN method and other baseline models. This demonstrates the superior transfer learning capability of our approach in leveraging information from the source domain to improve performance on the target domain.  In the updated manuscript, we have updated Table 1 to include these results.
>
>
> Table R4:  Averaged Micro F1 score (\%) of various methods, over 10 replicates,  with source training rate fixed at 0.75 and target training rate fixed at  0.03.
>
> | Target             | Source | Trans-GCR | GCR            | AdaGCN | UDAGCN | Pre-trained GNNs | GPRGNN | GRAND | GCN   | APPNP | SGC   | GAT   |
> |--------------------|--------|---------------------|----------------|--------|--------|------------------|--------|-------|-------|-------|-------|-------|
> | D | C      | **76.53**    | 72.30          | 75.14  | 69.52  | 73.15            | 74.48  | 67.23 | 71.59 | 73.10 | 72.12 | 71.74 |
> |                    | A      | **75.16**      | 69.75          | 74.52  | 58.24  | 70.03            | 72.07  | 66.78 | 69.12 | 70.82 | 68.64 | 67.34 |
> |                    | C&A                   | **76.61** | 70.09  | 74.87  | 71.15            | 72.89  | 75.11 | 69.33 | 64.35 | 70.94 | 71.31 | 62.48 |
> | C | D      | **78.99**      | 72.82          | 77.85  | 61.63  | 75.32            | 75.36  | 67.13 | 72.97 | 75.26 | 77.56 | 73.17 |
> |                    | A      | **80.37**     | 77.16          | 79.29  | 71.85  | 67.48            | 75.02  | 73.24 | 73.85 | 75.86 | 76.73 | 73.39 |
> |                    | D& A                   | **80.58** | 77.23  | 78.91  | 73.35            | 76.21  | 75.72 | 69.67 | 70.53 | 74.81 | 77.31 | 70.52 |
> | A | D      | **72.61**      | 69.54          | 72.35  | 53.35  | 66.78            | 71.67  | 66.63 | 66.87 | 66.56 | 69.26 | 66.67 |
> |                    | C      | **73.56**    | 71.17          | 73.32  | 55.52  | 68.94            | 72.95  | 56.74 | 66.10 | 67.33 | 72.52 | 67.79 |
> |                    | D&C                   | **73.78** | 71.32  | 73.26  | 65.85            | 69.61  | 72.91 | 67.31 | 63.19 | 64.81 | 70.18 | 63.80 |
>
>
> [1] W Hu, B Liu, J Gomes, M Zitnik, P Liang, V Pande, and J Leskovec. Strategies for pre-training graph neural networks. In International Conference on Learning Representations (ICLR), 2020.

---

> > ### Author Response · Authors · 2024-11-24
> > **Response to reviewer wstE: 2/2**
> >
> > >  **Comment:** For the theoretical analysis, how do we obtain the assumption 4.3 (sparsity), is it assumed directly from the conclusions and why is it equal to O(1)?
> >
> > **Response:**  Thank you for your comment. We appreciate the opportunity to clarify how this assumption is obtained and its significance in our work.
> >
> > Detailed response about Assumption 4.3 can be found in the general response. In summary, Assumption 4.3 states that the true data-generating vector, $\beta^{(0)}$, is $s$-sparse, meaning that only $s$ coordinates of $\beta^{(0)}$ are non-zero. This assumption is not derived from our conclusions but is a foundational premise necessary for our theoretical framework, especially in the context of ultra-high-dimensional data, where the dimension of $\beta^{(0)}$, denoted as $d$, is much larger than the sample size $n$.
> >
> > In high-dimensional settings, sparsity is necessary for consistent estimation of the underlying parameter; without it, the problem becomes unidentifiable due to the "curse of dimensionality". In other words, when the number of potential covariates far exceeds the sample size, it is standard to assume that only a small subset of covariates are active, and the goal is to identify and estimate these active covariates.  The $o(1)$ denotes a term that converges to zero as $n \to \infty$, i.e., we require $s\frac{\log^2{d}}{n}\frac{(1-2p)}{4p\log{\left((1-p)/p\right)}} \to 0$, as $n  \to \infty$, where $s$ is the sparsity level.  This condition ensures that the impact of high dimensionality and network dependencies diminishes relative to the sample size, allowing the estimator to consistently recover the true parameter vector. Sparstiy assumption is a standard practice in high-dimensional statistical analysis and is essential for deriving meaningful theoretical guarantees.
> >
> >
> > More importantly, our method does not require prior knowledge of the sparsity level $s$. It can be applied regardless of the sparsity of $\beta^{(0)}$  and will produce an estimator. However, the theoretical guarantees (such as consistency and convergence rates) are only established under the sparsity assumption (Assumption 4.3).

---

> > > ### Comment · Reviewer_wstE · 2024-11-25
> > >
> > > Thank you for your detailed response. I think overall the paper is ok with the experiments added and I will keep my current score.

---

> > > > ### Author Response · Authors · 2024-11-26
> > > > **Response to Reviewer wstE**
> > > >
> > > > We are glad that the additional experiments addressed your concerns. Please feel free to let us know if you have any further questions or suggestions. Thank you for your time!

---

### Official Review · Reviewer_kBsC · 2024-11-02

**Soundness:** 2
**Presentation:** 2
**Contribution:** 2
**Rating:** 3
**Confidence:** 5

**Summary:**

This paper studies transfer learning for node classification using a so called Graph Convolutional
Multinomial Logistic Lasso Regression (GCR). Experiments on limited datasets are conducted to show the performance of GCR.

**Strengths:**

1. The presentation of the proposed method is clear.

2. Assumptions based on which GCR and its theoretical results are derived are presented clear.

**Weaknesses:**

There are several major concerns for this paper.

1. GCR is based on an indeed very strong assumption, that is, there is a linear relationship between aggregated features and labels. There is an obvious concern that features clearly depends on the GCN architecture and its training process, and it is risky to assume that there is linear relationship between the features obtained by a particular GCN architecture and its training process without clear theoretical or empirical study.

2. The assumptions 4.1-4.3 for the theoretical results in Section 4 are particularly restrictive and some of them can hardly hold in practice. For example, Assumption 4.1 needs to the node attributes to follow sub-gaussian distribution, which are often not the case in real data including the real data used by this paper in the experiments. For another example, when can the sparsity parameter $s$ satisfy the particular condition in line 227-228? Furthermore, how sharp is the bound in Theorem 4.4, and how does it compare to the literature? Without a clear comparison to prior art, the significance of Theorem 4.4 is unclear.

3. Experiments are very limited, and the real graph data used in this paper are all small graphs. Experiments on graphs of much larger scale are expected to justify the effectiveness of GCR.

**Questions:**

See weaknesses.

---

> ### Author Response · Authors · 2024-11-24
> **Response to reviewer kBsC: 1/3**
>
> We thank the reviewer for the dedicated and insightful review. Please see below for our response.
>
> >  **Comment:** GCR is based on an indeed very strong assumption, that is, there is a linear relationship between aggregated features and labels.
>
>
> **Response:**  Thank you for your comment.  We agree with you that linear assumption is a restriction. In line "513-515" in the original manuscript, we have acknowledged  that one limitation of our paper is restricted to linear relationship between aggregated features and labels.  However, as the statistician George Box famously stated, **"All models are wrong, but some are useful."** While nonlinear models are indeed more general and capable of capturing complex relationships, linear models offer simplicity, interpretability, and computational efficiency, which are valuable in many practical applications.
>
> The linear model for aggregated features and labels has been **proven to be useful** by  existing literature.  For example, the work by Wu et al. in "Simplifying Graph Convolutional Networks" (2019) demonstrated that removing nonlinearities between GCN layers and simplifying the architecture can still achieve competitive performance. Their findings suggest that much of the representational power of GCNs arises from the network structure and feature aggregation process rather than from complex nonlinear transformations.
>
> Furthermore, the primary goal of our paper is not to propose a model that outperforms existing GCNs but to propose a statistical transfer learning method in a simplified linear network convolutional regression model. By focusing on a linear relationship, we aim to create a foundational model that is both analytically tractable and interpretable. **This simplification allows us to derive theoretical guarantees of the transfer learning process on graphs**, which will be more challenging with complex nonlinear models.
>
> In summary, while the linear assumption is indeed a limitation, it allows us to provide a clear and interpretable framework for statistical transfer learning on graphs. Our linear approach serves as a starting point that can be extended in future work to incorporate nonlinearities.

---

> > ### Author Response · Authors · 2024-11-24
> > **Response to reviewer kBsC: 2/3**
> >
> > >  **Comment:** The assumptions 4.1-4.3 for the theoretical results in Section 4 are particularly restrictive and some of them can hardly hold in practice. For example, Assumption 4.1 needs to the node attributes to follow sub-gaussian distribution, which are often not the case in real data including the real data used by this paper in the experiments. For another example, when can the sparsity parameter $s$ satisfy the particular condition in line 227-228? Furthermore, how sharp is the bound in Theorem 4.4, and how does it compare to the literature? Without a clear comparison to prior art, the significance of Theorem 4.4 is unclear.
> >
> >
> > **Response:** Thank you for your thoughtful comments. We will address each of your concerns individually.
> >
> >
> > 1. **Assumption 4.1.** It is worth noting that the majority of existing literature on ultra-high-dimensional models under the i.i.d. setup also assumes subgaussian errors (e.g., see the book [1] for details).  **Our assumptions align with standard practices in the field.** Extending our theoretical results to accommodate heavier-tailed distributions would require more sophiscated robust loss functions, such as the Huber loss.  However, given the complexity of our analysis, which operates under the challenging regime of ultra-high-dimensional dependent covariates, we have chosen to work with subgaussian errors. While extending to more geenral scenario is a valuable direction for future research, it is beyond the scope of our current study. Although our methodology will continue to work in the presence of heavy-tailed error, the rate can be suboptimal unless we change the loss function.
> >
> > 2. **Assumption 4.3.** Detailed response can be found in our general response. In summary, sparsity is a fundamental assumption in any research conducted under an ultra-high-dimensional setup. Our sparsity condition is similar to a very commonly used sparsity condition ($s \log d /n \to 0$)  for i.i.d data. In our setting, observations are correlated through a network structure characterized by the connectivity probability $p$.   To account for these dependencies, our sparsity condition includes two additional terms: (1) $\log d$, which grows very slowly even with large $d$, so its impact is modest. For example, when $d=1,000,000$, we have $\log d \approx  13.8$. (2) $\psi(p)=\frac{(1-2p)}{4p\log{\left((1-p)/p\right)}}$. When $p \to 1$, $\psi(p) \to 0$. In this case,  our sparsity condition can be less restrictive than the standard condition for i.i.d. data, potentially allowing for higher sparsity levels. In summary, our sparsity condition does not impose significantly stronger restrictions compared to the standard condition for i.i.d. data.
> >
> >
> >
> >
> >
> > 3. **Sharpness of the Bound in Theorem 4.4 and Comparison to the Literature**. The estimation error bound we derive in Theorem 4.4 has the rate  $\frac{s \log d}{n}$, which matches the convergence rate commonly obtained in the i.i.d. setting for high-dimensional models.  This indicates that our bound is sharp.  However, it is important to note that our work differs fundamentally from prior studies due to the nature of the data and the underlying assumptions. Specifically, we consider ultra-high-dimensional generalized linear models with network-induced dependencies among observations.  To the best of our knowledge, our work is the first to establish such an estimation error bound in this context. It is thus not possible to directly compare our results with prior art.
> >
> >
> >
> > [1] Bühlmann, Peter, and Sara Van De Geer. Statistics for high-dimensional data: methods, theory and applications. Springer Science & Business Media, 2011.

---

> > > ### Author Response · Authors · 2024-11-24
> > > **Response to reviewer kBsC: 3/3**
> > >
> > > >  **Comment:** Experiments are very limited, and the real graph data used in this paper are all small graphs. Experiments on graphs of much larger scale are expected to justify the effectiveness of GCR.
> > >
> > > **Response:** Thank you very much for your suggestions. In response, we have conducted additional experiments on two large-scale datasets. We utilized the ogbn-arxiv dataset from the Open Graph Benchmark (OGB), which is a large-scale, real-world graph dataset widely used in graph representation learning research. The ogbn-arxiv dataset is a directed citation network between computer science arXiv papers. To create a domain adaptation scenario, we split the dataset based on publication years: (1) ogbn-arxiv1 which contains 90,941 papers published up to 2017, and (2) ogbn-arxiv2 which  includes 78,402 papers published since 2018. By splitting the dataset in this way, we mimic a real-world setting where a model trained on historical data is applied to future data, capturing potential distribution shifts over time.
> > >
> > >
> > > We treat one subset as the source domain and the other as the target domain to evaluate the transferability of our GCR method. The performance of our method and baseline models on these large-scale graphs is summarized in the following Table R3. Note that we were unable to obtain results for AdaGCN due to its high computational cost on such large datasets. Table R3 suggests that our transfer learning method (Trans-Our) achieves the highest accuracy in both scenarios, outperforming other state-of-the-art models. These additional experiments validate the effectiveness of our GCR method on large-scale graphs, addressing the scalability concerns you raised.  We have updated the manuscript to include these new experimental results (see Table S7).
> > >
> > >
> > >
> > > Table R3: Averaged Micro F1 score (\%) of various methods for additional datasets over 10 replicates,  with source training rate fixed at 0.75 and target training rate fixed at  0.03.
> > > | Target         | Source         | **Trans-Our** | Our   | AdaGCN | UDAGCN | GPRGNN | GRAND | GCN   | APPNP | SGC   | GAT   |
> > > |----------------|----------------|----------------|-------|--------|--------|--------|-------|-------|-------|-------|-------|
> > > | ogbn-arxiv2    | ogbn-arxiv1    | **62.95**      | 59.76 |  NA     | 60.24  | 34.03  | 28.19 | 58.56 | 27.89 | 56.81 | 53.36 |
> > > | ogbn-arxiv1    | ogbn-arxiv2    | **60.48**      | 57.99 |   NA     | 57.58  | 27.78  | 29.96 | 54.35 | 20.61 | 47.93 | 41.33 |

---

> > > > ### Author Response · Authors · 2024-11-27
> > > > **Follow-Up on Review**
> > > >
> > > > I am writing to kindly inquire about the follow-up of your review. We deeply value your insights and are eager to receive your feedback on our rebuttal. We understand that reviewing takes time and effort, and we truly appreciate your dedication.
> > > >
> > > > Thank you very much for your time and contribution to the review process.

---

> > > > > ### Author Response · Authors · 2024-11-29
> > > > > **Follow-Up on Review**
> > > > >
> > > > > Dear reviewer,
> > > > >
> > > > > I am writing to kindly inquire about the follow-up of your review. We deeply value your insights and are eager to receive your feedback on our rebuttal. We understand that reviewing takes time and effort, and we truly appreciate your dedication.
> > > > >
> > > > > Thank you very much for your time and contribution to the review process.

---

### Official Review · Reviewer_rYNY · 2024-11-03

**Soundness:** 3
**Presentation:** 3
**Contribution:** 2
**Rating:** 6
**Confidence:** 5

**Summary:**

This paper tackles the challenge of node classification, where labeling nodes can be costly. The authors propose Trans-GCR, a transfer learning method using a simplified Graph Convolutional Regression (GCR) model. Unlike existing methods, Trans-GCR offers theoretical guarantees, performs well empirically, and is computationally efficient with fewer hyperparameters, making it suitable for practical use in complex graphs.

**Strengths:**

1. **Theoretical Convergence Analysis:** The authors provide a convergence analysis, offering theoretical guarantees for the convergence.

2. **Simplicity and Efficiency:** The model’s structure is both simple and fast, enhancing accessibility and scalability without compromising performance.

3. **Innovative Transfer Learning Framework:** The paper introduces a focused transfer learning framework for deep graph learning, addressing a significant gap in handling graph-structured data with limited labels, which can benefit many practical applications.

**Weaknesses:**

1. **Triviality of Convergence Proof**: The theoretical proof for convergence appears trivial. Although the paper claims that $Z$ is non-i.i.d., Assumption 4.1 indicates that $X$ is i.i.d., leading to a clear covariance matrix for $Z$ as $A^\top A / np$. Following Modification 1 in Appendix C, previous theorems can be trivially adapted with straightforward modifications. Thus, the convergence proof lacks substantial novelty and rigor.

2. **Limited Parameterization**: According to Equation 2.2, it seems that only the final logits layer contains parameters $\beta$, while the preceding $S^M X$ lacks parameters. The absence of parameters in these earlier layers raises concerns about why only the last layer is parameterized, which could lead to over-smoothing due to unparameterized iterations of $S X$ and consequently limit the model’s expressiveness.

3. **Basic Transfer Learning Approach**: The transfer learning method employed, a simple $\delta$ fine-tuning, appears overly basic. There is little exploration of alternative, established methods in transfer learning or meta-learning that could potentially enhance the model’s adaptability and robustness.

4. **Issues in Hyperparameter Sensitivity Testing**: The sensitivity experiments on hyperparameters are limited. For instance, in the $\lambda$ experiment, the model fails to achieve the optimal solution seen at $M=5$. Additionally, the range of $\lambda$ tested is narrow; a broader, exponential scale (e.g., 0.01, 0.001, 0.0001) would provide a more comprehensive understanding of the model’s sensitivity.

5. **Lack of Notational Clarity**: The notation lacks clarity and could benefit from a dedicated section outlining all definitions. Many symbols, such as $X_j$, are undefined in Appendix A. A coherent notation guide would improve readability and help readers follow the technical details more effectively.

**Questions:**

See above.

---

> ### Author Response · Authors · 2024-11-24
> **Response to reviewer rYNY: 1/3**
>
> We thank the reviewer for the dedicated and insightful review. Please see below for our response.
>
>
> >  **Comment:** **Triviality of Convergence Proof:** The theoretical proof for convergence appears trivial. Although the paper claims that $Z$ is non-i.i.d., Assumption 4.1 indicates that $X$ is i.i.d., leading to a clear covariance matrix for $Z$ as $A^TA/np$. Following Modification 1 in Appendix C, previous theorems can be trivially adapted with straightforward modifications. Thus, the convergence proof lacks substantial novelty and rigor.
>
> **Response:** Thank you for your comment. We will respond to each of your points individually.
>
> 1. *Covariance matrix of $Z$.* We would like to clarify that $A^TA/np$ is **not related** to the covariance matrix of $Z$. To see this, note that each observation $Z_i$ is $d$-dimensional, so the covariance matrix of $Z$ is of size $d \times d$, whereas $A^TA/np$ is $n \times n$. Additionally, $Z$ is obtained by **pre-multiplying** $X$ with $A$, not **post-multiplying**. Therefore, **the covariance matrix of $Z$ is not equal to $A^TA/np$**. The pre-multiplication by $A$ introduces dependence among the rows $X_1, \dots, X_n$, not the columns of $X_i$, which significantly complicates the analysis.
>
> 2. *Following analysis after modification 1.* While Modification 1 addresses certain aspects of the analysis, it is not the only adjustment required.  In fact, **Modification 3 involves  the most substantial effort**. A central challenge is that the observations ${Z_i}, {1 \leq i \leq n}$ are dependent. One of the critical aspects of analyzing high-dimensional models is establishing the *restricted strong convexity/restricted eigenvalue* condition (e.g., see [1]). Even for i.i.d. or independent observations, proving this condition has only recently been achieved through considerable work (e.g., see [2], [3]). For our case, the dependence among ${Z_i}, {1 \leq i \leq n}$ makes this significantly more challenging. Existing techniques for time series data with mixing conditions are not directly applicable due to the special network-type dependency structure in our problem. Specifically, the dependencies are governed by the network's adjacency matrix $A$, which introduces complex correlations that are neither time-based nor easily characterized by standard mixing conditions. To address this, we developed novel techniques to handle the network dependencies and to establish the RSC condition for our model. This forms a critical part of our proof, particularly in Lemma C.2.
>
> [1] Bickel, Peter J., Ya’acov Ritov, and Alexandre B. Tsybakov. "Simultaneous analysis of Lasso and Dantzig selector." (2009): 1705-1732.
>
> [2] Rudelson, Mark, and Shuheng Zhou. "Reconstruction from anisotropic random measurements." Conference on Learning Theory. JMLR Workshop and Conference Proceedings, 2012.
>
> [3] Kasiviswanathan, Shiva Prasad, and Mark Rudelson. "Restricted isometry property under high correlations." arXiv preprint arXiv:1904.05510 (2019).
>
>
>
> >  **Comment:** **Limited Parameterization:** According to Equation 2.2, it seems that only the final logits layer contains parameters $\beta$, while the preceding $S^MX$ lacks parameters. The absence of parameters in these earlier layers raises concerns about why only the last layer is parameterized, which could lead to over-smoothing due to unparameterized iterations of $SX$ and consequently limit the model’s expressiveness.
>
> **Response:** Thank you for the insightful commets. Equation 2.2 is motivated by [1], which simplifies GCN by removing nonlinerity between GCN layers. Thus the $M$-layer GCN is simplified as $\hat{Y} = Softmax(SS \cdots S X \beta^{(1)}\beta^{(2)}\cdots\beta^{(M)})$. These weight parameters can be reparametrized as a single weight matrix $\beta= \beta^{(1)}\beta^{(2)}\cdots\beta^{(M)}$.
>
> By reparametrizing the multiple parameters into one, we effectively  capture the cumulative effect of parameterization across all iterations without explicitly parameterizing each step. This approach helps in reducing the complexity of the model and prevents issues like over-smoothing or overfitting that could arise from excessive parameterization,  especially in scenarios with limited training data.
>
> [1] Felix Wu, Amauri Souza, Tianyi Zhang, Christopher Fifty, Tao Yu, and Kilian Weinberger. Simplifying graph convolutional networks. In International conference on machine learning, pp. 6861–6871. PMLR, 2019.

---

> > ### Author Response · Authors · 2024-11-24
> > **Response to reviewer rYNY: 2/3**
> >
> > >  **Comment:** **Basic Transfer Learning Approach:** The transfer learning method employed, a simple $\delta$ fine-tuning, appears overly basic. There is little exploration of alternative, established methods in transfer learning or meta-learning that could potentially enhance the model’s adaptability and robustness.
> >
> > **Response:** Although our $\delta$ fine-tuning method may appear straight-forward, we would like to highlight that we are operating under ultra-high-dimensional regime, where the dimension $d$ can be much larger than $n$; in fact, it can be as large as $e^{n^c}$ for some $0 < c < 1$. In this ultra-high-dimensional regime, more complicated methods are often prone to overfitting, and consequently, simple methods are preferred.
> >
> > Furthermore, theoretical analysis of this simple $\delta$ fine-tuning based method, even in the case of simple linear regression in an ultra-high-dimensional setup, was done very recently in [1]. Our analysis is even more difficult as we allow dependence among the observations, and consequently, no standard techniques for analyzing the i.i.d. data are readily applicable. Therefore, the $\delta$ fine-tuning method, though seem straightforward at first, requires a significant amount of work to operationalize it and establish theoretical guarantees as the dimension of the underlying observations is very high.
> >
> > In summary, while more sophisticated methods could be explored, they often come with increased computational costs and a higher risk of overfitting in ultra-high-dimensional contexts with dependent data. Our approach provides a solid foundation and demonstrates that even seemingly simple methods can yield powerful results when carefully adapted to the complexities of the data.
> >
> > We acknowledge the potential for further advancements and plan to investigate more complex methodologies in future work. However, we believe that our current contributions significantly advance the understanding and application of statistical methods in ultra-high-dimensional settings with network dependencies.
> >
> > [1] Li, Sai, T. Tony Cai, and Hongzhe Li. "Transfer learning for high-dimensional linear regression: Prediction, estimation and minimax optimality." Journal of the Royal Statistical Society Series B: Statistical Methodology 84.1 (2022): 149-173.

---

> > > ### Author Response · Authors · 2024-11-24
> > > **Response to reviewer rYNY: 3/3**
> > >
> > > >  **Comment:** **Issues in Hyperparameter Sensitivity Testing:** The sensitivity experiments on hyperparameters are limited. For instance, in the $\lambda$ experiment, the model fails to achieve the optimal solution seen at $M=5$. Additionally, the range of $\lambda$ tested is narrow; a broader, exponential scale (e.g., 0.01, 0.001, 0.0001) would provide a more comprehensive understanding of the model’s sensitivity.
> > >
> > >
> > > **Response:** Thank you for your valuable feedback. Following your suggestion, we have considered broader range of $\lambda$, i.e., 0.00005, 0.0001, 0.0005, 0.001, 0.0015,  0.01 and 0.1. The comprehensive results are presented in Table R2. As we can see when $\lambda$ is within 0.0001 and  0.0015, the performance is relatively stable. When $\lambda$ is as small as 0.00005 or as large as 0.01, the performance is poor. This is becasue $\lambda$ controls the trade-off between bias and variance.  When $\lambda$ is large, many coefficients are shrunk to zero, we may risk underfitting, where the model may miss important relationships in the data. When $\lambda$ is small, many  non-relevant variables are also included, we may risk ovefitting.
> > >
> > > It is worth noting that $\lambda$ plays a role similar to the regularization strength in Lasso regression, and our findings align with well-established results in the Lasso literature. As highlighted in prior studies [1], excessively small or large values of $\lambda$ should be avoided in practice. Following this principle, we recommend selecting $\lambda$ within a range that balances bias and variance effectively. In the revised manuscript, we have uoadted Table S6 to include these results.
> > >
> > >
> > > Table R2: Averaged Micro F1 score (\%) of our proposed Trans-GCR for varying $\lambda$, over 10 replicates,  with source training rate fixed at 0.75 and target training rate fixed at 0.03, while fixing $M=5$.
> > >
> > > | Target | Source  | $\lambda=0.00005$ | $\lambda=0.0001$ | $\lambda=0.0005$ | $\lambda=0.001$ | $\lambda=0.0015$ | $\lambda=0.01$ | $\lambda=0.1$ |
> > > |--------|---------|-------------------|------------------|------------------|-----------------|------------------|---------------|--------------|
> > > | D      | C       | 73.19            | 75.32           | 73.76           | 76.73          | 75.12           | 59.71        | 21.57        |
> > > |        | A       | 72.45            | 73.82           | 75.14           | 74.57          | 75.38           | 56.17        | 21.96        |
> > > |        | C & A   | 71.32            | 74.63           | 77.08           | 74.67          | 73.82           | 58.20        | 21.47        |
> > > | C      | D       | 78.26            | 79.18           | 79.96           | 78.91          | 78.68           | 65.53        | 25.62        |
> > > |        | A       | 77.16            | 80.24           | 81.17           | 79.27          | 79.39           | 67.63        | 25.01        |
> > > |        | D & A   | 77.64            | 80.45           | 80.85           | 80.39          | 79.98           | 65.61        | 25.47        |
> > > | A      | D       | 70.69            | 67.91           | 70.64           | 72.41          | 72.97           | 58.15        | 20.33        |
> > > |        | C       | 70.02            | 73.16           | 72.37           | 74.18          | 74.08           | 59.43        | 20.23        |
> > > |        | D & C   | 70.44            | 72.59           | 73.82           | 73.18          | 72.01           | 58.76        | 20.34        |
> > >
> > >
> > > [1] Tibshirani, R. (1996). Regression shrinkage and selection via the lasso. Journal of the Royal Statistical Society Series B: Statistical Methodology, 58(1), 267-288.
> > >
> > > >  **Comment:** **Lack of Notational Clarity:** The notation lacks clarity and could benefit from a dedicated section outlining all definitions. Many symbols, such as $X_j$, are undefined in Appendix A. A coherent notation guide would improve readability and help readers follow the technical details more effectively.
> > >
> > > **Response:** Thank you for the comments. The definition of $X_j$ can be found in Line 129 in the original manuscript. Specifically, $X_j=(X_{j1}, \ldots,X_{jd})$ is the $j$th row of $X$.  To enhance readability, we have updated the notation table (Table S1) to include definitions for $X_j$.

---

> > > > ### Comment · Reviewer_rYNY · 2024-11-26
> > > >
> > > > The results for $\lambda=0.001$ appear to be inconsistent between the original version and the revised version. Could you clarify the reason for this discrepancy?

---

> > ### Comment · Reviewer_rYNY · 2024-11-26
> >
> > As $M$ increases, $S^M$ converges toward a matrix of all ones, making $X$ indistinguishable and resulting in a loss of meaningful differentiation. To address this issue, I suggest adopting a method similar to APPNP, which employs a formulation such as $\alpha \circ S \alpha \circ S \cdots \alpha \circ S$, where $\alpha$ is a scalar hyperparameter with $\alpha < 1$. This approach ensures better control over the propagation process and might mitigate the indistinguishability problem.

---

> ### Author Response · Authors · 2024-11-26
> **Response to Reviewer rYNY**
>
> >  **Comment about $\mathbf{S}^M$**
>
> Response: Thank you for pointing out the potential issue with $\mathbf{S}^M$ converging to a matrix of all ones as $M$ increases and for suggesting the APPNP-inspired formulation using $\alpha < 1$ to mitigate this indistinguishability problem.
>
> We agree that incorporating such a mechanism could provide better control over the propagation process. We acknowledge its importance and plan to explore this direction in future research. We will include the following statement in the discussion section (section 7) of the manuscript.
>
> " As $M$ increases, $\mathbf{S}^M$ converges to a matrix of ones, resulting in oversmoothing, where node representations become indistinguishable. Future work could explore adaptive propagation mechanisms, such as APPNP, to mitigate oversmoothing and preserve meaningful differentiation between node representations in deeper architectures. "
>
> Thank you for your thoughtful feedback.
>
> >  **Comment about $\lambda$**
>
> Thank you for your comment. The differences arise because we reran the experiments during the revision process. Each result is averaged across 10 independent replications, where the training, validation, and testing splits differ for each replication. This introduces some natural variability, which accounts for the numerical differences observed.
>
> Despite numerical differences, the optimal $\lambda$ values remain largely consistent between the original and revised versions, as shown in the following table.  We have three observations: (1) For 6 out of 9 tasks, the optimal $\lambda$ values remain identical ($0.0005$ or $0.0015$) between the original and new results.  (2) In three tasks ($D \rightarrow C$, $D \rightarrow A$, and $A \rightarrow C$), minor differences in the optimal $\lambda$ (between $0.0005$ and $0.0015$) ) are observed. These variations are likely due to the fine-scale sensitivity of these specific configurations to random data splits during replication. (3)  Tasks involving multiple source domains  demonstrate perfect stability, consistently maintaining $\lambda = 0.0005$ across both versions.
>
> Table: Comparison of optimal lambda between the old version and the new version.
> | Target | Source  | Optimal Lambda (old) | Optimal Lambda (new) |
> |--------|---------|-----------------------|-----------------------|
> | D      | C       | 0.0005                | 0.001               |
> | D      | A       | 0.0005               | 0.0015               |
> | D      | C & A   | 0.0005               | 0.0005               |
> | C      | D       | 0.0005               | 0.0005               |
> | C      | A       | 0.0005               | 0.0005               |
> | C      | D & A   | 0.0005               | 0.0005               |
> | A      | D       | 0.0015               | 0.0015               |
> | A      | C       | 0.0005                | 0.001               |
> | A      | D & C   | 0.0005               | 0.0005               |

---

> > ### Comment · Reviewer_rYNY · 2024-11-27
> >
> > Thank you for your response. All of my concerns have been addressed. I increased my score.

---

> > > ### Author Response · Authors · 2024-11-27
> > > **Thank you!**
> > >
> > > We sincerely appreciate your valuable insights and are glad the responses have addressed your concerns. Please feel free to share any additional questions or suggestions. Thank you for your thoughtful feedback and time!

---

### Official Review · Reviewer_9h22 · 2024-11-04

**Soundness:** 2
**Presentation:** 2
**Contribution:** 2
**Rating:** 6
**Confidence:** 3

**Summary:**

This paper addresses the challenge of node classification in high-dimensional settings, particularly in scenarios with limited labeled data. It introduces a novel transfer learning method called Trans-GCR, based on a Graph Convolutional Multinomial Logistic Lasso Regression (GCR) model. The GCR model assumes that classification labels depend on graph-aggregated node features followed by a multinomial logistic regression, allowing for effective handling of high-dimensional features. The authors highlight the limitations of existing GCN-based transfer learning methods, which often lack theoretical guarantees, are sensitive to hyperparameters, and impose restrictive conditions. In contrast, Trans-GCR provides theoretical guarantees under mild conditions and demonstrates superior empirical performance with a lower computational cost. The method requires only two hyperparameters, making it more accessible for practical applications.

**Strengths:**

1. The paper is well-written and presents its ideas clearly.
2. The proposed Trans-GCR method appears to be a sensible solution to the challenges of graph transfer learning, addressing the issue of node classification effectively. Additionally, the authors provide theoretical guarantees for their method under specific conditions, enhancing the robustness of their approach.

**Weaknesses:**

1. Some parts of the paper are confusing. In line 260, the authors state that $p$ is the expected degree, but according to the definition in Assumption 4.2, the network connectivity parameter $p$ should be a probability. This inconsistency may leads to misunderstanding.
2. I find the condition $p \log d \to 0 $ as $ n \to \infty $ in Assumption 4.2 somewhat perplexing. It would be helpful for the authors to explain why the feature dimension $d$ is relevant when defining network connectivity.
3. Since one of the contributions of the paper is providing theoretical guarantees, and the authors have made several modifications to the proof approach compared to previous work, a brief sketch of these changes would be helpful for readers to understand the arguments better.
4. On synthetic data, since the true $\beta$ values and sparse patterns are known, using MSE as the evaluation metric is insufficient. It would be more comprehensive to consider additional metrics such as True Positive Rate (TPR) and False Discovery Rate (FDR) to assess the model's performance.

**Questions:**

1. Please address the concerns mentioned in the Weaknesses.

2. Is Assumption 4.3 a technical condition? Specifically, does the term $s \frac{\log d}{n} \times \log d \times \psi(p)$ provide additional insight compared to the condition $s \frac{\log d}{n}$?

3. What are the technical challenges in proving results for the SBM in the context of this paper compared to the ER graph? Specifically, under conditions similar to Assumption 4.2, given that $p, q = \omega(\log n / n)$, what complexities arise in this setting?

---

> ### Author Response · Authors · 2024-11-24
> **Response to reviewer 9h22: 1/3**
>
> We thank the reviewer for the dedicated and insightful review. Please see below for our response.
>
>
> >  ***Comment:** In line 260, the authors state that $p$ is the expected degree, but according to the definition in Assumption 4.2, the network connectivity parameter $p$ should be a probability.*
>
> **Response:** Thank you for pointing out this typo. We have corrected it in the revised manuscript. In line 260,  $p$ is the network connectivity probability, and  $np$ is the expected degree.
>
> >  **Comment:** I find the condition $p\log d \to 0$ as $n \to \infty$ in Assumption 4.2 somewhat perplexing. It would be helpful for the authors to explain why the feature dimension $d$ is relevant when defining network connectivity.
>
> **Response:** Thank you for your insightful comment! The main reason for imposing the condition on the dimension is that we **do not** assume the dimension d to be fixed; instead, d can be significantly larger than the sample size n. Our theoretical results are established in an ultra-high dimensional regime, where d can grow as large as $e^{n^c}$ for some $c < 1$. Therefore, to take care of this growing dimension, we require this additional condition. When the dimension is fixed or grows slowly, the condition $p\log{d} \to 0$ simplifies to $p \to 0$, aligning with the standard framework in network regression analysis.
>
> In an ultra-high dimensional feature space, the impact of each additional feature can amplify the complexity of the network structure. Thus, to ensure that the network remains sufficiently sparse for our theoretical results to hold, $p$ must decrease at a rate that compensates for the growth in $\log d$. The condition $p\log d \to 0$ as $n \to \infty$ ensures that, despite the increasing number of features, the overall connectivity remains controlled. Thank you again for pointing this out, and we hope this clarification helps address your concern.
>
> >  **Comment:** Since one of the contributions of the paper is providing theoretical guarantees, and the authors have made several modifications to the proof approach compared to previous work, a brief sketch of these changes would be helpful for readers to understand the arguments better.
>
> **Response:**  Thank you very much for your comment! We have now included a roadmap in the Appendix (Section B) that highlights the key differences and modifications made in the proof to account for the dependency among the observations.

---

> > ### Author Response · Authors · 2024-11-24
> > **Response to reviewer 9h22: 2/3**
> >
> > >  **Comment:** On synthetic data, since the true $\beta$ values and sparse patterns are known, using MSE as the evaluation metric is insufficient. It would be more comprehensive to consider additional metrics such as True Positive Rate (TPR) and False Discovery Rate (FDR) to assess the model's performance.
> >
> > **Response:** Thank you for the insightful comment. Following your suggestion, we have included TPR and FDR as additional evaluation metric. The following Table R1 summarizes the performance of three methods (Trans-GCR, GCR, and Naive TL) in terms of True Positive Rate (TPR) and False Discovery Proportion (FDP) for variable selection in simulation data, under varying source data sample sizes.  Specifically, FDP is calculated as the proportion of incorrectly selected variables  among all selected variables, while TPR is calcualted as the proportion of true variables  that are correctly identified by the method.
> >
> > From Table R1, We have two observatoions. (1) In terms of FDP, Trans-GCR demonstrates superior robustness by maintaining substantially lower FDP values compared to both GCR and Naive TL, particularly as the sample size increases. (2) In terms of  TPR, Trans-GCR consistently outperforms GCR across all sample sizes, with significant improvements in sensitivity to identifying relevant variables, particularly at smaller sample sizes. Naive TL achieves the highest TPR across all sample sizes, indicating strong sensitivity; however, this comes at the cost of higher FDP values compared to Trans-GCR.
> >
> >
> > In summary, Trans-GCR achieves the best performance in  maintaining strong variable selection capabilities (higher TPR) while minimizing false discoveries (lower FDP). We have included this table in  Table S2 in  the updated manuscrit.
> >
> >
> > Table R1: Performance comparison on averaged true positive rate (TPR) and false discovery proportion (FDP) for varying source data sample size, over 10 replicates. TPRs and FDPs were calculated overall for coefficient matrices.
> >
> >
> > | Method      | Metric | 100   | 200   | 300   | 400   | 500   | 600   | 700   | 800   | 900   | 1000  |
> > |-------------|--------|-------|-------|-------|-------|-------|-------|-------|-------|-------|-------|
> > | Trans-GCR   | TPR    | 0.478 | 0.496 | 0.556 | 0.572 | 0.578 | 0.582 | 0.758 | 0.762 | 0.712 | 0.714 |
> > |             | FDP    | 0.378 | 0.342 | 0.275 | 0.263 | 0.133 | 0.123 | 0.237 | 0.134 | 0.164 | 0.112 |
> > | GCR         | TPR    | 0.072 | 0.054 | 0.032 | 0.050 | 0.072 | 0.042 | 0.050 | 0.022 | 0.066 | 0.026 |
> > |             | FDP    | 0.521 | 0.739 | 0.813 | 0.613 | 0.452 | 0.625 | 0.523 | 0.546 | 0.567 | 0.715 |
> > | Naive TL    | TPR    | 0.660 | 0.664 | 0.904 | 0.930 | 0.972 | 0.976 | 0.982 | 0.984 | 0.962 | 0.986 |
> > |             | FDP    | 0.717 | 0.715 | 0.722 | 0.717 | 0.606 | 0.677 | 0.667 | 0.668 | 0.662 | 0.669 |
> >
> >
> > >  **Comment:** Is Assumption 4.3 a technical condition? Specifically, does the term $s\frac{\log d}{n} \times\log d\times\psi(p)$ provide additional insight compared to the condition $s\frac{\log d}{n}$?
> >
> >
> > **Response:** The condition $(s\log{d})/n \to 0$ is a standard requirement in the analysis of ultra high-dimensional models under independent and identically distributed observations. However, in our scenario, the observations are correlated through the network, with the connectivity parameter being $p$. This dependency affects the concentration inequalities and the convergence rates in our theoretical analysis.
> >
> >
> > Consequently, it is intuitive that the above-mentioned condition requires a suitable modification to account for the dependency introduced by the network structure. The parameter $\psi(p)$ represents the sub-Gaussian constant of a Bernoulli random variable with success probability $p$. The additional $\log{d}$ factor arises due to the combination of two facts; i) we do not know which $s$ coordinates are active, and ii) errors are assumed to be subgaussian. Even in cases of heavy-tailed errors, one can still get this $\log{d}$ factor by using robust (e.g., Huber) loss function.
> >
> >
> > In summary, the additional $\psi(p)\log{d}$ term reflects the effect of data dependency in the model. Thank you for raising this insightful question, and we hope this explanation addresses your concern.

---

> > > ### Author Response · Authors · 2024-11-24
> > > **Response to reviewer 9h22: 3/3**
> > >
> > > >  **Comment:** What are the technical challenges in proving results for the SBM in the context of this paper compared to the ER graph? Specifically, under conditions similar to Assumption 4.2, given that $p,q=\omega(\log n/n)$, what complexities arise in this setting?
> > >
> > > **Response:**  Thank you for your thoughtful question. You are absolutely correct that extending the results from ER graphs to SBM graphs is a natural next step, and this is indeed part of our ongoing work. One of the central technical challenges in deriving the rate of estimation in ultra-high-dimensional models is establishing the Restricted Strong Convexity (RSC) condition. In essence, the RSC condition ensures that the sample covariance matrix, which is low rank due to the ultra-high-dimensional setting ($d \gg n$), has its minimum eigenvalue bounded away from zero in certain directions.
> > >
> > > For i.i.d. samples, standard results link the concentration of the sample covariance matrix to its population counterpart, which can be leveraged to establish the RSC condition. However, when data dependency is introduced —- especially as a result of the network structure -- such results are not readily available. Specifically, for network-based dependency, no established RSC results currently exist. As a result, we had to derive this condition by applying advanced tools from random matrix theory.
> > >
> > > The ER graph offers a degree of analytical tractability because the edges are i.i.d., simplifying the derivation of key results. In contrast, the SBM framework introduces additional complexities because the edges are no longer i.i.d.. This dependency requires even more sophisticated techniques from random matrix theory to address. Furthermore, additional challenges arise in modifying standard concentration inequalities, such as sub-Gaussian concentration bounds, to account for these dependencies. We appreciate your insightful question and hope this explanation clarifies the specific challenges involved and the steps we are taking to address them.

---

> > > > ### Author Response · Authors · 2024-11-27
> > > > **Follow-Up on Review**
> > > >
> > > > I am writing to kindly inquire about the follow-up of your review. We deeply value your insights and are eager to receive your feedback on our rebuttal. We understand that reviewing takes time and effort, and we truly appreciate your dedication.
> > > >
> > > > Thank you very much for your time and contribution to the review process.

---

> > > > ### Comment · Reviewer_9h22 · 2024-11-28
> > > >
> > > > Thank you for your response. The concerns regarding the assumption and experimental metrics on synthetic data have been addressed. However, I still have some doubts about whether extending the work to SBM would be trivial. Based on the assumptions made in the paper, a more suitable testbed model would be the CSBM. There have already been significant results in CSBM research, including concentration results at different levels of sparsity, as well as phase transition conditions defined by $p, q, d, n$. According to my intuition, most of the challenging proofs have already been covered by these papers. Could you explain why you didn't consider directly using the CSBM when initially writing this paper, and what difficulties you encountered?

---

> > > > > ### Author Response · Authors · 2024-11-28
> > > > > **Response to Reviewer 9h22**
> > > > >
> > > > > Thank you very much for your insightful comment. While it is interesting to extend our work to the SBM or CSBM, it requires substantial revisions and theoretical challenges.
> > > > >
> > > > > Specifically, in our theoretical analysis, we leverage the assumption that edges are independently and identically distributed as $\text{Bernoulli}(p)$ to establish the **restricted strong convexity (RSC)** condition, which is crucial for proving the consistency of our estimator in the ultra-high-dimensional regime (see the proof of Lemma C.1). However, in SBM, although the edges of $\mathbf{A}$ are independent, they are not identically distributed across different communities. This non-identical distribution necessitates careful adjustments in all relevant steps of our proofs (e.g., equation (D.4)) to account for the varying edge probabilities. Furthermore, extending our analysis to SBM involves community detection estimation. The estimation errors in identifying community structures would propagate through our framework, influencing critical parameters such as the upper bound of $\lambda_\beta$ in our proofs. Incorporating these additional sources of error complicates the error analysis, making it challenging to determine which errors dominate and how they interact with other components of the proof.
> > > > >
> > > > > In CSBM, an additional complication arises because $\mathbf{X}$ and $\mathbf{A}$ are not independent. This correlation requires a nearly completely different revision of our proofs for Lemmas C.1 and C.2, as these proofs heavily rely on conditioning on $\mathbf{X}$, which is not directly applicable in the context of SBM.
> > > > >
> > > > > Given the length and complexity of the current proofs, we have chosen to focus on the ER graph as a starting point to establish the theoretical foundation. In summary, while we agree that extending our proofs to SBM or CSBM is feasible with additional technical effort, such an extension involves significant revisions that are beyond the scope of the present work and will be addressed in future research.

---

> > > > > > ### Comment · Reviewer_9h22 · 2024-11-29
> > > > > >
> > > > > > Thank you for your response. My goal is not to suggest a direct transition to CSBM analysis during the rebuttal phase, as it also presents certain technical challenges, and it is indeed outside the scope of this paper. Rather, I aim to provide a more cautious evaluation of the paper’s novelty.
> > > > > >
> > > > > > Most of my concerns have been addressed. I will increase my score.

---

> > > > > > > ### Author Response · Authors · 2024-11-29
> > > > > > > **Thank you!**
> > > > > > >
> > > > > > > We sincerely appreciate your thoughtful feedback and the time you dedicated to carefully evaluating our work. Please feel free to let us know if you have any further questions or suggestions. Thank you again for your time and effort!

---

### Author Response · Authors · 2024-11-24
**Response to all reviewers 1/2**

We thank all four reviewers for their valuable feedback. Below we address concerns shared by Reviewers 9h22, kBsC and wstE.



**1. Sparsity condition in Assumption 4.3.**

(1) **Necessity of the Sparsity Assumption in Ultra-High Dimensions.** In Assumption 4.3 we assume that the true data generating parameter $\mathbb{\beta}^{(0)}$ is $s$-sparse, i.e. only $s$ many coordinates of $\mathbb{\beta}^{(0)}$ are non-zero. Sparsity is a fundamental assumption in any research conducted under an ultra-high-dimensional setup, where the dimension $d$ of $\mathbb{\beta}^{(0)}$ can be significantly larger than the sample size n, i.e., it can be as large as $e^{n^c}$ where $c < 1$. To illustrate the necessity of this assumption, consider a simple scenario when we want to solve a linear regression problem with $n = 100$ samples with $d = 1000$ covariates. It is intuitive that we cannot hope to estimate the coefficients of all $1000$ covariates consistently only using $100$ samples. In that case, the standard assumption is that among these $1000$ covariates, only few of them are *active*, i.e. have non-zero coefficients. Then the problem to identify and estimate those active coefficients is amenable and consistent estimation is also possible. For more details, readers may consider the books [1, 2].

(2) **Common  Standard Sparsity Conditions.** Specifically, we require that $\textstyle
        s\frac{\log^2{d}}{n}\psi(p) \to 0$, where $\psi(p)=\frac{(1-2p)}{4p\log{\left((1-p)/p\right)}}$.

First, in standard linear regression for i.i.d data, a common assumption on the sparsity parameter is $s\log{d}/n \to 0$. Ignoring the $\log{d}$ which is often quite small even if $d$ is very large, the condition roughly indicates that $s/n \to 0$, i.e., the number of active covariates should be smaller than the sample size. This is a necessary condition for consistent estimator of the parameters as illustrated in the previous paragraph; if the number of active covariates is too large, then again consistent estimation is not possible. Furthermore this condition is minimax optimal for linear regression in presence of i.i.d. data, as established in [3].

Second, in our scenario, the observations are correlated through the network, with the connectivity parameter being $p$. This dependency affects the concentration inequalities and the  convergence rate in our theoretical analysis. Consequently, it is intuitive that the above-mentioned condition, i.e., $s\log{d}/n \to 0$, requires a suitable modification to account for the dependency introduced by the network structure. The parameter $\psi(p)$ represents the sub-Gaussian constant of a Bernoulli random variable with success probability $p$. The additional $\log{d}$ factor arises due to the combination of two facts; i) we do not know which $s$ coordinates are active, and ii) errors are assumed to be subgaussian. Even in cases of heavy-tailed errors, one can still get this $\log{d}$ factor by using robust (e.g., Huber) loss function.

In summary, the additional $\psi(p)\log{d}$ term reflects the effect of data dependency in the model. In other words, this is the price we pay for *not knowing* the indices/positions of active covariates.

[1] Bühlmann, Peter, and Sara Van De Geer. Statistics for high-dimensional data: methods, theory and applications. Springer Science & Business Media, 2011.

[2] Hastie, Trevor, Robert Tibshirani, and Martin Wainwright. "Statistical learning with sparsity." Monographs on statistics and applied probability 143.143 (2015).

[3] Raskutti, Garvesh, Martin J. Wainwright, and Bin Yu. "Minimax rates of estimation for high-dimensional linear regression over $\ell_q$-balls." IEEE transactions on information theory 57.10 (2011): 6976-6994.

---

> ### Author Response · Authors · 2024-11-24
> **Response to all reviewers 2/2**
>
> **2. Summary of revisions in the paper:**
>
> **Main text**:
>
> (1) **Additional Method for Comparison**: We have included an additional method, Pretraining GNNs [1], to provide a more comprehensive comparison in our experiments.
>
> (2) **Updated Table 1**: We have updated Table 1 by implementing each method using cross-validation to select optimal hyperparameter settings, ensuring a fair and rigorous evaluation.
>
> (3) **Typo Correction**: A typo regarding the parameter $p$ on page 5 has been corrected.
>
> **Appendix**:
>
> (1) **Clarification of Notation:** An explanation of the notation $\mathbf{X}_j$ has been added to Table S1 to enhance clarity.
>
> (2) **Proof Roadmap:** Section B has been added to include a roadmap of the proof, providing readers with a structured overview of our theoretical analysis.
>
> (3) **Additional Evaluation Metrics:** We have added the Table S2 which includes two additional evaluation metrics—True Positive Rate (TPR) and False Discovery Proportion (FDP)—to offer a more detailed assessment of model performance.
>
> (4) **Extended Hyperparameter Range:** We have updated Table S6 to explore a broader, exponential range of the regularization parameter $\lambda$, specifically the values: 0.00005, 0.0001, 0.0005, 0.001, 0.0015, 0.01, and 0.1.
>
> (5) **Scalability Investigation:** We have added the Table S7, which includes two additional large datasets to investigate the scalability of our method, demonstrating its effectiveness on a larger scale.
>
>
> [1]  W Hu, B Liu, J Gomes, M Zitnik, P Liang, V Pande, and J Leskovec. Strategies for pre-training graph neural networks. In International Conference on Learning Representations (ICLR), 2020.

---

### Note · Authors · 2025-01-22

I have read and agree with the venue's withdrawal policy on behalf of myself and my co-authors.